# European Space Agency Dataset and Benchmark for Anomaly Detection in Real-World Time Series

**Krzysztof Kotowski**[1,*]   **Christoph Haskamp**[2,*]   **Jacek Andrzejewski**[1]
**Bogdan Ruszczak**[3,1]   **Jakub Nalepa**[4,1]   **Daniel Lakey**[5]   **Peter Collins**[6]
**Jose Martínez-Heras**[7]   **Gabriele De Canio**[6,*]
[1]KP Labs   [2]Airbus Defence and Space   [3]Opole University of Technology
[4]Silesian University of Technology   [5]CGI   [6]European Space Agency
[7]Solenix

## Abstract

Time series from spacecraft sensors are high-dimensional, nonstationary, nonlinear, irregularly sampled, and exhibit both spatial and temporal dependencies. Detecting anomalies in such signals is critical for both on-ground and in-orbit space operations. The potential of machine learning in this task is currently hampered by a lack of comprehensive datasets and benchmarks that capture its real-world complexity. The European Space Agency Benchmark for Anomaly Detection (ESA-ADB) addresses this issue and establishes a new standard in the domain. It is a result of close cooperation between engineers from the European Space Operations Center and machine learning experts from industry and academia. Our newly introduced dataset (zenodo.org/records/15237121) contains several years of real-life raw data from 3 large spacecraft, including 224 channels, 821 control signals, and 1430 annotated events, which makes it the biggest dataset of its kind in the literature. The associated benchmark defines 9 specific requirements and 5 evaluation metrics for assessing anomaly detection algorithms in operational practice. The results indicate that widely used anomaly detection algorithms, even with our proposed adaptations, are not yet suitable for effective deployment. Thus, ESA-ADB remains an open challenge, being further explored through a dedicated Kaggle competition (kaggle.com/competitions/esa-adb-challenge).

## 1   Introduction

Monitoring anomalies in time series data from spacecraft sensors (spacecraft telemetry) is a daily practice of thousands of spacecraft operations engineers (SOEs) in mission control centers worldwide. It ensures safe and uninterrupted operations of multiple scientific, communication, observation, and navigation satellites. SOEs are typically supported by simple automatic anomaly detection systems that alarm when a measurement falls outside its predefined nominal limits or when a measurement correlates with a known anomalous pattern [1]. However, more sophisticated anomalies are usually detected manually, which is a very expensive and error-prone task [2]. For this reason, all major space-related entities have been actively researching, developing, and testing advanced automatic anomaly detection systems in recent years, including space agencies from Europe [3–6], USA [7], Canada [8], Korea [9], and Japan [10], and multiple private companies [11–13]. It is also a prioritized domain of the Artificial Intelligence for Automation Roadmap of the European Space Agency (ESA) [14], and there is a growing trend in applying such systems directly onboard spacecraft for faster alarming and autonomous operations [15]. However, spacecraft telemetry is an especially complex example of multivariate time series data of high dimensionality and volume (years of recordings from up to

---

*Corresponding authors: kkotowski@kplabs.pl, christoph.haskamp@airbus.com, gabriele.decanio@esa.int

Submitted to 39th Conference on Neural Information Processing Systems (NeurIPS 2025). Do not distribute.

thousands of channels per spacecraft [16]), complex characteristics (i.e., nonstationarity, nonlinearity, spatiotemporal dependencies, varying sampling frequencies, and data gaps), diverse data types (i.e., large variety and ranges of physical measures, categorical status flags, counters, and binary telecommands), and inherent noise related to the space environment.

**Related work.** There are hundreds of algorithms for time series anomaly detection (TSAD) proposed in the literature (158 according to Schmidl et al. [17]) that could be viable solutions for spacecraft telemetry, but currently, the main challenge is the evaluation of different approaches. This occurs because there are relatively few anomalies in flying spacecraft [2] and no comprehensive data collection from multiple sources. Thus, it is difficult to objectively conclude that one approach works better than the other. Moreover, multiple recent papers show that many publicly available datasets, benchmarks, and metrics for TSAD are flawed and cannot be used for an unbiased evaluation of emerging machine learning (ML) techniques, especially in complex real-world settings [18–21]. Specifically, the most popular NASA SMAP and MSL datasets of spacecraft telemetry [7] are too trivial and have unrealistic anomaly density, inconsistent ground truth, and run-to-failure bias [20]. There are a few TSAD benchmarks that avoid these flaws, but they are either univariate [20], artificial [22], or do not represent complexities of real systems (varying sampling rates, different channel types, or real-time processing). See Appendix 2.6 for detailed analysis of related datasets and benchmarks.

**Contributions.** The proposed **European Space Agency Benchmark for Anomaly Detection (ESA-ADB)** directly addresses all the mentioned issues and establishes a new standard of validating algorithms for anomaly detection in real-world time series from spacecraft. It is a result of close cooperation between SOEs from the European Space Operations Center (ESOC) and ML experts from industry and academia. ESA-ADB consists of three main components (Figure 1):

1. **ESA Anomalies Dataset (ESA-AD)** – a large-scale, curated, and structured collection of real-world spacecraft telemetry data, collected from **3 ESA missions** and annotated by SOEs and ML experts.

2. **Evaluation pipeline** designed by ML experts for the practical needs of SOEs. It introduces a list of **9 requirements** and **5 metrics** designed for real-world spacecraft telemetry anomaly detection according to the latest advancements in TSAD. It simulates real operational scenarios, i.e., **5 different mission phases** and real-time monitoring.

3. **Baseline results for 8 TSAD algorithms**, filtered from the 71 available in the TimeEval framework [23] and adapted to be feasible for real-world time series data.

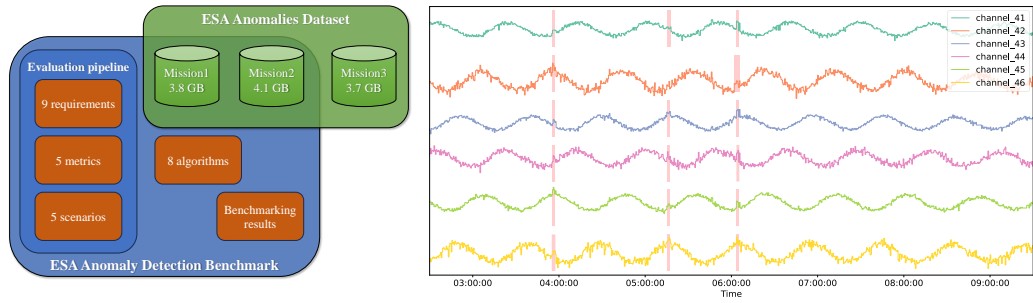

Figure 1: **Content of the ESA-ADB.** Left: Main elements of the proposed benchmark. Right: Example annotated event id_155 from Mission1 (highlighted with light red boxes).

The main goal of ESA-ADB is to allow researchers and practitioners to design and thoroughly assess if an algorithm could be applied as a support for SOEs in real-world operational environments, taking into account all challenges of this complex time series data. To support that, we also launched a Kaggle competition based on a separate private test set (kaggle.com/competitions/esa-adb-challenge). ESA-ADB has been already downloaded more than 2,000 times and is actively used in several projects by ESA and its partners.

## 2 ESA Anomalies Dataset

The dataset is publicly available at (zenodo.org/records/15237121). The nomenclature (e.g., channel types, telecommands, and event categories) is explained in the first section of the Appendix.

### 2.1 Dataset collection and curation

Three missions (spacecraft) of different types (purposes, orbits, and launch dates) were selected by SOEs from the ESA portfolio based the presence of historical anomalies that are problematic to detect using existing out-of-limit approaches. The data selection was focused on collecting a large dataset with a possibly diverse spectrum of signals and anomalies as reported in the Anomaly Report Tracking System (artsops.esa.int) used at ESOC. Although each spacecraft collects thousands of telemetry signals (Appendix 2), our dataset includes only limited subsets of channels and telecommands, identified by SOEs as essential for anomaly investigation. This selection was necessary to keep the annotation effort and overall dataset size at a manageable level. The data was initially annotated using the OXI annotation tool (oxi.kplabs.pl) [24] and the annotations were iteratively refined with assistance of unsupervised and semi-supervised algorithms. For detailed description of the annotation process, see Appendix 2.3 and our previous related works [25, 26].

**Design choices.** Our dataset has several features distinguishing it from other related datasets (Appendix 2.6). It is intended to reflect the raw telemetry data accessible for SOEs, with all its pros and cons, volume, varying sampling rates, data gaps, and an overabundance of telecommands. It distinguishes events of different types, not only anomalies, i.e., rare nominal events, communication gaps, and invalid segments. Each channel is annotated independently, following the approach used in recent datasets such as SMD [27], CATS [22], and TELCO [28]. This allows for evaluating not only whether an anomaly is detected, but also which specific channels are correctly identified as affected. Furthermore, a single annotated anomaly can consist of multiple non-contiguous segments, separated by periods of nominal behavior. For example, a series of short attitude disturbances caused by the same underlying issue is treated as one event (see Figure 1). This design choice avoids unfairly penalizing models for detecting each segment as a separate anomaly in the benchmark. The dataset was consistently structured to facilitate its usage in ML-based pipelines (Appendix 2.5).

**Anonymization.** Some information, such as mission and channel names, timelines, or units of measured values, are anonymized to avoid disclosing sensitive information. The anonymization does not affect the data integrity and it was verified that algorithms produce the same results as for the original data (Appendix 2.4). It does prevent using physics-informed approaches or domain-specific knowledge to design algorithms (for example, to match telecommands and channels by names or to expect anomalies in specific times, e.g., during increased solar activity). However, it enforces the usage of universal data-driven approaches, instead of focusing on mission-specific intricacies.

### 2.2 Dataset content

The summary statistics of the dataset are presented in Table 1. The dataset contains 224 channels, 821 telecommands, and 1430 annotated events (including 157 anomalies) across 3 missions. Channels are categorized into target (monitored for anomalies) or non-target (auxiliary); and numerical (e.g., sensor measurements) or categorical (e.g., status flags and operating modes) ones. Channels originate from 6 common spacecraft subsystems and are clustered into groups of related channels (e.g., coming from similar sensors or showing similar characteristics). There are hundreds of different telecommands with millions of executions and they are grouped by SOEs according to the impact on the mission data (Appendix 2.3) – from 0 (low impact) to 3 (high impact).

Missions differ significantly in aspects such as the proportion of categorical channels, the number of telecommands, and the distribution of event categories. They also vary in terms of signal characteristics and specific challenges posed for TSAD algorithms (Appendix 2). However, they are all equally big (around 4GBs and 750M data points each) and the hundreds of annotated events constitute just a small fraction of the dataset ($< 2\%$), addressing the flaw of unrealistic anomaly density [20].

Each anomaly and rare nominal event is described by three attributes corresponding to its dimensionality (uni-/multivariate), locality (local/global), and length (point/subsequence) according to the adjusted nomenclature of anomaly types by Blázquez-García et al. [29]. Most annotations are categorized as multivariate global subsequences, but there is also a diverse set of other types of

Table 1: **Statistics of the ESA Anomalies Dataset.**

|  | **Mission1** | **Mission2** | **Mission3** | **All** |
|---|---|---|---|---|
| **Channels** | **76** | **100** | **48** | **224** |
| Target / Non-target | 58 / 18 | 47 / 53 | 24 / 24 | 129 / 95 |
| Numerical / Categorical | 76 / 0 | 90 / 10 | 4 / 44 | 170 / 54 |
| Channel groups | 18 | 29 | 12 | 59 |
| Subsystems | 4 | 5 | 3 | 6$^*$ |
| **Telecommands** | **698** | **123** | **0** | **821** |
| Priority 0/1/2/3 | 345 / 323 / 19 / 11 | 0 / 0 / 119 / 4 | 0 / 0 / 0 / 0 | 345 / 323 / 138 / 15 |
| Total executions | 1,594,722 | 1,918,002 | 0 | 3,512,724 |
| **Data points** | **774,856,895** | **776,734,364** | **744,530,898** | **2,296,122,157** |
| Duration (anonymized) | 14 years | 3.5 years | 8 years | 25.5 years |
| Compressed size [GB] | 3.8 | 4.1 | 3.7 | 11.6 |
| Annotated points [%] | 1.80 | 0.58 | 1.03 | 1.14 |
| **Annotated events** | **200** | **644** | **586** | **1,430** |
| Anomalies | 118 | 31 | 8 | 157 |
| Rare nominal events | 78 | 613 | 25 | 716 |
| Communication gaps | 4 | 0 | 397 | 401 |
| Invalid segments | 0 | 0 | 156 | 156 |
| Univariate / Multivariate | 32 / 164 | 9 / 635 | 8 / 25 | 49 / 824 |
| Global / Local | 113 / 83 | 585 / 59 | 28 / 5 | 726 / 147 |
| Point / Subsequence | 12 / 184 | 0 / 644 | 3 / 30 | 15 / 858 |
| **Distinct event classes** | **22** | **32** | **6** | **60** |

*There are 3 matching subsystems between all missions.

anomalies (Appendix 3.1), including some especially challenging ones (Appendix 2.2). Additionally, events of similar characteristics are grouped into classes by SOEs, so it is easier to analyze results and design anomaly classifiers. The distributions of classes of events across missions' timelines are presented in Figure 2.

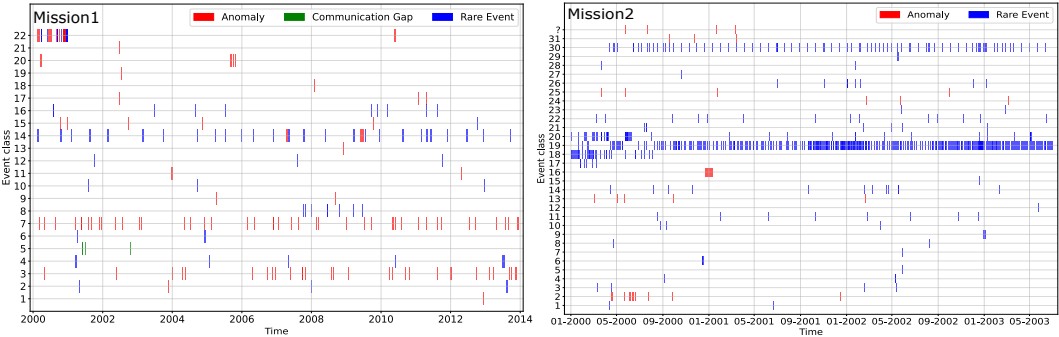

Figure 2: **Distributions of different classes and categories of events across timelines of Mission1 (left) and Mission2 (right).** The bar width corresponds to the event length, but for better visualization, the minimum width was limited to 10 and 2.5 days for Mission1 and Mission2, respectively. The question mark represents anomalies of unknown class.

## 2.3 Dataset split for the benchmark

Mission3 was excluded from the benchmark because of the small number and triviality of anomalies (according to Definition 1 from Wu & Keogh [20]), and many communication gaps and invalid segments (Table 1). Remaining missions are split in half: the first half is used for training, and the second half for testing. This results in 84 months of training data for Mission1 and 21 months for Mission2. Within each training set, the last 3 months are reserved for validation. This validation window was chosen in agreement between ML experts and SOEs, as it is sufficiently long to assess algorithm performance under recent environmental conditions. Crucially, this temporal split ensures that no future information leaks into the training process, preserving the integrity of the evaluation. Anomalies appear in all sets, including training and validation ones. The default 50/50 split reflects the later (mature) phases of missions, where a substantial amount of telemetry data is already available

for training. However, it is also important to deploy anomaly detection systems early in the mission lifecycle. To address this, additional scenarios with shorter training periods are explored in Appendix 4.4. These scenarios aim to evaluate how well the algorithms perform under limited data conditions, assess their robustness to evolving mission environments, and determine the earliest point in a mission when reliable detectors can be trained.

**Lightweight subsets of channels.** In the default setting of ESA-ADB, all channels and the highest priority telecommands are used as input, and all target channels are used as output from algorithms. However, anomaly detection in tens or hundreds of channels is a very challenging task which takes a lot of computing power, so for initial experiments, familiarization, simpler models, and potential on-board applications, there are lightweight subsets of channels proposed in ESA-ADB. These are channels 41-46 for Mission1 and channels 18-28 for Mission2. This selection is subjective, but our main goal was to provide channels that are challenging for algorithms, interesting for SOEs, relatively easy to visualize and analyze manually, and not strongly dependent on other channels or subsystems. Selected channels from these subsets are presented in Figure 1 and Appendix 4.1.

# 3 ESA Anomaly Detection Benchmark

The objective of the benchmark is to validate the performance of widely used TSAD algorithms on the ESA-AD dataset using the proposed evaluation procedures. The code is publicly available at github.com/kplabs-pl/ESA-ADB to ensure full reproducibility of the benchmark.

## 3.1 Algorithms selection

There are several recent comprehensive reviews of TSAD approaches that list hundreds of ML algorithms [17, 19, 30–32]. Algorithms for our benchmark have been preselected based on the work by Schmidl et al. [17] and its corresponding TimeEval framework [23] because of the largest number of implemented algorithms (more than 70). This framework also includes the most widely used deep learning algorithm for anomaly detection in spacecraft telemetry – Telemanom by NASA [7] – which we use as the primary baseline in our benchmark.

**Functional requirements.** To support the selection of algorithms, nine functional requirements (R1-R9) for anomaly detection algorithms in real-world space operations have been formulated by our team (Table 2) and evaluated against the capabilities of multivariate algorithms within the TimeEval framework. Based no the detailed analysis in Appendix 3.4.5, it turned out that no algorithm meets all the requirements. The widely used algorithms have problems especially with learning from known anomalies (R4), including auxiliary variables (R6), test-time learning (R7), irregular time series data (R8), and most importantly, handling large volumes of data without memory errors and time-outs (R9). Thus, additional work is necessary to adapt algorithms to our real-world use case.

**Algorithms adaptation and final selection.** To establish a simple baseline, five unsupervised algorithms based on traditional ML have been used without any special adaptation: Histogram-based Outlier Score (HBOS) [33], k-nearest neighbors (KNN) [34], principal components classifiers (PCC) [35], and isolation forest (iForest) [36] with its windowed version. The more advanced semi-supervised LSTM-based Telemanom algorithm [7] has been significantly adapted (Appendix 3.4.1) to meet requirements R4, R6, and R9. The adapted version, called Telemanom-ESA, comes with pruned and non-pruned modes. Additionally, two methods have been added to the framework: a simple algorithm that mimics the classic out-of-limits approach by detecting values being more than $N$ standard deviations away from the mean (Global STD$N$, described in Appendix 3.4.2) and the Dilated Convolutional Variational AutoEncoder (DC-VAE) [28] with a series of adaptations (DC-VAE-ESA in Appendix 3.4.3), including a thresholding based on $N$ confidence intervals generated from VAE.

## 3.2 Real-world evaluation of unsupervised algorithms

Default evaluation procedures for unsupervised outlier detection algorithms in TimeEval (and many other frameworks, e.g., the popular PyOD [37]) do not include any separate training step on a training set. The algorithms are run on all available data at once to look for outliers. This setup is unrealistic for real-world anomaly monitoring, where future data is not available at training time, and it introduces information leakage that gives unsupervised algorithms an unfair advantage in benchmarks. To address this, we modified the TimeEval framework so that each unsupervised

Table 2: **Requirements for anomaly detection algorithms in real-world spacecraft telemetry.**

| ID | Priority | Requirement | Description |
|---|---|---|---|
| R1 | must | provide binary responses | Algorithms must define a clear threshold for triggering alarms. SOEs cannot rely solely on abstract anomaly scores. |
| R2 | must | be able to model dependencies between channels | Algorithms are particularly needed to detect complex anomalies that only become apparent when analyzing multiple channels simultaneously. |
| R3 | must | allow for online streaming anomaly detection | Algorithms in real-world settings must detect anomalies continuously without using future samples to generate predictions. |
| R4 | should | learn from known anomalies in the training set | Algorithms should be able to leverage information about historical anomalies to effectively detect similar ones during inference. |
| R5 | should | provide a list of affected channels | With thousands of channels in real missions, identifying affected channels would save SOEs significant effort and improve trust in the algorithm. |
| R6 | should | support auxiliary variables in the input | Real-world data exists within a broader system of external variables (i.e., non-target channels) and control signals (i.e., telecommands). Algorithms should leverage this context to make better-informed decisions. |
| R7 | should | allow for test-time learning from human feedback | In real-world settings, algorithms should be able to adapt based on feedback from domain experts – for example, to stop raising alarms for rare but known nominal events. |
| R8 | should | natively handle irregular time series data | Varying sampling frequencies and data gaps are common in real-world time series. Standard resampling and interpolation methods make algorithms unaware of this fact and may lead to many incorrect detections. |
| R9 | should | be able to run on a single high-end PC with a modern GPU (Appendix 4.5) | ML algorithms are often run on a dedicated PC within the mission control room to minimize reliance on external systems and ensure data privacy, security, and integrity. |

algorithm is first initialized only on the training set – this includes calculating contamination levels, setting thresholds, and computing standardization parameters – and then applied to the test set.

## 3.3 Preprocessing

Our dataset contains raw telemetry in which channels have different, irregular sampling rates. There are no algorithms in the TimeEval framework that can handle such data without any preprocessing. Additionally, there are many different types of channels, so a consistent preprocessing is needed to run and compare all algorithms.

**Resampling.** Propagating the last known value (forward fill) is a widely recommended interpolation method for real-world sensor data [24, 38]. It is especially well suited for binary or quantized signals – such as status flags or measurements from analog-to-digital converters – because, unlike linear or Fourier-based interpolation, it avoids generating artificial or invalid intermediate values. Crucially, this method only uses past data, making it appropriate for real-time streaming applications where future samples are not yet available. Hence, this method was used for resampling (Appendix 3.3).

**Encoding telecommands.** Telecommands in the original data are represented as lists of timestamps indicating when they were executed on board the spacecraft. For use in ESA-ADB as input channels to algorithms, telecommands are encoded as binary impulse signals – single-sample spikes aligned with the target resampling resolution.

**Standardization.** This step is essential for certain algorithms, such as KNN, and can also improve the performance of neural networks [39]. In our preprocessing, each channel is standardized independently to have zero mean and unit standard deviation, based on the nominal (non-anomalous) points in the training set after resampling. Exceptions are constant and binary channels (e.g., encoded telecommands) that are just normalized to <0, 1> range. Monotonic channels (e.g., counters and cumulative readings) are differentiated and categorical channels (e.g., status flags) are enumerated before standardization (Appendix 3.3).

## 3.4 Metrics and hierarchical evaluation

The selection of metrics and evaluation pipeline is a crucial step in establishing a reliable benchmark. Despite many years of research in the domain, there is no consensus on a reliable and unified set of TSAD metrics. Many recent advances criticize popular sample-wise and point-adjust protocols

for being overoptimistic, and propose better alternatives [18, 19, 21, 30, 31, 40–47]. Besides, there are several constraints on the selection of metrics arising directly from the functional requirements in Table 2. Metrics should operate on binary detections (R1), handle ground truth with irregular sampling (R8), and have reasonable computational complexity to handle large datasets (R9). Detailed analysis of all recent metrics in the context of our benchmark is presented in Appendix 3.2.1.

First, SOEs identified and prioritized five most important aspects of anomaly detection in real-world mission operations. They are listed in Table 3 together with the proposed metrics to assess them. Importantly, each metric is designed to focus solely on a single specific aspect, in the maximum isolation from the other factors. There are several reasons for this: 1) to improve the interpretability of results by avoiding complex metrics combining multiple aspects at once, 2) to allow researchers from different domains to easily reorder or discard priorities, and 3) to enable the hierarchical evaluation approach in which algorithms are compared using one aspect at a time, from the highest to the lowest priority. This kind of evaluation has three important practical advantages: 1) it puts a strong emphasis on the priorities suggested by SOEs, 2) there is no need to select relative weights of specific aspects, and 3) it saves computational time by calculating only the necessary metrics.

Table 3: **Priority aspects and proposed metrics for assessing algorithms in ESA-ADB.**

| Group | Aspect with priority level and brief description | Proposed metric |
|---|---|---|
| Primary | **1a.** *No false alarms* – minimize the number of false detections | Corrected event-wise $F_{0.5}$-score |
| | **1b.** *Anomaly existence* – maximize the number of correctly detected anomalies | |
| | **2a.** *Subsystems identification* – find a list of affected subsystems | Subsystem-aware $F_{0.5}$-score |
| | **2b.** *Channels identification* – find a list of affected channels | Channel-aware $F_{0.5}$-score |
| Secondary | **3.** *Exactly one detection per anomaly* – avoid multiple detections for the same annotated segment | Event-wise alarming precision |
| | **4.** *Detection timing* – determine the anomaly start time as precisely as possible | Anomaly detection timing quality curve (ADTQC) |
| | **5.** *Anomaly range and proximity* – find the exact duration of the anomaly and promote detections in close proximity to the ground truth | Modified affiliation-based $F_{0.5}$-score |

The highest priority aspect relates to the proper identification of anomalous events with a strong emphasis on avoiding false alarms. This is because false positives are costly to resolve and deter operators from using the system. A high false positive rate is one of the main obstacles to the wider adoption of anomaly detection algorithms in space operations [7]. In the main text, we focus only on this most important aspect and the corresponding corrected event-wise metric, but other aspects are thoroughly described and assessed in Appendix 3.2.3.

**Corrected event-wise $F_{0.5}$-score.** The event-wise scoring used for spacecraft telemetry by Hundman et al. [7] is better than sample-wise approach in real-world scenarios as it 1) weighs all anomalies equally (not by their length) and 2) does not focus on the level of overlap between detections and the ground truth (in practice, it is enough to give an approximate location of the anomaly to human operators). However, the simple event-wise precision has one serious flaw – an algorithm that detects anomalies in every sample would have a perfect score (example in Appendix 3.2). To mitigate this, we use the correction proposed by Sehili and Zhang [18] that penalizes sample-wise (time-wise) false positives in the computation of the event-wise precision $\text{Pr}_e$ (Equation 1):

$$\text{Pr}_e = \frac{\text{TP}_e}{\text{TP}_e + \text{FP}_e} \cdot \left(1 - \frac{\text{FP}_t}{\text{N}_t}\right) \quad , \tag{1}$$

where $\text{TP}_e$ is the number of event-wise true positives, $\text{FP}_e$ is the number of event-wise false positives, $\text{FP}_t$ is the total duration of false positives, and $\text{N}_t$ is the total duration of nominal signal. Based on that, the corrected event-wise $F_\beta$-score is defined by Equation 2:

$$\text{F}_{\beta_e} = \frac{(1 + \beta^2) \cdot \text{Pr}_e \cdot \text{Rec}_e}{\beta^2 \cdot \text{Pr}_e + \text{Rec}_e} \quad , \quad \text{Rec}_e = \frac{\text{TP}_e}{\text{TP}_e + \text{FN}_e} \quad , \tag{2}$$

where $\text{Rec}_e$ is the event-wise recall and $\text{FN}_e$ is the number of event-wise false negatives. We use $\beta$ of 0.5 following Hundman et al. [7] to additionally penalize false detections.

## 3.5 Results and discussion

The goal of this benchmark is to provide a solid foundation for future research, rather than to identify the single best algorithm for real-world time series. Therefore, the experiments do not involve extensive hyperparameter tuning, which would be computationally prohibitive given the dataset size. Instead, we use default settings recommended by the original authors of each algorithm, with minor adjustments to match the specific characteristics of our dataset (Appendix 3.4.6). This approach is intentional – it reflects typical TSAD practices and is meant to encourage the research community to build upon these results.

There are no algorithms in the TimeEval framework that can explicitly distinguish between anomalies and rare nominal events, so the results are presented for both types combined. However, separate results considering only anomalies are available in Appendix 4.2. The corrected event-wise scores for Missions 1 and 2 are presented in Table 4. Results for lower priority metrics are available in Appendix 4. Scores are rounded to 3 significant digits to account for the inherent uncertainty of annotations in real-world data. The processing times of the algorithms are given in Appendix 4.6.

Table 4: **Corrected event-wise scores for detection of anomalies and rare nominal events in lightweight and full sets of channels for Mission1 and Mission2.** Boldfaced results indicate the best values among all algorithms. OOM – out-of-memory.

| Model | Mission1 | | | Mission2 | | |
|---|---|---|---|---|---|---|
| | $\mathbf{Pr}_e$ | $\mathbf{Rec}_e$ | $\mathbf{F}_{0.5_e}$ | $\mathbf{Pr}_e$ | $\mathbf{Rec}_e$ | $\mathbf{F}_{0.5_e}$ |
| *Trained and tested on **lightweight subsets of channels*** | | | | | | |
| PCC | <0.001 | 0.554 | <0.001 | 0.029 | **1.000** | 0.036 |
| HBOS | <0.001 | 0.585 | <0.001 | 0.055 | 0.911 | 0.068 |
| iForest | <0.001 | 0.585 | <0.001 | 0.557 | 0.974 | 0.609 |
| Windowed iForest | <0.001 | 0.738 | <0.001 | 0.951 | 0.940 | **0.949** |
| KNN | <0.001 | 0.754 | <0.001 | 0.000 | **1.000** | 0.001 |
| Global STD3 | 0.001 | 0.431 | 0.001 | 0.006 | **1.000** | 0.007 |
| Global STD5 | 0.288 | 0.169 | 0.253 | 0.061 | **1.000** | 0.075 |
| DC-VAE-ESA STD3 | 0.002 | 0.554 | 0.003 | 0.003 | **1.000** | 0.003 |
| DC-VAE-ESA STD5 | 0.063 | 0.338 | 0.075 | 0.064 | **1.000** | 0.079 |
| Telemanom-ESA | 0.148 | **0.894** | 0.178 | 0.188 | 0.986 | 0.224 |
| Telemanom-ESA Pruned | **0.999** | 0.424 | **0.786** | **0.978** | 0.540 | 0.842 |
| *Trained and tested on **full set of channels*** | | | | | | |
| PCC | <0.001 | 0.870 | <0.001 | 0.082 | 0.983 | 0.100 |
| HBOS | <0.001 | 0.957 | <0.001 | 0.016 | 0.820 | 0.020 |
| iForest | <0.001 | **0.967** | <0.001 | 0.022 | 0.903 | 0.027 |
| Windowed iForest | OOM | OOM | OOM | 0.034 | 0.746 | 0.042 |
| KNN | OOM | OOM | OOM | OOM | OOM | OOM |
| Global STD3 | <0.001 | 0.848 | <0.001 | 0.014 | **0.997** | 0.018 |
| Global STD5 | 0.002 | 0.761 | 0.003 | **0.203** | 0.972 | **0.241** |
| DC-VAE-ESA STD3 | <0.001 | 0.924 | <0.001 | 0.002 | **0.997** | 0.002 |
| DC-VAE-ESA STD5 | 0.005 | 0.804 | 0.007 | 0.008 | 0.904 | 0.011 |
| Telemanom-ESA | 0.007 | 0.946 | 0.008 | 0.052 | 0.992 | 0.064 |
| Telemanom-ESA Pruned | **0.050** | 0.870 | **0.061** | 0.058 | 0.964 | 0.071 |

**Mission1.** The pruned Telemanom-ESA has achieved the highest corrected event-wise $F_{0.5_e}$ and the lowest number of redundant alarms in both channel sets and all mission phases (Appendix 4.4). The huge advantage of Telemanom in terms of these metrics is its dynamic thresholding scheme and additional pruning. This highlights the importance of proper thresholding and postprocessing methods in real-world settings. On the other hand, pruning significantly decreases subsystem-/channel-aware, detection timing (ADTQC), and affiliation-based scores, so the anomalies may be harder to identify. Unsupervised algorithms perform very poorly for Mission1 in terms of event-wise scores. DC-VAE-ESA and GlobalSTD are just slightly better which is especially disappointing for the former deep learning method. The main problem of these algorithms is a massive number of false detections caused by the noise and varying sampling rates in the data, as visible in the examples in Appendix 4.1. However, DC-VAE-ESA has the best timing and affiliation-based scores – higher than Telemanom-ESA. This suggests that more advanced thresholding or postprocessing would significantly improve the event-wise scores of DC-VAE.

**Mission2.** Surprisingly, the simple windowed iForest and GlobalSTD5 algorithms turned out to be the best algorithms for the lightweight and full sets, respectively. Overall, unsupervised algorithms perform relatively well for Mission2, sometimes better than the deep learning-based ones. It supports

the claim to always consider simple algorithms as a baseline [20, 48]. Windowed iForest achieved a very high corrected event-wise $F_{0.5_e}$ (0.949), ADTQC (0.985), and affiliation-based $F_{0.5_e}$ (0.959) scores. The main reason is the relative triviality of the lightweight subset of Mission2 which contains mainly rare nominal events characterized by significant sudden changes in the signal (Appendix 2). However, the full set is much more challenging as reflected by much lower corrected event-wise scores. Moreover, metrics for anomalies alone (Appendix 4.2) show that no algorithm was able to accurately identify all 9 actual anomalies in the overabundance of rare nominal events. This is one of the main practical challenges in many missions. Mission2 is also particularly problematic for Telemanom-ESA because of a lack of clear periodicity and many commanded events that are impossible to forecast.

**Full sets vs. lightweight subsets.** In most cases, the results for full sets of channels are much worse than for lightweight subsets. While the two are not directly comparable (since the lightweight test sets contain fewer annotated events), Appendix 4.3 includes a direct comparison that supports this observation. This is one of the main challenges of high-dimensional real-world data – the more target channels there are, the higher the chance of false detections is. Additionally, due to the strong interconnections between channels, false detections frequently seep into many irrelevant channels.

# 4 Conclusions

ESA-ADB is a departure point for further development of better algorithms for anomaly detection in real-world time series (e.g., spacecraft telemetry). It was designed in close collaboration between ML and domain experts to fulfill the need for a reliable benchmark for both communities. Our goal was to ensure that improving the results of ESA-ADB does not just create an *illusion of progress* but solves real-world challenges in the TSAD domain – Appendix 2.6 gives a summary of how ESA-ADB addresses common flaws listed by Wu and Keogh [20]. The requirements analysis and results show that our dataset poses a significant challenge for popular TSAD algorithms, and many changes had to be applied in the TimeEval framework [23], training procedures, and algorithms to make them applicable to real-world data. While the results of Telemanom-ESA on subsets of channels may appear promising, the approach is highly parameterized, and the chosen thresholds may not generalize well to other missions. More importantly, the main challenge lies in scaling these algorithms to the full set of channels in our dataset – and to thousands of channels in real-world operations.

**Limitations and future work.** ESA-ADB has several limitations that we were not able to address in the scope of this study. Despite our best efforts, labeling inaccuracies are inevitable in such volumes of real-world data, so we are open to requests for corrections and plan to release updated versions of the dataset. Additionally, anonymization of physical units and timelines may constrain certain use cases. The dataset is still just a small fragment of real-world telemetry, but its complexity already poses a high entry barrier and requires some effort to fully understand. Due to the computational, functional (Table 2), and framework-related constraints, the current benchmark includes a limited range of algorithms and does not involve extensive hyperparameter tuning. As such, ESA-ADB should be viewed not as a comprehensive benchmark of TSAD methods, but rather as a solid baseline for future research on real-world time series. Promising directions for extending this work include adapting Matrix Profile methods [49] and transformer-based models with positional time encoding [50–52]. Beyond TSAD, the dataset also holds potential for research in time series forecasting, telemetry data compression, continual learning, and foundation models.

# 5 Acknowledgments

This work was financially supported by the European Space Agency under the contract number 4000137682/22/D/SR. Authors thank all employees of ESOC involved in the project. Authors are grateful to Alicja Musiał, Szymon Rogoziński, and Dawid Lazaj from KP Labs for their valuable insights into the evaluation process.

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

## A    Technical Appendices and Supplementary Material

The Technical Appendix is available as a separate PDF.

