# European Space Agency Dataset and Benchmark for Anomaly Detection in Real-World Time Series

Krzysztof Kotowski, Christoph Haskamp, Jacek Andrzejewski, Bogdan Ruszczak, Jakub Nalepa, Daniel Lakey, Peter Collins, Jose Martínez-Heras, and Gabriele De Canio

## Technical Appendix

1.  DEFINITIONS

*1.1. Channel vs parameter*

Satellite telemetry consists of multiple time series that are called *parameters* by SOEs. This name is very problematic from the ML point of view because it collides with its fundamental nomenclature in which the *parameter* already has a couple of different meanings:

- a *parameter* of the model that is updated during the training, i.e., a single weight of the neural network;
- a *parameter* (or hyperparameter) of the algorithm which controls its behavior;
- a *parameter* of a statistical test (e.g., mean or variance of the estimated Gaussian distribution).

European Space Agency Dataset and Benchmark for Anomaly Detection in Real-World Time Series

Hence, the *parameter* was replaced with the *channel* for purposes of ESA-ADB to avoid potential nomenclature collisions. *Channels* represent measurements from different sensors, status flags, and payload-related information. Each channel contains a list of samples defined by pairs of timestamps and signal values.

### 1.2. Subsystems

Satellites are typically composed of multiple specialized parts (subsystems) including propulsion, electrical power, thermal control, attitude and orbit control, communication, and data handling subsystems. There are also unique satellite-specific subsystems in some missions. A subsystem gathers all components (and channels) responsible for a specific function.

### 1.3. Telecommands

Telecommands (TCs) are sent from the Earth to the satellite in order to control different aspects of its operation. There are hundreds or thousands of different TCs for each mission with millions of total executions, affecting different subsystems and specific components. Many different TCs are frequently executed simultaneously or in series to perform specific instructions. They may affect the observed telemetry in various ways, from no visible changes to strong disruptions. In our dataset, each TC is a binary signal with values of 1 in the exact timestamps of TC's executions on-board the satellite. TCs are not expected to contain any anomalies and even if they were, anomalies (e.g., missing TCs) would be impossible to identify automatically without additional expert knowledge and information about mission plans. Thus, they are not monitored nor annotated for anomalies.

### 1.4. Target and non-target channels

Not every channel can be a target for anomaly detection benchmarking. Like telecommands, some channels are not expected to contain any anomalies, and it would be impossible to annotate them without additional external data anyway. Examples include status flags, counters, and metadata, such as location coordinates. They often contain important information in the context of anomaly detection but are not monitored nor annotated for anomalies. They may contain outliers that are, however, irrelevant (or nominal) for SOEs. They are called *non-target* channels in ESA-ADB. This aspect is usually not considered in existing multivariate anomaly detection datasets and benchmarks. The selection of *target* and *non-target* channels is somewhat subjective and it may turn out that some algorithms would be able to properly handle some *non-target* channels by discovering some unknown relationships in the data. However, the metrics in ESA-ADB are calculated only for *target* channels. *Non-target* channels may and should be used as input features for algorithms.

### 1.5. Event class vs category vs type

Each annotated event can be assigned to a different class, category, and type:
- *event classes* relate to main causes of events and their specific variations (subclasses) as identified by SOEs. For example, attitude disturbances (with subclasses depending on the specific cause), resets, power drops, latch-ups, solar flares, etc.;
- *event categories* relate to the categorization of events from the operational point of view, i.e., anomalies, rare nominal events, communication gaps, and invalid segments, as described in the next section;
- *event types* relate to the signal characteristics of events according to the taxonomy introduced in Appendix Section 1.7.

Note that each feature is independent of others, that is, events of the same class can have different categories and types, e.g., resets caused by telecommands are categorized as rare nominal events, but unexpected non-commanded resets are categorized as anomalies.

### 1.6. Event categories

For the purposes of our project, 4 categories of events are introduced: anomalies, rare nominal events, communication gaps, and invalid segments. They are defined in Table 5. The main reason was to distinguish atypical changes in the telemetry that should not be alarmed to operators (rare nominal events, communication gaps, and invalid segments) from unexpected ones that should be alarmed (anomalies). Rare nominal events are not anomalies from the operators' point of view and they are usually not reported in anomaly tracking systems. Eventually, they are recorded in the mission log as special operations. For some missions, i.e., Mission2, there is a significant number of such operations causing (not so) rare events. Hence, the ideal algorithm should not alarm for rare nominal events, but it is usually impossible to distinguish between novel rare nominal events and anomalies without additional a priori expert knowledge. As agreed with SOEs, it would be acceptable if an anomaly detection system shows a false alarm for the first occurrence of the specific rare event, but it should not alarm for any subsequent occurrences of similar rare events. In machine learning, we can define that problem as active one-shot learning. To enable evaluation in such a scenario using ESA-AD, it is necessary to distinguish rare events from anomalies in ground truth annotations. Besides, such a division allows us to calculate separate performance metrics for rare events and "real" anomalies. It also helps to interpret the results in case of false negative or false positive detections for rare events.

### 1.7. Event types

To the best of our knowledge, the taxonomy by Blázquez-García et al. [1] is the only one in the literature that comprehensively defines multivariate anomaly types, and our definitions are built based on this foundation. It divides event types into point and subsequence ones, where point events are defined as single outlying data points. However, this definition does not take into

European Space Agency Dataset and Benchmark for Anomaly Detection in Real-World Time Series
account varying sampling rates for which the length of "a single data point" may differ in time. Thus, for our purposes, multi-instance point anomalies are allowed if they are relatively short fragments of the signal that resemble points or peaks (i.e., up to 3 samples) when inspected using a typical sampling frequency for the channel. Both point and subsequence anomalies may be univariate or multivariate depending on whether they affect one or more channels. Anomalies can additionally be divided into global and local (contextual) ones, similarly as proposed in behavior-driven taxonomy by Lai et al. [2]. To make the original definitions more specific in our taxonomy, the global subsequence anomaly is defined as a subsequence of anomalous values in which at least one instance can be treated as a global point anomaly.

In the proposed taxonomy, each anomaly type can be described by three attributes: *dimensionality* (uni-/multi-variate), *locality* (local/global), and *length* (point/subsequence), as presented in Fig. 11. These attributes can be automatically inferred from per-channel annotations:

1. **Dimensionality** can be inferred by counting the number of channels affected by an anomaly. One affected channel makes it *univariate* and more affected channels make it *multivariate*.
2. To infer **locality**, we calculate the minimum and maximum values of all nominal samples in the dataset for each channel. If any sample of an annotated event lays out of <min, max> range for any channel, we mark it as *global*, otherwise it is *local*. This approach is a bit simplistic taking into account severe distribution shifts and different nominal levels of the signal in some missions, but it should be enough to identify *global* anomalies which could be detected with an out-of-distribution approach from more challenging *local* anomalies.
3. In terms of **length**, considering non-uniform sampling rates and the differences between mission and channels, it is hard to give a strict definition of a *point* anomaly. Our proposition is to make it dependent on the dominant sampling frequency for each mission (0.033 Hz for Mission1, 0.056 Hz for Mission2 and 0.065 Hz for Mission3). A point anomaly is defined as a sequence of up to 3 samples after signal resampling to the dominant sampling frequency. Importantly, some anomalies are fragmented into several non-overlapping annotated regions. In this case, we treat each region separately, so even if an anomaly contains several regions it can be a point anomaly if all of these regions are categorized as point anomalies.

Such automatically inferred attributes for every anomaly and rare event are given in anomaly_types.csv for each mission, taking into account annotations for all channels. However, when working with subsets of channels, only the specific subset of channels should be considered to infer anomaly types. For this purpose, the script infer_anomaly_types.py is available in the code repository. The attributes are not inferred for communication gaps and invalid fragments.

TABLE 5
DEFINITIONS OF EVENT CATEGORIES

| Event category | Definition | Typical examples | Alarming |
|---|---|---|---|
| *Anomaly* | Atypical, rare, unplanned, and unwanted change in the telemetry. | Micrometeorite impacts, solar flares, hardware or software failures, latch-ups, unexpected responses to telecommands | Every occurrence should be alarmed. |
| *Rare nominal event* | Atypical and rare but expected or planned change in the telemetry. It can be triggered by known telecommands (commanded rare event) or by any other non-commanded special event in the mission timeline. | *Commanded*: maneuvers, resets, calibrations, switching devices on/off 

 *Non-commanded*: planned autonomous operations, eclipses, lunar transitions | Only the first occurrence of a rare nominal event from each class may be alarmed. Subsequent occurrences should not be alarmed. |
| *Communication gap* | Unusually long gap in the telemetry (missing data in some or all channels) not directly related to known anomalies. | Problems with the ground infrastructure, effects of resets | It should not be alarmed unless explicitly stated to do so. |
| *Invalid segment* | Fragment of telemetry data containing invalid or forbidden values not directly related to known anomalies. It is neither nominal nor anomalous. | Telemetry does not meet clearly defined validity rules of the mission. | It should not be alarmed unless explicitly stated to do so. |

## 2. ESA ANOMALIES DATASET

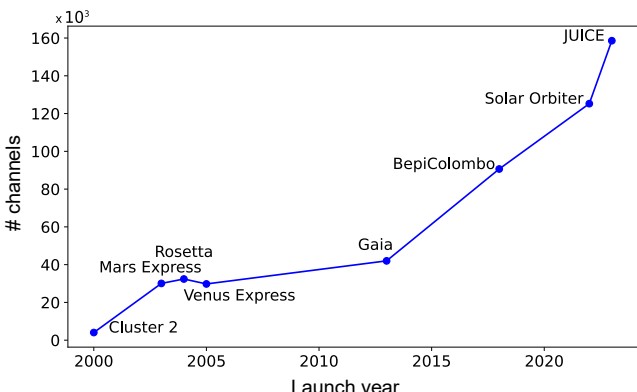

**Fig. 4**. Increasing complexity of selected ESA spacecrafts over time [3].

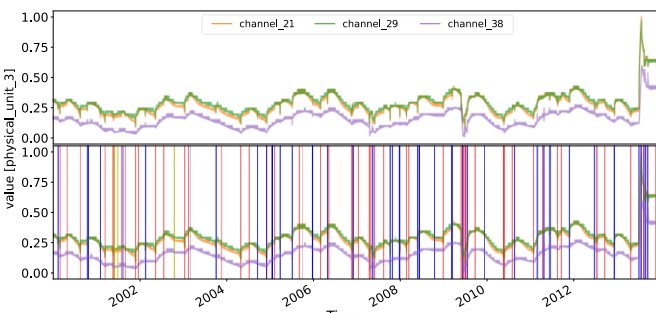

**Fig. 5.** Overview of 3 channels from group 4 in Mission1 without annotations (top panel) and annotated (bottom panel). Blue, yellow, and red vertical bars are rare nominal events, communication gaps, and anomalies, respectively.

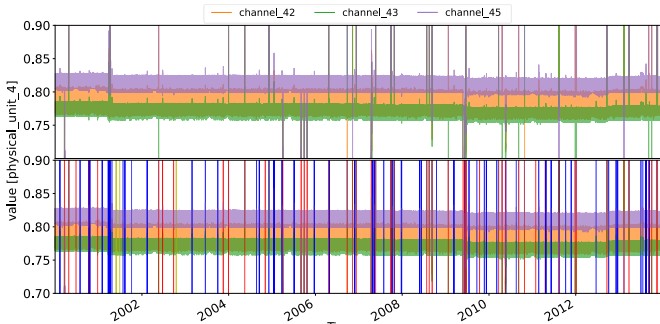

**Fig. 6**. Overview of 3 channels from group 8 in Mission1 without annotations (top panel) and annotated (bottom panel).. Blue, yellow, and red vertical bars are rare nominal events, communication gaps, and anomalies, respectively. The close-up of these channels is presented in Fig 2. in the main text.

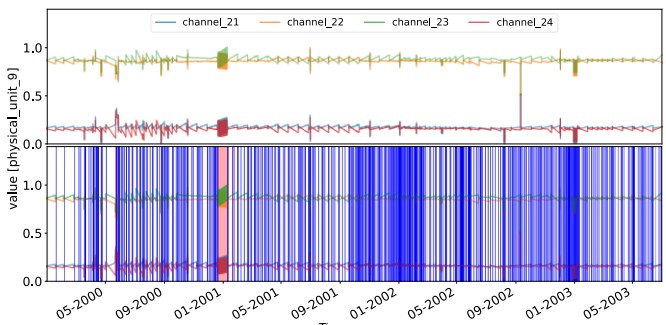

**Fig. 7.** Overview of 4 channels from group 2 in Mission2 without annotations (top panel) and annotated (bottom panel). Blue and red vertical bars are rare nominal events and anomalies, respectively.

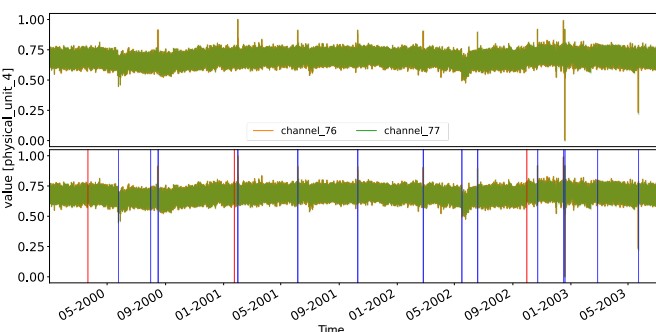

**Fig. 8.** Overview of 2 channels from group 31 in Mission2 without annotations (top panel) and annotated (bottom panel). Blue and red vertical bars are rare nominal events and anomalies, respectively. Note that channels have very similar values, so it is hard to distinguish them.

TABLE 6
MAIN CHALLENGES POSED FOR ALGORITHMS BY MISSIONS IN ESA-ADB

| Mission | Main challenges for algorithms |
|---|---|
| 1 | • Several anomalies are hard to spot (see Table 7)
• Several huge outliers (usually related to rare nominal events)
• Low signal-to-noise ratio in channels from group 8
• Monotonically non-decreasing signals in channels from group 2
• Last 18 months include a severe concept drift in channels from groups 4, 7, and 13
• There is a visible seasonality with a very long period length
• Overabundance of telecommands |
| 2 | • Several anomalies are hard to spot when looking at individual channels only (see Table 7)
• Overabundance of rare nominal events and a very small number of anomalies
• No obvious periodicity of the signal
• Monotonically non-decreasing signals in channels from group 20
• Many categorical and non-target channels |

European Space Agency Dataset and Benchmark for Anomaly Detection in Real-World Time Series

## 2.1. Mission3

Mission3 is a part of ESA-AD but is omitted in ESA-ADB. It was omitted mainly because of many communication gaps (see Fig. 9), invalid segments (corrupted data), long periods of constant signals, lack of telecommands, and a small number of anomalies that are trivial to detect according to Definition 1 of Wu & Keogh [4]. However, it may still be an interesting resource for practitioners in the domain as it is fully annotated and contains a unique set of challenges related to satellite telemetry.

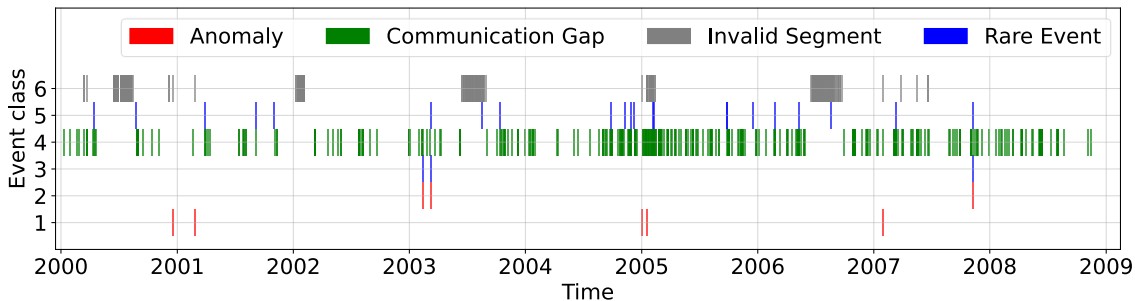

**Fig. 9.** Distributions of classes of annotated events across the timeline of Mission3.

## 2.2. Examples of challenging events to detect

As mentioned in the main text, the initial selection of missions was based on the presence of challenging anomalies according to SOEs. To support the analysis of results, a list of selected events of this type in test sets of ESA-ADB is provided in Table 7. It is not a complete list. It is limited to test sets and includes only subjectively selected examples among many others. Example detections by semi-supervised algorithms trained on full (suffix "-*Full*") and lightweight (suffix "-*Light*") subsets for selected events are presented in Appendix Section 4. as a series of figures referenced in Table 7.

Other interesting examples include events from classes 2, 14, 15, and 22 in Mission1 where similar changes in the same channel are sometimes categorized as anomalies and sometimes as rare nominal events, depending on the presence of TCs. A similar case for Mission2 is visualized in Fig. 10 for the non-commanded anomaly id_618 and the commanded rare event id_609. One of the important future works is to design algorithms that would be able to distinguish between such cases.

There are also some interesting nominal fragments related to atypical changes in sampling frequency in Mission1. There are 3 main examples of such behavior in the training set on days 2001-05-28, 2001-05-31, and 2001-06-27 where rapidly changing sampling rate causes small atypical "gaps" in data for channels 41-46. In the refinement process, it was observed that those gaps are detected as anomalies by many algorithms. However, we decided that they should not be annotated, because varying sampling rates are expected in satellite telemetry and these false detections are mainly since the selected algorithms are not aware of frequency changes.

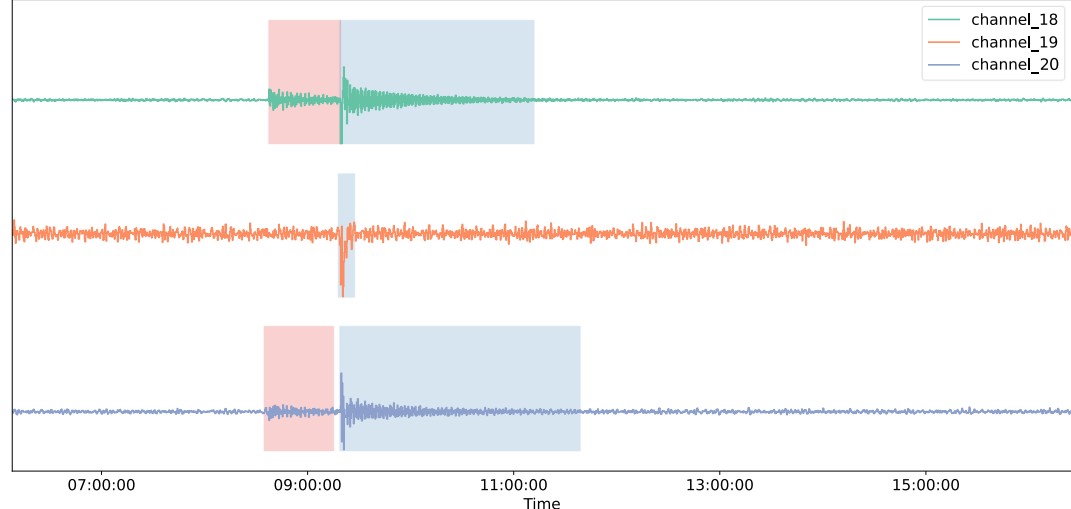

**Fig. 10**. Annotated anomaly id_618 (marked in red) directly preceding the commanded rare nominal event id_609 (marked in blue) in Mission2. The Y-axis is omitted, as channels are normalized and vertically shifted for improved visualization.

TABLE 7
LIST OF SELECTED CHALLENGING EVENTS ANNOTATED IN TEST SETS OF ESA-ADB

| Mission | Event category | Event ID | Start time (YYYY-MM-DD hh:mm:ss) | Duration | Reason for selection |
|---|---|---|---|---|---|
| Mission1 | Rare Event | id_24 | 2012-12-18 06:32:09 | 24h 15m | Hard to spot and not commanded. Not found by SOEs initially and added during the refinement process. It is related to a temporary change of nominal operational conditions. |
| | Rare Event | id_49 | 2011-10-08 07:08:39 | 10h 25m | Hard to spot, especially when looking at too narrow context. Caused by a rare TC. Fig. 17 |
| | Rare Event | id_51 | 2011-08-14 19:12:39 | 1h 19m | Hard to spot, especially when looking at too narrow context. Caused by a rare TC. Fig. 18 |
| | Rare Event | id_55 | 2011-04-23 08:19:39 | 0s (point) | Hard to spot in both lightweight and full sets. Caused by a unique execution of TC of priority 1. Overlaps with the rare event id_155. |
| | Anomaly | id_138 | 2009-10-13 06:39:17 | 1d 20h | Hard to spot using the lightweight subset of channels 41-46 only. Much easier to spot in channels 58-60. Fig. 19 |
| | Anomaly | id_153 | 2011-01-28 22:29:18 | 15h 14m | Hard to spot using the lightweight subset of channels 41-46 only. Much easier to spot in channels 64-66. Fig. 20 |
| | Rare Event | id_155 | 2011-04-21 22:15:52 | 11d | Hard to spot using the lightweight subset of channels 41-46 only. Easier to spot in multiple other channels, but hard to accurately identify the start time due to very slow changes. Main text Fig 2. and Fig. 21 |
| | Anomaly | id_157 | 2011-04-19 07:09:39 | 14h 35m | Hard to spot using the lightweight subset of channels 41-46 only. Much easier to spot in channels 64-66. |
| | Rare Event | id_159 | 2011-06-09 02:57:09 | 10d 21h | Hard to spot using the lightweight subset of channels 41-46 only. Very long annotations in other affected channels. Fig. 22 |
| Mission2 | Rare Event | id_466 | 2003-02-08 16:25:19 | 1h 10m | Small disturbance in 7 channels which may be easily overlooked, especially when using only the lightweight subset. |
| | Rare Event | id_591 | 2002-04-16 16:30:53 | 35m | Small disturbance in 7 channels which may be easily overlooked. |
| | Anomaly | id_631 | 2001-12-14 19:16:29 | 1h 18m | Small disturbance of unknown source in 7 channels which may be easily overlooked by operators. Fig. 23 |
| | Anomaly | id_644 | 2002-02-18 05:42:45 | 9h 48m | Divergence of channel 81 from channel 73 which can only be detected when looking at both channels in the proper context window. |

*2.3. Annotation details*

While annotating, a special focus was put on the precise identification of anomaly starting points for all channels. On the other hand, anomaly end times may be less accurate, because they are much harder to identify objectively, especially for long anomalies. Importantly, ARTS reports are intended for human use and are not well-suited for ML purposes. They usually include only approximate time ranges and a small fraction of affected channels. Moreover, well-known anomalies and rare nominal events are often not reported. Thus, the whole signal was carefully revisited by the ML team in search of any suspicious events. An initial list of subsystems, channels, and telecommands relevant for anomaly detection was proposed by SOEs, but was gradually extended during several iterations of the annotation refinement process in which overlooked anomalies were discovered by the ML team using different TSAD algorithms. During this process, channels were divided into target and non-target for anomaly detection. Non-target channels should only be used as additional information for the algorithms. They are not annotated and are not assessed in the benchmark. Examples include status flags, counters, and metadata such as location coordinates, where anomalies are not expected or it is not possible to check for anomalies without external data. Related channels measuring the same physical values and showing similar characteristics are organized into numbered groups, so it is easier to manage the dataset for ML purposes, e.g., to train group-specific models or to visualize results.

There are hundreds of different telecommands (TCs) in each mission. Some of them are critical for detecting annotated anomalies (i.e., when there is no reaction to the TC or the reaction is different than usual) or distinguishing anomalies from rare nominal events. However, it may be impractical to use them all in anomaly detection algorithms. Thus, 4 different priority levels for TCs were introduced as a suggestion about their potential usefulness for anomaly detection algorithms. The priorities from the least important to the most important are:

0. TCs not directly related to any subsystem included in the dataset.
1. TCs related to subsystems included in the dataset but not marked as potentially valuable for anomaly detection by SOEs.
2. TCs selected as potentially valuable for anomaly detection by SOEs.

3. A fraction of TCs of priority 2. assessed as valuable for anomaly detection by the ML team. The main rejection criteria were the scarcity of occurrences in the training data (less than 3) or no occurrences in the test data.

TCs of priority 3 are used as input for DC-VAE-ESA and Telemanom-ESA algorithms trained on full sets of channels. These priorities are only suggestions and ESA-ADB users are welcome to experiment with any combination of TCs.

*2.4. Anonymization details*

The anonymization had to be applied to conform with the ESA privacy policy and to avoid any accidental disclosure of sensitive mission-specific information or metadata. The anonymization process was carefully designed to maintain data integrity, so the results are independent of the anonymization. The following modifications were applied as a part of the anonymization process for each mission:

- *Renaming* of missions, subsystems, channels, telecommands, physical units, anomaly classes, and event types. They were consistently numbered according to their order of occurrence in files. Subsystems and physical units have consistent naming across missions, so it is possible to train cross-mission models.
- *Time scaling and shifting* of each mission. The timeline of every mission was scaled by a non-disclosed factor larger than 1 and shifted to start on 1st January 2000.
- *Normalizing values within channel groups to <0, 1> range.* Normalization per group was applied to preserve the same dependencies between similar channels before and after anonymization.

It was verified that the anonymization is fully reversible and there are no numerical errors related to the limited floating point resolution of values or timestamps. Additionally, it was verified that all deterministic algorithms in the benchmark produce the same results before anonymization.

*2.5. Dataset structure*

ESA-AD consists of three folders, one per each mission. Each folder has the same structure presented in Table 8. There is a subfolder named *channels* and an optional subfolder named *telecommands*. Both subfolders include serialised and compressed *Pickle* files (docs.python.org/3/library/pickle.html, protocol version 4.0, *zip* compression), one for each channel and telecommand. Each file contains a single *pandas DataFrame* (pandas.pydata.org/pandas-docs/stable/reference/api/pandas.DataFrame.html, *pandas* version 1.5.3) including an index with consecutive timestamps and a single column with the corresponding raw telemetry values. Annotations of all events are in a separate file called *labels.csv* placed directly in the mission folder. It contains rows that describe anomalous fragments using 4 columns: the anomaly identifier (ID), the name of the channel affected by the anomaly, the start time, and the end time of the anomalous segment. Start and end times are defined as closed ranges and they usually represent timestamps of actual points in the dataset. There may be multiple segments with the same ID and channel name, but their time ranges cannot overlap. Additional information on anomalies can be found in the *anomaly_types.csv* file. It describes each anomaly ID with its class, subclass, category, and type. The channels are described in the *channels.csv* file using the channel name, the associated subsystem, and the physical unit. The channel description also includes group numbers that indicate similar channels and the information if the channel is a target channel. If telecommands are included in the dataset their priority is described in the *telecommands.csv* file.

TABLE 8
FOLDER STRUCTURE OF ESA-AD

- `ESA-Mission/`
  - `channels/` folder including all channels of the mission
    - `*.zip` compressed *Pickle* files for each channel
  - `telecommands/` *(optional)* folder including all telecommands of the mission
    - `*.zip` compressed *Pickle* files for each telecommand
  - `labels.csv` annotations
  - `anomaly_types.csv` description of anomalies and rare nominal events
  - `channels.csv` description of channels
  - `events.csv` *(optional)* list of special operations and mission events
  - `telecommands.csv` *(optional)* description of telecommands

Some files are marked as optional, these files are not mandatory for the dataset or there might be missions not including these files. It should be possible to apply anomaly detection algorithms to the datasets not using the optional data, but it is expected that the optional data enhances the performance of algorithms when used. Mission2 includes an optional file *events.csv* which lists special operations and events with their start and end times according to the mission plan provided by SOEs. It was used to identify rare nominal events annotated in *labels.csv*, usually with slightly different start and end times due to different propagation times between channels.

### 2.6. Comparison to related public datasets

**Space-related dataset.** A quantitative comparison of the missions included in ESA-ADB and other public spacecraft-related telemetry datasets from the literature is presented in Table 9. There are 5 other public datasets of real-life spacecraft telemetry – 3 by NASA and 2 by ESA – and a single simulated one (CATS). The most popular ones are Soil Moisture Active Passive (SMAP) and Mars Science Laboratory (MSL) released by NASA [5]. According to the search for "SMAP" and "MSL" terms in Google Scholar since 2018, there are more than 200 documents that mention these datasets in the set of more than 500 citations of the source paper [5]. Besides a lot of criticism of these datasets in the recent literature [4], [6], [7], there is a common misconception about the number of channels included in these datasets. The data may come from 82 different physical channels in total, but there is a separate fragment for each channel without any synchronization with other channels, so they cannot be used effectively as a multivariate dataset. This is made clear in the description of the dataset in Table 9. NASA LASP WebTCAD [8] has tens of millions of points, but there are only 5 partially overlapping channels and no annotations of anomalies. ESA Mars Express Power Challenge [9] is popular in satellite telemetry forecasting, but does not contain anomalies annotations. The very recently published ESA OPSSAT-AD [16] is a toy dataset with 2213 univariate fragments of real OPS-SAT telemetry. It is contains anomaly annotations for whole fragments and is designed specifically for on-board applications.

**Non-space-related datasets.** There are also several related real-life datasets from outside the domain of satellite telemetry that are frequently used to benchmark multivariate TSAD algorithms. Notable examples include the Secure Water Treatment (SWaT) [10] and Water Distribution (WADI) [11] datasets which contain recordings from tens of channels from a real-world water treatment plant within several days. Server Machine Dataset (SMD) [12] including 5-week-long data from 38 parameters of 28 machines from 3 servers at a large internet company. The recent TELCO dataset [13] is worth noting due to related ideas of separate annotations for each channel, anomalies in training sets, and gradually increasing training set sizes. It contains 12 channels corresponding to real measurements collected over 7 months at an operation mobile internet service provider. To the best of our knowledge, the Exathlon benchmark [14], including real data traces from tens of repeated executions of streaming jobs with 2283 parameters (channels) on a Spark cluster over 2.5 months, is the only related dataset of volume comparable to ESA-AD, with more than 5 billion samples and 25 GB of data. However, it does not contain per-channel annotations and has been criticized for unrealistic anomaly density and positional bias [6].

Table 10 gives qualitative description of how ESA-ADB addresses main 4 flaws reported by Wu & Keogh [4].

TABLE 9
QUANTITATIVE COMPARISON OF ESA-AD AND OTHER PUBLIC SPACECRAFT DATASETS.

| Dataset name | Number of channels | Total volume | Number of annotated events |
|---|---|---|---|
| ESA-AD (Ours) | **Mission1**: 1 fragment with 76 channels and 698 commands

**Mission2**: 1 fragment with 100 channels and 123 commands | 1,551,591,259 samples

3,512,724 commands executions | 842 (1.17% of all samples) |
| NASA SMAP and MSL [5] | **SMAP**: 55 fragments with 1 channel and 24 commands

**MSL**: 27 fragments with 1 channel and 55 commands | 706,971 samples

410,030 commands executions | 105 (8.98% of all samples) |
| NASA LASP WebTCAD [8] | 1 fragment with 5 partially overlapping channels | 55,258,122 samples | not annotated |
| NASA Shuttle Valve Data (cs.fit.edu/~pkc/nasa/data) | **TEK**: 12 fragments with 1 channel
**VT1**: 27 fragments with 1 channel | 552'000 samples | 8 whole fragments |
| CATS [15] (simulated) | 1 fragment with 17 channels | 85,000,000 samples | 200 (2.15% of all samples) |
| ESA Mars Express Power Challenge [9] | **Train**: 3 fragments with 38 channels (including 5 metadata-related)

**Test**: 1 testing fragment with 5 metadata channels | 198,045,083 samples | not annotated for anomalies |
| ESA OPSSAT-AD [16] | 2123 univariate fragments from 9 different channels | 303,493 samples | 434 whole segments (20.44%) |

TABLE 10
A LIST OF FLAWS REPORTED BY WU & 2.6 [4] AND HOW THEY ARE ADDRESSED BY ESA-ADB.

| Flaw | How does ESA-ADB address it? |
|---|---|
| Triviality | • ESA-AD is large and contains a diverse set of anomaly types and concept drifts which hamper the usage of simple algorithms
• ESA-AD offers a selection of non-trivial anomalies, so they can be evaluated separately (Appendix Section 2.2)
• ESA-ADB includes a set of simple algorithms to verify the potential triviality of anomalies |
| Unrealistic anomaly density | • ESA-AD is large and the anomaly density in the dataset is below 2% of data points
• There are only dozens of anomalous events per year
• Series of separate annotated segments within a short region are usually assigned to the same event and are treated as such when computing metrics |
| Mislabelled ground truth | • While this flaw cannot be fully resolved in real-life datasets there were several iterations of the annotation refinement process aided by unsupervised and semi-supervised algorithms to identify potential mislabelling [17] |
| Run-to-failure bias | • Anomalies are scattered across long, failure-free, operational periods of acquired telemetry data from real satellite missions |

3. METHODS

*3.1. Anomaly types*

Fig. 11 gives an overview of the proposed taxonomy of anomaly types and Table 11 gives statistics of each anomaly types across all missions in ESA-AD.

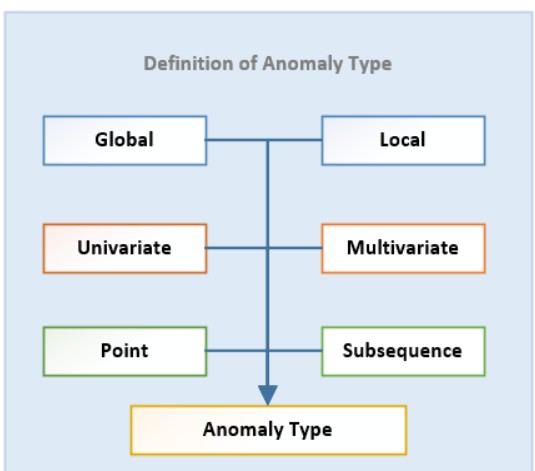

**Fig. 11**. Anomaly types considered in ESA-ADB.

TABLE 11

DISTRIBUTION OF 8 COMBINATIONS OF ANOMALY TYPES ACROSS MISSIONS

| Length | Locality | Dimensionality | Mission1 | Mission2 | Mission3 |
|---|---|---|---|---|---|
| Point | Global | Univariate | 0.00% | 0.00% | 9.09% |
| | | Multivariate | 5.10% | 0.00% | 0.00% |
| | Local | Univariate | 0.51% | 0.00% | 0.00% |
| | | Multivariate | 0.51% | 0.00% | 0.00% |
| Subsequence | Global | Univariate | 12.24% | 0.00% | 60.61% |
| | | Multivariate | 40.31% | 90.84% | 15.15% |
| | Local | Univariate | 3.57% | 1.40% | 6.06% |
| | | Multivariate | 37.76% | 7.76% | 9.09% |

*3.2. Metrics*

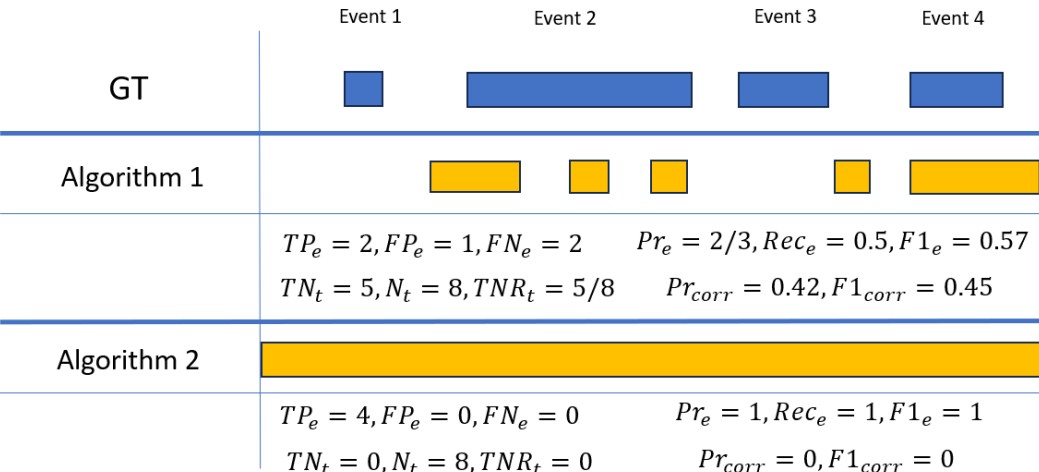

**Fig. 12.** Visualization of differences between the original and corrected event-wise F-scores.

European Space Agency Dataset and Benchmark for Anomaly Detection in Real-World Time Series

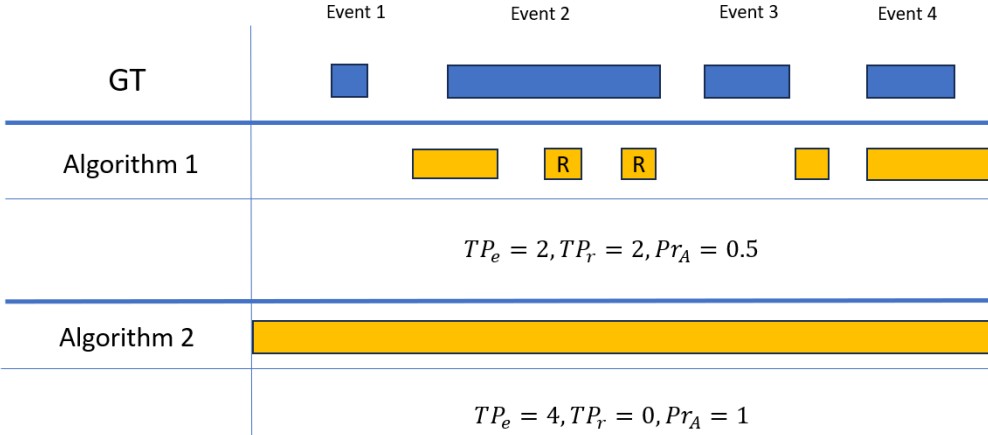

**Fig. 13.** Visualization of the event-wise alarming precision calculation.

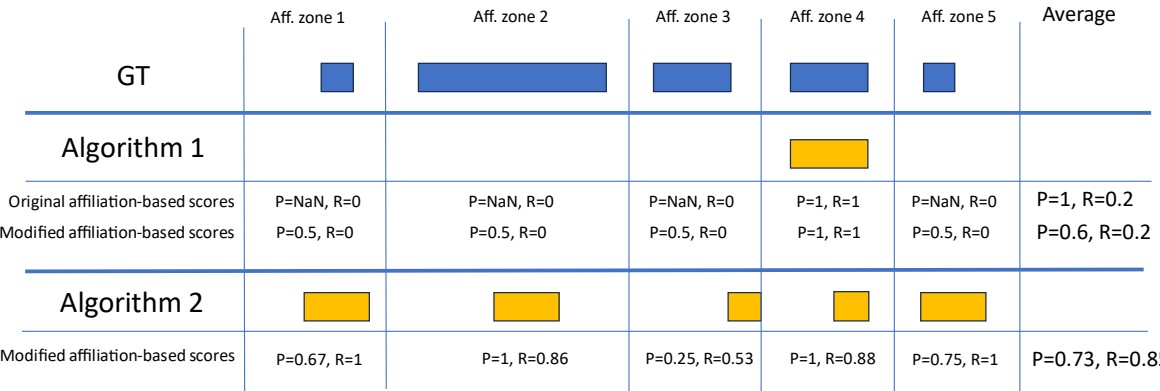

**Fig. 14.** Visualization of differences between the original and modified affiliation-based scores on a simulated example. (P – precision, R – Recall)

European Space Agency Dataset and Benchmark for Anomaly Detection in Real-World Time Series

### 3.2.1. Analysis of metrics from the literature

There are some recent comprehensive reviews of TSAD metrics in the literature [18], [19]. It is an active area of research in which many new approaches were introduced recently. However, none of these metrics is universal, they focus on different aspects of detection quality which may differ between applications. Table 12 gives an overview of our analysis of state-of-the-art metrics together with their pros and cons in the context of satellite telemetry anomaly detection.

TABLE 12
ANALYSIS OF EXISTING METRICS FROM THE LITERATURE WITH REASONS FOR INCLUSION/EXCLUSION IN ESA-ADB.

| Category | Metric | Included in ESA-ADB | Reason for inclusion/exclusion |
|---|---|---|---|
| **Sample-wise** | All classic precision-based and recall-based metrics treating each time point as an independent sample | No | Simple per-sample metrics are not aware of temporal aspects of time series in which anomalies are usually continuous sequences of correlated samples (like a vast majority of anomalies in satellite telemetry). This leads to multiple problems with a proper assessment of algorithms, i.e., longer anomalies are getting more importance than shorter ones, and close detections are not rewarded. |
| **Range-based** | NAB score [20] | No | It is only applicable to domains where an anomaly is a single sample rather than a series of samples. Moreover, there are certain ambiguities in the scoring functions and magic numbers for its parameters [21]. |
| | Point-adjust [22] | No | It is recently widely criticized for various reasons [23], [23], [24], [25], [26]. Mainly because it is overly optimistic and even a random anomaly score can reach state-of-the-art results in this metric in specific adversarial cases [23], [24], [26]. Also, it gives more importance to longer anomalies and is unable to give a larger score to a model which finds the true range of the anomaly better. |
| | Event-wise score [5] | No | It is improvement of point-adjust approach which partially solve the problem with higher importance of long anomalies, i.e., long anomalies have the same "weight" as short ones, but there is still a much higher probability that a random detector will detect the longer anomaly. Moreover, an algorithm that simply detects anomalies for every sample in the dataset would have a perfect event-wise precision. |
| | Composite F-score [27] | No | It calculates the precision in a per-sample manner and recall in an event-wise manner, so it still gives smaller weights to short false positives which are equally or more annoying than longer ones. |
| | Corrected event-wise F-score [26] | Yes | It resolves issues with the event-wise precision using a correction for per-sample true negative rate. It increases importance of short false positives and penalizes overly long detections. |
| | Precision and Recall for Time Series [28] | No | It requires 4 hyperparameters to be tuned. The recall is not monotonically decreasing with an increasing threshold. Such a behavior can even lead to problems when computing aggregated metrics that assume recall consistency [6]. Its calculation time does not scale well for large datasets. |
| | TaPR and eTaPR [23] | No | It requires 3 hyperparameters and its calculation time does not scale well for large datasets. |
| | Affiliation-based score [24] | Yes | It is parameter-free, locally and statistically interpretable, robust to "adversary" predictions, and scales well for large datasets. |
| | Volume Under the Surface [29] | No | It operates on continuous anomaly scores and we require binary outputs. It has very high computational complexity and does not scale well for large datasets. |
| **Detection timing** | Early Detection [30] | No | It assumes that anomalies can only be detected within the ground truth interval – *after* they appear in the signal. It does not account for too early detections which may be treated as false positives by spacecraft operators. |
| | Before/After-TP [31] | No | This approach requires calculation of two different values (Before-TP and After-TP) which are not obvious to aggregate to assess the overall detection timing quality. Moreover, it is impossible to calculate both values for every detections. |

| | | | |
|---|---|---|---|
| **Anomaly diagnosis (Channels identification)** | HitRate@P% [12] | No | It needs information about the relative relevance of detections that is not available when using binary outputs. It does not penalize models detecting too many channels. To always achieve a score of 1 model can simply mark all channels as anomalous. It is a crucial problem in satellite telemetry because we aim to mark only relevant channels to investigate by space operations. Any additional irrelevant detections should be penalized. |
| | Normalized Discounted Cumulative Gain [32] | No | It needs information about the relative relevance of detections that is not available when using binary outputs. |

### 3.2.2. Description of metrics priorities

The highest priority aspect relates to the proper identification of anomalous events, but with a strong emphasis on avoiding false alarms at the same time (aspects 1a. "No false alarms" and 1b. "Anomaly existence" in Table 3 in the main text). This is because false positives are costly to resolve and deter operators from using the system. A high false positive rate is reported in the literature as the main obstacle to the wider adoption of anomaly detection algorithms in space operations [5]. This fact additionally supports our idea of hierarchical evaluation, since a high false positive rate disqualifies an algorithm even if it obtains perfect scores in other aspects. Moreover, most other aspects focus only on performance for true positive detections (i.e., channel identification, alarming precision, timing quality), so they indirectly depend on the anomaly existence aspect.

The second highest priority for SOEs is to have information about subsystems and channels affected by anomalies (aspects 2a and 2b). Proper subsystem identification is more important for SOEs as it gives a more concise overview of the situation than a long list of specific affected channels. Again, it is of paramount important to avoid false positives. It is strongly preferable to miss some channels rather than to wrongly identify many irrelevant channels. ESA-AD contains tens of target channels which is already hardly manageable for manual analysis, moreover, it is just a fraction of channels from actual missions. Hence, an algorithm which does not provide affected channels is of low practical utility, or even worse, it may amplify the "black box" nature of advanced algorithms and decrease trust in this kind of system among operators. That is why it was considered as the second of two *primary* aspects of highest priority.

The following 3 *secondary* aspects are not so crucial for SOEs but are certainly useful to differentiate between algorithms having the same *primary* scores. The 3rd priority is to avoid algorithms that frequently repeat alarms for the same continuous anomaly segment (aspect 3. "Exactly one detection per anomaly"). It is strongly connected to the 1a. "No false alarms" priority, because even if all repeated alarms are true positives, they would be annoying and confusing to operators, nearly as badly as false positives. The last 2 priority levels directly relate to the anomaly detection timing. It is better to detect anomalies earlier than later (aspect 4. "Detection timing"), it is preferable to detect a whole time range of an anomaly instead of just a part of it, and, in case of false detections, it is better to show them close to real anomalies (aspect 5. "Anomaly range and proximity"). These aspects are often highly emphasized in TSAD benchmarks from the literature, e.g., NAB [20] and Exathlon [14]. However, they are relatively less important for on-ground mission control. Additionally, the latter aspect cannot be precisely assessed due to the problems with the objective identification of anomaly end times (discussed in Appendix Section 2.3).

### 3.2.3. ESA-ADB metrics definitions

The definitions of the proposed metrics are given in the following paragraphs. All implementations are available in the published code. All metrics are defined in the <0, 1> range where 1 is the perfect score. Technical details of implementations are listed in Section 3.2.4.

**Subsystem/channel-aware F-scores.** Typical TSAD metrics are applicable only in univariate settings. To get a single score for multiple channels, there must be some aggregation performed. Such aggregation loses information about the performance for individual subsystems or channels, so it is impossible to assess their correct identification. In recent articles [33], [34], special *anomaly diagnosis* metrics are proposed to address this issue, namely *HitRate* and *Normalized Discounted Cumulative Gain*. These metrics measure how relevant are the detected channels according to the list of annotated channels. However, they need information about the relative relevance of detections which is not available when using binary outputs. Thus, a new approach is proposed based on precisions and recalls of identifying the list of affected subsystems and channels.

SOEs inspect potential anomaly sources at two levels of detail. First, they check which subsystems are affected by the anomaly. Later on, they look at the specific channels affected in those subsystems. The usefulness of algorithms supporting such inspection is proposed to be measured with *subsystem-aware* (SA) and *channels-aware* (CA) F-scores. A subsystem is counted as true positive ($TP_{SA}$) if it has at least one annotated channel and at least one detected channel (not necessarily the same) overlapping with the full time span of an event. A subsystem is considered a false negative ($FN_{SA}$) if it has at least one annotated channel but no such detections. A false positive subsystem ($FP_{SA}$) has no annotated channels but has at least one such detection. Thus, the *subsystem-aware F-score* $F_{\beta_{SA}}$ is given by (3).

$$F_{\beta_{SA}} = (1 + \beta^2)\frac{Pr_{SA} \cdot Rec_{SA}}{(\beta^2 \cdot Pr_{SA}) + Rec_{SA}},$$

$$Pr_{SA} = \frac{TP_{SA}}{TP_{SA} + FP_{SA}}, \qquad Rec_{SA} = \frac{TP_{SA}}{TP_{SA} + FN_{SA}}$$

(3)

The *channel-aware F-score* is defined analogously, taking into account separate channels instead of subsystems. Again, 0.5 is used for $\beta$ as a baseline to be consistent with the event-wise F-score. For the lightweight subsets of channels (selected from a single subsystem), the subsystem-aware F-score is not reported.

**Event-wise alarming precision.** The corrected event-wise F-score counts only a single true positive even if there are multiple separated detections for the same fragment in the ground truth (Fig. 13). In practice, such redundant detections may cause repeated alarms which may be annoying for operators. The *event-wise alarming precision* measures the ratio of correctly detected events to the sum of correctly detected events and redundant alarms. This metric may seem too strict in some cases, i.e., many short detections close to each other, but it represents a practical aspect of mission operations and encourages applying better thresholding or postprocessing approaches to avoid redundant alarms.

**Anomaly detection timing quality curve (ADTQC).** The goal of this novel metric is to assess the accuracy of the anomaly start time identification from the SOEs point of view. Some existing metrics of the anomaly detection latency, such as *After-TP* [31] or *Early Detection* (ED) [30], assume that an anomaly can be detected only within its ground truth interval – *after* it appears in the signal. However, the question arises how to assess algorithms that detect anomalies too early – *before* they start. They cannot be assessed using *After-TP* or *ED* metrics but they certainly have some value. The *Before-TP* metric [31] and the *NAB score* [20] rank earlier anomaly *predictions* (to distinguish them from *detections*) as better. However, in practice, as suggested by SOEs, too-early detections may be seen as false positives by operators if they cannot confirm the existence of an anomaly within a definable time. Thus, too early detections may decrease operators' trust in an algorithm and, in this context, are much worse than late detections of comparable distance from an anomaly start time. According to SOEs, the quality of anomaly detection timing should decrease exponentially for detections before the actual start time as opposed to much slower degradation of quality for moderately late detections. A survey was conducted and confronted across SOEs from different missions in ESA and KP Labs to define the timing quality in the range from 0 to 1 as a function of detection start time. The resulting consensus reflecting the common operators' point of view is reflected in the ADTQC described by Appendix Equation (4) in Appendix Section 3.2.2.

**Modified affiliation-based F-score.** The affiliation-based approach by Huet et al. [24] claims to resolve all the major flaws of previous range-based metrics. That is, it is aware of the temporal adjacency of samples and anomalies duration, has no parameters, and is locally and statistically interpretable. The main idea is to divide ground truth into local zones affiliated with consecutive anomaly ranges. The borders of such *affiliation zones* lie in midpoints between consecutive anomalies. Precision and recall are calculated separately for each affiliation zone based on the average directed distance between sets of annotated and detected points, either the distance from annotated to detected (precision) or from detected to annotated (recall). Affiliation-based F-score with $\beta$ of 0.5 is calculated to underscore the strong practical need to minimize the number of false positives. The final global F-score is calculated as the arithmetic average of all local F-scores (with each affiliation zone having the same weight). The important modification to the original implementation relates to frequent situations when it is impossible to calculate the precision in an affiliation zone (when there is no detection there are no true positives or false positives). In the original formulation, such an affiliation zone was simply ignored when calculating an average precision over all affiliation zones. However, this approach makes it hard to robustly compare different algorithms because of the different numbers of affiliation zones taken into account, e.g., it gives a higher score to an algorithm that detects a single anomaly very precisely and misses 4 others than to an algorithm that detects all 5 anomalies relatively well – see Fig. 14. In our formulation, empty detections get a precision of 0.5. Such a value can be interpreted as a random detection, so this modification promotes algorithms that would rather give an empty detection than a false detection that is worse than random. There are also some other technical adaptations to handle point anomalies and fragmented annotations, as described in Appendix Section 3.2.2.

### 3.2.4. Implementation details

**Operating in the time domain**. ESA-AD has varying sampling rates and we keep them on purpose to maintain the true characteristics of satellite telemetry data. Our evaluation pipeline should handle this issue to consistently evaluate the results of algorithms using different sets of timestamps on the output. The only way to achieve this is to use metrics operating in the time domain instead of the samples domain, so that the ground truth and the detections can use completely different sets of timestamps (of different lengths and varying sampling rates). Original implementations of most metrics do not support timestamped arrays. They assume that the ground truth and the detections have the same uniformly sampled timeline. Our metrics operate on arbitrarily timestamped ground truth and detection arrays (possibly of different lengths and sampling frequencies). Hence, no matter the sampling frequency used in the algorithm, the metrics are always calculated relative to the original non-uniformly sampled ground truth. For operations on time ranges, we use the *portion* library (github.com/AlexandreDecan/portion).

**Timestamps as real numbers.** Our modified version of the affiliation-based metric operates on timestamped arrays, but the timestamps are transformed into the number of nanoseconds since the beginning of the dataset, so the internal implementation of the affiliation-based score is unchanged (it can operate on real numbers only). Additionally, point events (with the same start and

end times) are adjusted, so that the end time is 1 nanosecond later than the start time. Such modification had to be applied because the affiliation-based score cannot be calculated for point events of zero length. The same point anomalies adjustment was applied to the channel-aware F-scores.

**Multiple annotations for the same event.** It is a common situation in our dataset that multiple non-overlapping fragments close to each other are annotated with the same event ID (e.g., Fig 2. in the main text). This is usually because the source of the anomaly is the same for all fragments. In such cases, we should treat all fragments as a single anomaly (i.e., when selecting affiliation zones and calculating distances) as suggested in recent literature [4], [24]. To implement such correction in the affiliation-based score without changing its internal assumptions and implementation, a macro-averaging across anomaly IDs was introduced, i.e., it first aggregates zones affiliated with the same anomaly IDs by averaging their precision and recall scores and then calculates an average across all anomaly IDs. It also affects the implementation of the alarming precision metric which does not penalize redundant detections belonging to non-overlapping ground truth fragments.

**Overlapping events.** There are several cases in our dataset where annotations of different events for the same channel are overlapping in time, i.e., when an anomaly occurred during a longer rare event. The affiliation-based metric is unable to separate such events because it is impossible to create non-overlapping affiliation zones for them, so there are no corrections for this situation to not interfere with the main principles and assumptions of the metric. For subsystem-aware and channel-aware metrics, each event is analyzed separately, i.e., separate true positives and false negatives are counted for each event. Moreover, any false positives related to correct detections of other overlapping anomalies are discarded.

**ADTQC details.** The *anomaly detection timing quality curve (ADTQC)* is defined by (4) and visualized in Fig. 15.

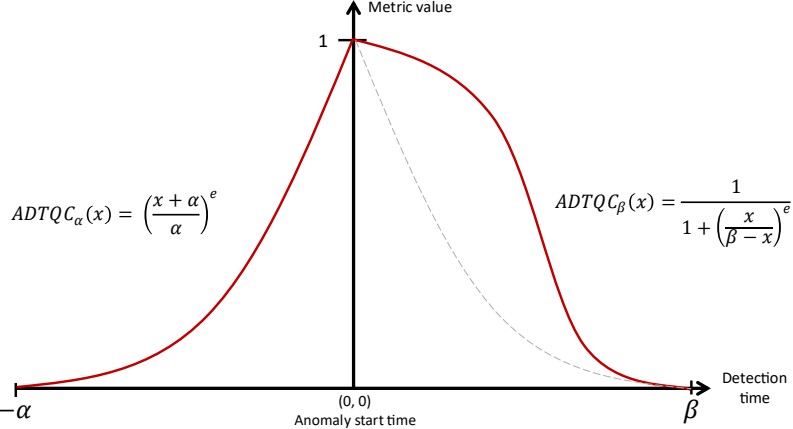

**Fig. 15.** Anomaly detection timing quality curve (ADTQC).

$$ADTQC(x) = \begin{cases} 0 & , & -\infty < x \leq -\alpha \\ \left(\dfrac{x+\alpha}{\alpha}\right)^e & , & -\alpha < x \leq 0 \\ \dfrac{1}{1+\left(\dfrac{x}{\beta-x}\right)^e} & , & 0 < x < \beta \\ 0 & , & \beta \leq x < +\infty \end{cases} \quad , \quad \alpha, \beta > 0 \tag{4}$$

$$ADTQC(x) = \begin{cases} 0, & x \neq 0 \\ 1, & x = 0 \end{cases}, \quad (\alpha = 0 \wedge x \leq 0) \vee (\beta = 0 \wedge x \geq 0)$$

$$\alpha = \min(anomaly\ length, anomaly\ start\ time - previous\ anomaly\ start\ time)$$

$$\beta = anomaly\ length$$

After agreeing on the shape of ADTQC, the most important issue was to select the operational range of values for which the function should return a quality higher than 0, that is, for which the detection is not useless from the practical point of view. The first straightforward step was to define detections later than the anomaly end time ($\beta$) as useless. Accordingly, detections earlier than the anomaly length from the start time were also considered useless. Hence, the shorter the anomaly the more accurately it must be detected to achieve similar quality value. In the extreme case of point anomalies, ADTQC returns a value of 1 for exact detections and 0 otherwise. It makes sense from the practical point of view for two reasons, 1) detections for short, hardly noticeable anomalies are likely to be considered false alarms if not well-timed, and 2) end times of long anomalies are usually much harder to annotate precisely than for short anomalies. Another unacceptable situation was identified when a detection is earlier than the previous anomaly start time. When anomalies are close to each other, the detection timing must be even more accurate to ensure their better separation.

The ADTQC metric value for the specific anomaly is determined by simply calculating the value of the $ADTQC(x)$ function where $x$ is the difference between the detection start time and the anomaly start time. Similarly to Before/After-TP [31], the metric is calculated and averaged across all correctly detected events to get a final score in the range from 0 to 1. To support the analysis of the results, the ratio of detections after the anomaly starting points to all detections is calculated (called the *after ratio*).

For multivariate anomalies, the ADTQC metric is calculated between the logical sums of annotations and detections across all target channels. It does not matter if the detections are for correct channels because the metric focuses on the timing alone. The second possible approach in the multivariate setting would be to calculate the ADTQC metric for each affected channel separately. The average across all affected channels would be the final ADTQC score for a specific anomaly. While this alternative approach would allow for more detailed quantification of the anomaly detection timing across channels, it does not reflect the operators' perspective in which the first detection is the most important one, because it already enforces an action. Later detections for any other channel do not matter so much, because operators are already aware of the potential anomaly.

All metrics can be calculated excluding some specific event categories, classes, or types. For the corrected event-wise F-score, detections for excluded events are ignored when counting true and false positives, and a lack of detection is not counted as a false negative for them. For other metrics, excluded events are simply not considered when calculating the mean across events.

### 3.2.5. Approach for rare nominal events

Most algorithms in the TimeEval framework (and in the literature) do not support learning rare nominal events explicitly (i.e., by one-shot learning or keeping rare events in memory). For such standard algorithms, rare events will always be detected as anomalies, so for simplicity, rare nominal events are treated as anomalies in the current benchmark. However, we strongly encourage to use ESA-AD to design models that learn nominal rare events and avoid detecting them in the future, which would be of high practical importance for mission control. For this purpose, we propose a framework to assess them:

1) The first detection of a novel rare nominal event (not seen during training) should not be penalized. However, the algorithm should be able to actively learn from the operators' feedback (i.e., "*this is not an anomaly*") and should not detect similar events in the future (one-shot learning).

2) For known rare events (seen during training or actively learned during inference), every subsequent detection should be penalized, i.e., we should minimize per-event false positive rate (FP / (FP + TN)) where FP is falsely detected rare event and TN is a correctly undetected rare event.

### 3.3. Preprocessing

Fig. 16 presents an example of the proposed zero-order hold resampling scheme. It is implemented as follows:

1. *Construct a uniformly sampled list of timestamps in the target sampling frequency*. Set the first/last timestamp in the list to the value of the earliest/latest original timestamp across all channels rounded down/up to the target sampling resolution. Fill the list between the first and the last element using uniformly sampled timestamps in the target frequency, e.g., if we resample a list of original timestamps <8:10:12, 8:10:14, 8:10:38> to the target frequency of 1/10 Hz (target resolution of 10 seconds), the resampled list will be <8:10:10, 8:10:20, 8:10:30, 8:10:40>.

2. *Propagate the last known value and label from the original samples* (zero-order hold) to each timestamp in the constructed list. If there are still any missing values for the initial element of the list (i.e., when some channels start a little earlier than others), backpropagate the first known value from the original samples. This introduces a bit of information from the future, but it usually concerns only a few samples at the beginning of a test set.

3. *Apply a correction for missing anomalies* to ensure that no point events are removed due to the resampling. Iterate through consecutive pairs of unannotated timestamps in the resampled list and, if there are any annotated original points in between, take the last annotated sample and assign its value and label to the latter timestamp from the pair. The result of such a correction is visible in the rightmost sample of Channel_1 in Fig. 16.

Target sampling frequencies have been selected separately for each mission (0.033 Hz, 0.056 Hz, 0.065 Hz, for Missions 1, 2, 3, respectively) based on the analysis of the most densely sampled target channels to prevent losing any annotated anomalies, especially point anomalies.

Channels with categorical values and status flags are enumerated according to the order of occurrence of each state in the training set before standardization. This is a very naïve approach, but it does not require laborious manual analysis of all channels and preparation of state mappings for each potential mission. Also, it does not require special handling of categorical anomalies. Moreover, categorical channels are usually non-target.

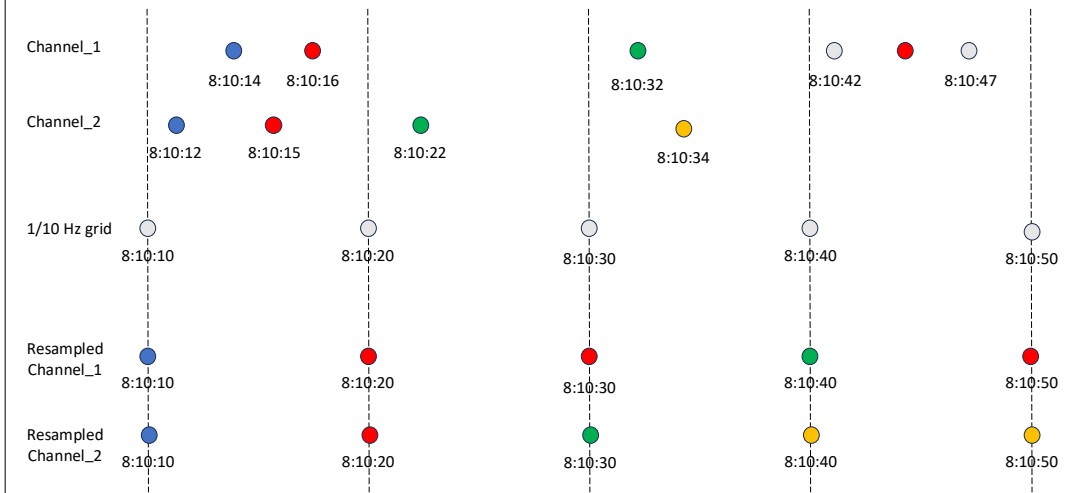

**Fig. 16.** Visualization of our resampling procedure for two non-uniformly sampled channels. Colors represent different values of the signal for each channel.

### 3.4. Algorithms

This section provides all details on algorithms implementation, selection, and parametrization in the benchmark.

#### 3.4.1. Telemanom adaptation (Telemanom-ESA)

The semi-supervised Telemanom algorithm proposed by NASA engineers [5] is an important point of reference in the domain. It can be considered the most popular algorithm for anomaly detection in satellite telemetry. Its core element is an LSTM-based RNN that learns to forecast a small number of time points (10 by default) for a single channel based on the hundreds of preceding samples (250 by default) from multiple input channels. The mean absolute difference between the forecasted samples and the real signal is treated as an anomaly score, which is thresholded using the non-parametric dynamic algorithm (NDT) to find anomalies. However, this "non-parametric" approach (in the sense that it does not use Gaussian distribution parameters to estimate thresholds) has several hyperparameters. In one of our previous works, a genetic algorithm was used to find optimal hyperparameters [35]. However, this wrapper approach would be too computationally expensive to run on our large datasets, so the default settings proposed by the authors were used.

Our proposed *Telemanom-ESA* solves several technical issues of the original Telemanom identified when working on ESA-ADB, including memory inefficiency, magic numbers causing problems in thresholding (*Telemanom-ESA-Pruned*), no proper handling of anomalies in training data, and a lack of scalability to hundreds of channels.

- **Memory inefficiency** – Telemanom was designed for small and simplified datasets provided by NASA. Hence, the code is not optimized to handle very large datasets and it results in out-of-memory errors, e.g., there are many unnecessary copies of data, all training windows are loaded into memory at once, and binary annotations are loaded to memory as floating-point numbers. **Telemanom-ESA:** The code is optimized for memory consumption by using lazy generators to prepare training batches, in-place operations instead of copying data to new variables, and optimized data types.

- **Magic numbers in thresholding** – there are several conditions in the thresholding code that are not well documented in the original article. Especially impactful is that windows with smoothed errors below 0.05 are never anomalous (github.com/khundman/telemanom/blob/26831a05d47857e194a7725fd982d5dea5402dd4/telemanom/errors.py#L339). This is a very data-specific condition that is not well-suited for channels with certain signal values. **Telemanom-ESA**: This specific condition was removed from the code. **Telemanom-ESA-Pruned**: The threshold of 0.05 is much too high for ESA-ADB, so it was changed to 0.007 based on the manual analysis of smoothed errors in the training data of both missions. This selection is highly subjective and is probably not optimal, but allows to assess the effect of such a pruning on the results.

- **No proper handling of anomalies in training data** – Telemanom assumes that there are no anomalies in the training set which is not true in our real-life setting. **Telemanom-ESA**: only continuous nominal parts longer than 260 samples and without any anomalies in any target channel are used for training and validation.

- **Only a single output from the LSTM model** – a single Telemanom model can take multiple input channels but it always outputs a prediction for a single target channel. This is a significant shortcoming when scaling this approach to hundreds of channels and gigabytes of data. The training of a single model may last hours or days, so training separate models for tens of channels can take months on a single PC. Also, it is impossible to provide different sets of input

(non-target channels, telecommands) and output (target) channels. **Telemanom-ESA**: the output of Telemanom is extended, so that is possible to forecast any number of channels at once from a single model, like in DC-VAE[13]. The channels are still analyzed separately, but there is no need to train a separate model for each channel.

- **Problems with GPU support** – the original implementation of Telemanom is based on TensorFlow version 2.0 which does not natively support the CUDA compute capability 8.6 of our Nvidia GPUs. Also, the TimeEval framework lacks GPU support. **Telemanom-ESA**: TensorFlow is upgraded to version 2.5 and the GPU support is added to TimeEval.

*3.4.2. GlobalSTD*

In this simple distribution-based approach, any samples deviating from the mean of the channel by more than $N$ its standard deviations are detected as anomalies. This approach is categorized as semi-supervised because only nominal samples (excluding annotated events) from the training set are used to compute means and standard deviations for each channel to avoid the influence of outliers. In practice, the threshold of 3 standard deviations (*GlobalSTD3*) is frequently used (following the empirical statistical rule that 99.7% of data occurs within 3 standard deviations from the mean within a normal distribution [36]), but it may not be optimal when the number of false positives should be minimized, so the threshold of 5 standard deviations (*GlobalSTD5*) is also tested to provide a versatile baseline for other algorithms. This algorithm is unable to detect local anomalies, so it is not a good choice in practice. It is also not aware of dependencies between channels and it is very vulnerable to changes in the data distribution during the mission. It also cannot use the information about non-target channels and telecommands.

*3.4.3. DC-VAE and its adapted version (DC-VAE-ESA)*

Dilated Convolutional-Variational Auto Encoder (DC-VAE) [13] is one of the latest published multivariate TSAD algorithms. It is a reconstruction-based method that relies on dilated convolutions to capture long and short-term dependencies without using computation- and memory-intensive multi-layer RNNs. Unlike the original Telemanom, it does not need a complicated thresholding scheme, because it also estimates nominal standard deviations for each sample in each channel, so thresholding can simply be applied by looking for real samples exceeding reconstructions by more than $N$ standard deviations. In the original implementation, $N$ is selected from integers between 2 and 7 to maximize the range-based F1-score for each channel in the training set. This approach does not scale well with the number of channels and assumes the similarity of anomalies between the training and test sets. Thus, in DC-VAE-ESA, only two values of $N$ are considered, 3 (*STD3*) and 5 (*STD5*).

The modified DC-VAE-ESA introduces only two small technical improvements to fully cover 7 of the 9 mentioned requirements, 1) an option to handle different numbers of input and output channels, 2) L2 regularization of convolutional layers with the 0.001 rate to stabilize the training of VAE in the presence of concept drifts.

*3.4.4. Experiments with transformers*

We have experimented with transformer-based anomaly detectors, namely TranAD [34] and Anomaly Transformer [37] algorithms, using the code from their original repositories. We have not included the results in the current benchmark for several reasons:

1. We were not able to achieve any reasonable qualitative results using the default hyperparameters of these algorithms. They would require additional investigation and hyperparameters selection that go beyond the scope of the study and computational resources allocated to it.
2. The algorithms are not a part of the TimeEval framework and it would require additional effort to integrate and make them reproducible within our pipeline.
3. The original implementations do not handle irregular sampling rates of data. Their data loaders assume uniform sampling when creating windows for training, so it is not possible to take advantage of the positional enconding of transformers.
4. The original implementation of Anomaly Transformer provides just a single global anomaly score, so it does not meet the requirement R5 from Table 2 in the main text ("provide a list of affected channels").

Nevertheless, transformers seem to be a promising direction for future work based on the ESA-ADB benchmark.

*3.4.5. Algorithms' selection*

Based on the initial requirements analysis, 20 algorithms were preselected among those available (or added) in the TimeEval framework that at least partially meet all primary requirements. Table 13 summarizes the detailed requirements analysis for those algorithms. Some examples of partially fulfilled requirements are for algorithms that R1) do not provide dedicated thresholding mechanisms, R2) technically allow for the online detection but with a large computational overhead, R4) handle anomalies in training data but cannot learn from them, R5) would need additional mechanisms or modifications of external libraries (i.e., PyOD [38]) to provide a list of affected channels, R7) give only a theoretical option to learn rare nominal events, or R9) are only possible to run for the lightweight subsets of channels (i.e., Windowed iForest and KNN). None of the preselected algorithms are able to explicitlyearn rare nominal events (R7) or handle varying sampling rates (R8).

Based on the detailed analysis of the requirements, eight algorithms of various types were selected for ESA-ADB, five unsupervised – principal components classifier (PCC) [39], histogram-based outlier score (HBOS) [40], isolation forest (iForest) [41], k-nearest neighbours (KNN) [42], and three semi-supervised ones – global standard deviations from nominal (GlobalSTD),

Telemanom [5], and DC-VAE [13]. The selected unsupervised algorithms have several important limitations in terms of TSAD. They may be give suboptimal results because of the assumptions of independence of samples and identical fractions of anomalies in training and test data (they fulfil R4 because they learn contamination levels from the training data). They only give global scores, so it is impossible to calculate subsystem-aware and channel-aware scores for them. They also do not support non-target channels and telecommands on input, so this information was not used. However, they establish a baseline for more advanced algorithms.

Among the rejected ones, Matrix Profile-based methods like DAMP [43] or MADRID [44] seem to be promising candidates due to their outstanding speed, high interpretability, and a theoretical possibility to memorize rare nominal events. However, they would need a special adaptation to support multidimensional data [45], they do not give option to annotate known anomalies in training data, and their implementations in Matlab pose several technical and licensing problems when integrated with TimeEval. The COPOD algorithm does not fulfil R9 after adapting it to online detection required by R2. LOF [46], k-Means [47], Torsk [48], and RobustPCA [49] showed very poor results in initial experiments. All semi-supervised algorithms that only partially fulfil R9 were rejected. The published code contains implementations of all methods listed in Table 13.

TABLE 13
ANALYSIS OF PRESELECTED ALGORITHMS ACCORDING TO ESA-ADB REQUIREMENTS. 0/0.5/1 MEANS THAT THE REQUIREMENT IS NOT/PARTIALLY/FULLY FULFILLED. ASTERISKS MARK NEW METHODS ADDED TO THE TIMEEVAL. BOLD-FACED REQUIREMENTS ARE "MUST".

| Algorithm | | R1 | R2 | R3 | R4 | R5 | R6 | R7 | R8 | R9 | Included in ESA-ADB |
|---|---|---|---|---|---|---|---|---|---|---|---|
| UNSUPERVISED | COPOD [50] | 1 | 1 | 0.5 | 0.5 | 1 | 0 | 0 | 0 | 0.5 | NO |
| | HBOS [40] | 1 | 0 | 1 | 0.5 | 0.5 | 0 | 0 | 0 | 1 | YES |
| | iForest [41] | 1 | 1 | 1 | 0.5 | 0.5 | 0 | 0 | 0 | 1 | YES |
| | Windowed iForest [41] | 1 | 1 | 1 | 0.5 | 0.5 | 0 | 0 | 0 | 0.5 | SUBSETS |
| | k-Means [47] | 1 | 1 | 1 | 0.5 | 0.5 | 0 | 0 | 0 | 0.5 | NO |
| | KNN [42] | 1 | 1 | 1 | 0.5 | 0.5 | 0 | 0.5 | 0 | 0.5 | SUBSETS |
| | LOF [46] | 1 | 1 | 1 | 0.5 | 0.5 | 0 | 0 | 0 | 0.5 | NO |
| | Matrix Profile [43], [44] | 1 | 0.5 | 1 | 0 | 0.5 | 0 | 0.5 | 0 | 1 | NO |
| | PCC [39] | 1 | 1 | 0.5 | 0.5 | 0.5 | 0 | 0 | 0 | 1 | YES |
| | Torsk [48] | 0.5 | 1 | 1 | 0.5 | 1 | 0 | 0 | 0 | 0.5 | NO |
| SEMI-SUPERVISED | DAE [51] | 0.5 | 1 | 1 | 0 | 1 | 0 | 0 | 0 | 0.5 | NO |
| | DC-VAE [13]* | 0.5 | 1 | 1 | 0 | 1 | 0 | 0 | 0 | 0.5 | NO |
| | DC-VAE-ESA* | 1 | 1 | 1 | 0.5 | 1 | 1 | 0 | 0 | 1 | YES |
| | GlobalSTD* | 1 | 0 | 1 | 0.5 | 1 | 0 | 0 | 0 | 1 | YES |
| | Hybrid KNN [52] | 1 | 1 | 1 | 0 | 0.5 | 0 | 0.5 | 0 | 0.5 | NO |
| | LSTM-AD [53] | 0.5 | 1 | 1 | 0 | 0 | 0 | 0 | 0 | 0.5 | NO |
| | OmniAnomaly [12] | 0.5 | 1 | 1 | 0 | 0.5 | 0 | 0 | 0 | 0.5 | NO |
| | RobustPCA [49] | 0.5 | 1 | 0.5 | 0.5 | 0.5 | 0 | 0 | 0 | 1 | NO |
| | Telemanom [5] | 1 | 1 | 1 | 0 | 1 | 0 | 0 | 0 | 0.5 | NO |
| | Telemanom-ESA* | 1 | 1 | 1 | 0.5 | 1 | 1 | 0 | 0 | 1 | YES |

*3.4.6. Algorithms' parametrization*

To support the full reproducibility of our results, Table 14 lists all the algorithms' parameters and their values used in our experiments. The parameters' names directly correspond to the published code based on the TimeEval framework [54]. They use default values or settings recommended by algorithms' authors, sometimes adjusted to the specific features of our datasets (boldfaced in the table).

The number of 50 bins in HBOS was arbitrarily selected based on the analysis of the histograms of channels because the default value of 10 seemed to be much too small for our dataset. The default window size in Windowed iForest was decreased

European Space Agency Dataset and Benchmark for Anomaly Detection in Real-World Time Series

from 100 to 17 to avoid out-of-memory errors for our datasets. Many parameters of DC-VAE were adjusted to our dataset. The scaling is not used because it is already present in our preprocessing. Outliers are not rejected (*wo_outliers* is False) because our preprocessing code removes known anomalies. The window size is increased to 256 to be similar to the default Telemanom's window size (250). Also, the value of 256 showed good results on similar data in the original DC-VAE paper [13]. The number of CNN units is decreased from the default 64 to 32 because a significant overfitting was noticed in the validation scores for 64 units. The latent space dimensionality depends on the number of input channels in the same way as suggested for the TELCO dataset in the original DC-VAE code. The two main changes to Telemanom are 1) the increased number of units for full set training sets depending on the total number of input and output channels, and 2) the new min_error_value parameter to avoid magic numbers in the Telemanom code. The default value of the min_error_value is set to 0 (no magic numbers), but for Telemanom-ESA-Pruned it is arbitrarily selected to be 0.007 based on a manual analysis of reconstruction errors for the validation set, since the default value of 0.05 was much too high for some channels.

Importantly, the number of batches per epoch was limited to 1000 to avoid extremely long epoch training times for our datasets and to provide frequent validation score updates. Thus, the number of (sub)epochs was increased tenfold to 1000, and the early stopping patience was doubled to 20 for both DC-VAE and Telemanom to compensate for this.

TABLE 14

PARAMETRIZATION OF ALGORITHMS USED IN ESA-ADB. BOLDFACED PARAMETERS AND VALUES ARE DIFFERENT FROM THE DEFAULT ONES

| Algorithm | Parameter name | Value(s) |
|---|---|---|
| PCC | max_iter | None |
| | n_components | None |
| | n_selected_components | None |
| | random_state | 42 |
| | svd_solver | auto |
| | tol | 0.0 |
| | whiten | False |
| HBOS | **n_bins** | **50** |
| | alpha | 0.1 |
| | bin_tol | 0.5 |
| | random_state | 42 |
| iForest | n_trees | 100 |
| | bootstrap | False |
| | max_features | 1.0 |
| | max_samples | None |
| | random_state | 42 |
| Windowed iForest | **n_trees** | **200** |
| | **window_size** | **17** |
| | bootstrap | False |
| | max_features | 1.0 |
| | max_samples | None |
| | random_state | 42 |
| KNN | distance_metric_order | 2 |
| | leaf_size | 30 |
| | method | Largest |
| | n_neighbors | 5 |
| GlobalSTD | tol | **3 (STD3) and 5 (STD5)** |
| | random_state | 42 |
| DC-VAE-ESA | **alpha** | **3 (STD3) and 5 (STD5)** |
| | **T (window size)** | **256** |
| | **cnn_units** | **32 (16 for Phase 1)** |
| | dil_rate | [1,2,4,8,16,32,64] |
| | kernel | 2 |
| | strs (stride length of CNN layers) | 1 |
| | batch_size | 64 |
| | **J (latent space dimensionality)** | **1/3 × total number of input channels and telecommands** |
| | **epochs** | **1000** |
| | lr (learning rate) | $10^{-3}$ |
| | seed | 123 |
| | early_stopping_delta | 0.001 |
| | **early_stopping_patience** | **20** |

European Space Agency Dataset and Benchmark for Anomaly Detection in Real-World Time Series

| Telemanom-ESA | batch_size | 70 |
| | dropout | 0.3 |
| | early_stopping_delta | 0.0003 |
| | **early_stopping_patience** | **20** |
| | **epochs** | **1000** |
| | error_buffer | 100 |
| | layers | 2 |
| | **number of units per layer** | 80 for lightweight subsets.
**Total number of input and output channels for full sets.** |
| | lstm_batch_size | 64 |
| | **min_error_value (newly introduced to avoid magic numbers)** | **0**
**(0.007 for Telemanom-ESA-Pruned)** |
| | prediction_window_size | 10 |
| | random_state | 42 |
| | smoothing_perc | 0.05 |
| | smoothing_window_size | 30 |
| | window_size | 250 |

4. BENCHMARKING RESULTS

TABLE 15

BENCHMARKING RESULTS FOR DETECTION OF ALL EVENTS (EXCLUDING COMMUNICATION GAPS) IN LIGHTWEIGHT SUBSETS OF CHANNELS AND ALL CHANNELS FOR MISSION1. BOLDFACED RESULTS INDICATE THE BEST VALUES AMONG ALL ALGORITHMS (EXCLUDING AFTER ratio OF ADTQC WHICH IS JUST A HELPER VALUE).

| **Mission1 – trained and tested on the lightweight subset of channels 41-46** | | | | | | | | | | | |
|---|---|---|---|---|---|---|---|---|---|---|---|
| Metric | | PCC | HBOS | iForest | Window iForest | KNN | Global STD3 | Global STD5 | DC-VAE-ESA STD3 | DC-VAE-ESA STD5 | Teleman-ESA | Teleman-ESA-Pruned |
| Event-wise | Precision | < 0.001 | < 0.001 | < 0.001 | < 0.001 | < 0.001 | 0.001 | 0.288 | 0.002 | 0.063 | 0.148 | **0.999** |
| | Recall | 0.554 | 0.585 | 0.585 | 0.738 | 0.754 | 0.431 | 0.169 | 0.554 | 0.338 | **0.894** | 0.424 |
| | **F0.5** | < 0.001 | < 0.001 | < 0.001 | < 0.001 | < 0.001 | 0.001 | 0.253 | 0.003 | 0.075 | 0.178 | **0.786** |
| Channel-aware | Precision | | | | | | 0.431 | 0.169 | 0.550 | 0.338 | **0.894** | 0.424 |
| | Recall | | Not available for unsupervised algorithms | | | | 0.285 | 0.159 | 0.463 | 0.221 | **0.738** | 0.275 |
| | F0.5 | | | | | | 0.351 | 0.167 | 0.514 | 0.283 | **0.837** | 0.362 |
| Alarming precision | | 0.033 | 0.047 | 0.017 | 0.015 | 0.017 | 0.057 | 0.035 | 0.070 | 0.028 | 0.868 | **0.875** |
| ADTQC | After ratio | 0.833 | 0.763 | 0.711 | 0.375 | 0.612 | 0.929 | 0.909 | 0.972 | 0.955 | 0.136 | 0.143 |
| | Score | 0.840 | 0.781 | 0.784 | 0.563 | 0.803 | 0.770 | 0.688 | **0.901** | 0.803 | 0.428 | 0.197 |
| Affiliation-based | Precision | 0.535 | 0.543 | 0.543 | 0.599 | 0.522 | 0.559 | 0.699 | 0.584 | **0.780** | 0.727 | 0.711 |
| | Recall | 0.334 | 0.352 | 0.357 | 0.424 | 0.322 | 0.375 | 0.422 | 0.377 | 0.593 | **0.662** | 0.423 |
| | F0.5 | 0.477 | 0.490 | 0.492 | 0.553 | 0.464 | 0.509 | 0.618 | 0.526 | **0.734** | 0.713 | 0.626 |
| **Mission1 – trained and tested on the full set of channels** | | | | | | | | | | | |
| Metric | | PCC | HBOS | iForest | Window iForest | KNN | Global STD3 | Global STD5 | DC-VAE-ESA STD3 | DC-VAE-ESA STD5 | Teleman-ESA | Teleman-ESA-Pruned |
| Event-wise | Precision | < 0.001 | < 0.001 | < 0.001 | | | < 0.001 | 0.002 | < 0.001 | 0.005 | 0.007 | **0.050** |
| | Recall | 0.870 | 0.957 | **0.967** | | | 0.848 | 0.761 | 0.924 | 0.804 | 0.946 | 0.870 |
| | **F0.5** | < 0.001 | < 0.001 | < 0.001 | | | < 0.001 | 0.003 | < 0.001 | 0.007 | 0.008 | **0.061** |
| Subsystem-aware | Precision | | Not available for unsupervised algorithms | | | | 0.520 | **0.728** | 0.526 | 0.640 | 0.676 | 0.395 |
| | Recall | | | | | | 0.694 | 0.538 | 0.764 | 0.670 | 0.859 | **0.861** |
| | F0.5 | | | | | | 0.528 | 0.664 | 0.538 | 0.623 | **0.689** | 0.436 |
| Channel-aware | Precision | | Not available for unsupervised algorithms | | Out-of-memory | Out-of-memory | 0.380 | 0.276 | 0.398 | 0.359 | **0.514** | 0.267 |
| | Recall | | | | | | 0.292 | 0.208 | 0.414 | 0.266 | 0.569 | **0.725** |
| | F0.5 | | | | | | 0.325 | 0.241 | 0.350 | 0.282 | **0.477** | 0.291 |
| Alarming precision | | 0.003 | 0.002 | 0.001 | | | 0.004 | 0.049 | 0.002 | 0.017 | 0.074 | **0.206** |
| ADTQC | After ratio | 0.613 | 0.443 | 0.438 | | | 0.718 | 0.743 | 0.647 | 0.716 | 0.322 | 0.463 |
| | Score | 0.642 | 0.603 | 0.685 | | | 0.723 | 0.691 | **0.752** | 0.692 | 0.673 | 0.684 |
| Affiliation-based | Precision | 0.563 | 0.539 | 0.538 | | | 0.560 | 0.575 | 0.559 | 0.578 | 0.545 | **0.649** |
| | Recall | 0.522 | **0.578** | 0.456 | | | 0.492 | 0.462 | 0.476 | 0.511 | 0.368 | 0.484 |
| | F0.5 | 0.554 | 0.547 | 0.519 | | | 0.545 | 0.548 | 0.540 | 0.563 | 0.497 | **0.607** |

European Space Agency Dataset and Benchmark for Anomaly Detection in Real-World Time Series

TABLE 16

BENCHMARKING RESULTS FOR DETECTION OF ALL EVENTS (EXCLUDING COMMUNICATION GAPS) IN LIGHTWEIGHT SUBSETS OF CHANNELS AND ALL CHANNELS FOR MISSION1. BOLDFACED RESULTS INDICATE THE BEST VALUES AMONG ALL ALGORITHMS (EXCLUDING AFTER RATIO OF ADTQC WHICH IS JUST A HELPER VALUE).

| Mission2 – trained and tested on the lightweight subset of channels 18-28 | | | | | | | | | | | |
|---|---|---|---|---|---|---|---|---|---|---|---|
| Metric | | PCC | HBOS | iForest | Window iForest | KNN | Global STD3 | Global STD5 | DC-VAE-ESA STD3 | DC-VAE-ESA STD5 | Teleman-ESA | Teleman-ESA-Pruned |
| Event-wise | Precision | 0.029 | 0.055 | 0.557 | 0.951 | < 0.001 | 0.006 | 0.061 | 0.003 | 0.064 | 0.188 | **0.978** |
| | Recall | **1.000** | 0.911 | 0.974 | 0.940 | **1.000** | **1.000** | **1.000** | **1.000** | **1.000** | 0.986 | 0.540 |
| | F0.5 | 0.036 | 0.068 | 0.609 | **0.949** | 0.001 | 0.007 | 0.075 | 0.003 | 0.079 | 0.224 | 0.842 |
| Channel-aware | Precision | | | | | | 0.951 | 0.992 | 0.904 | **0.995** | 0.831 | 0.465 |
| | Recall | Not available for unsupervised algorithms | | | | | 0.462 | 0.372 | 0.554 | 0.451 | **0.870** | 0.384 |
| | F0.5 | | | | | | 0.767 | 0.723 | 0.787 | 0.783 | **0.822** | 0.442 |
| Alarming precision | | 0.061 | 0.105 | 0.075 | 0.217 | 0.060 | 0.054 | 0.061 | 0.052 | 0.068 | **0.912** | 0.862 |
| ADTQC | After ratio | 0.983 | 0.994 | 1.000 | 0.948 | 0.391 | 0.946 | 0.989 | 0.908 | 0.991 | 0.087 | 0.351 |
| | Score | **0.999** | 0.990 | 0.991 | 0.985 | 0.724 | 0.997 | 0.997 | 0.996 | 0.997 | 0.507 | 0.757 |
| Affiliation-based | Precision | 0.890 | 0.936 | **0.982** | 0.968 | 0.561 | 0.740 | 0.935 | 0.680 | 0.939 | 0.688 | 0.759 |
| | Recall | 0.580 | 0.867 | **0.952** | 0.925 | 0.243 | 0.296 | 0.717 | 0.293 | 0.788 | 0.544 | 0.530 |
| | F0.5 | 0.804 | 0.921 | **0.976** | 0.959 | 0.445 | 0.569 | 0.881 | 0.538 | 0.904 | 0.654 | 0.699 |
| Mission2 – trained and tested on the full set of channels | | | | | | | | | | | |
| Metric | | PCC | HBOS | iForest | Window iForest | KNN | Global STD3 | Global STD5 | DC-VAE-ESA STD3 | DC-VAE-ESA STD5 | Teleman-ESA | Teleman-ESA-Pruned |
| Event-wise | Precision | 0.082 | 0.016 | 0.022 | 0.034 | | 0.014 | **0.203** | 0.002 | 0.008 | 0.052 | 0.058 |
| | Recall | 0.983 | 0.820 | 0.903 | 0.746 | | 0.997 | 0.972 | **0.997** | 0.994 | 0.992 | 0.964 |
| | F0.5 | 0.1 | 0.02 | 0.027 | 0.042 | | 0.018 | **0.241** | 0.002 | 0.011 | 0.064 | 0.071 |
| Subsystem-aware | Precision | | | | | | 0.922 | **0.961** | 0.672 | 0.911 | 0.409 | 0.258 |
| | Recall | Not available for unsupervised algorithms | | | | | 0.953 | 0.923 | 0.967 | 0.952 | **0.984** | 0.896 |
| | F0.5 | | | | | | 0.919 | **0.946** | 0.699 | 0.907 | 0.451 | 0.298 |
| Channel-aware | Precision | | | | | Out-of-memory | 0.913 | **0.956** | 0.774 | 0.931 | 0.584 | 0.326 |
| | Recall | Not available for unsupervised algorithms | | | | | 0.454 | 0.376 | 0.592 | 0.507 | 0.783 | **0.823** |
| | F0.5 | | | | | | 0.745 | 0.715 | 0.713 | **0.783** | 0.592 | 0.368 |
| Alarming precision | | 0.183 | 0.148 | 0.112 | 0.179 | | 0.112 | 0.179 | 0.066 | 0.083 | 0.771 | **0.790** |
| ADTQC | After ratio | 0.980 | 0.906 | 0.939 | 0.852 | | 0.953 | 0.994 | 0.663 | 0.930 | 0.104 | 0.274 |
| | Score | 0.984 | 0.939 | 0.967 | 0.928 | | 0.983 | **0.992** | 0.825 | 0.985 | 0.513 | 0.648 |
| Affiliation-based | Precision | 0.758 | 0.570 | 0.621 | 0.608 | | 0.718 | **0.961** | 0.603 | 0.859 | 0.586 | 0.591 |
| | Recall | 0.636 | 0.455 | 0.499 | 0.474 | | 0.385 | **0.833** | 0.324 | 0.625 | 0.348 | 0.347 |
| | F0.5 | 0.730 | 0.543 | 0.592 | 0.575 | | 0.612 | **0.932** | 0.515 | 0.799 | 0.516 | 0.518 |

European Space Agency Dataset and Benchmark for Anomaly Detection in Real-World Time Series

*4.1. Example detections*

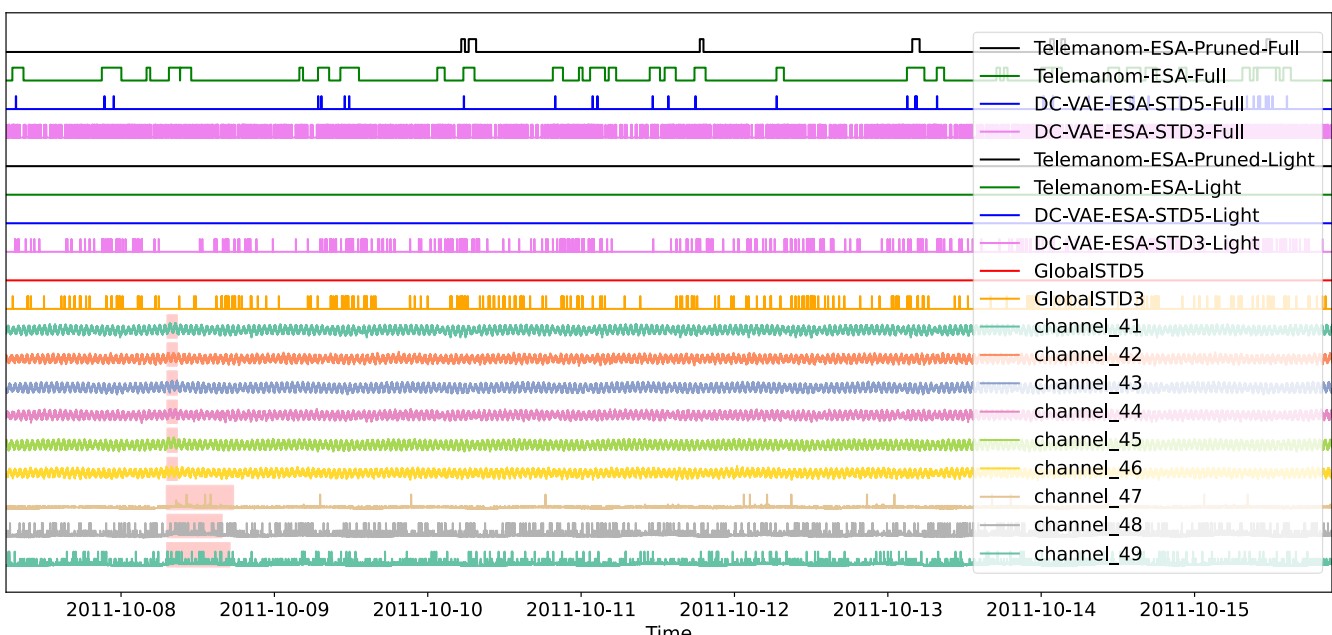

**Fig. 17.** Detections of rare nominal event id_49 (marked in red) for Mission1. It is not detected when using only the lightweight subset of channels 41-46. For the full set, only Telemanom-ESA shows a reasonable detection, but it is surrounded by many false detections.

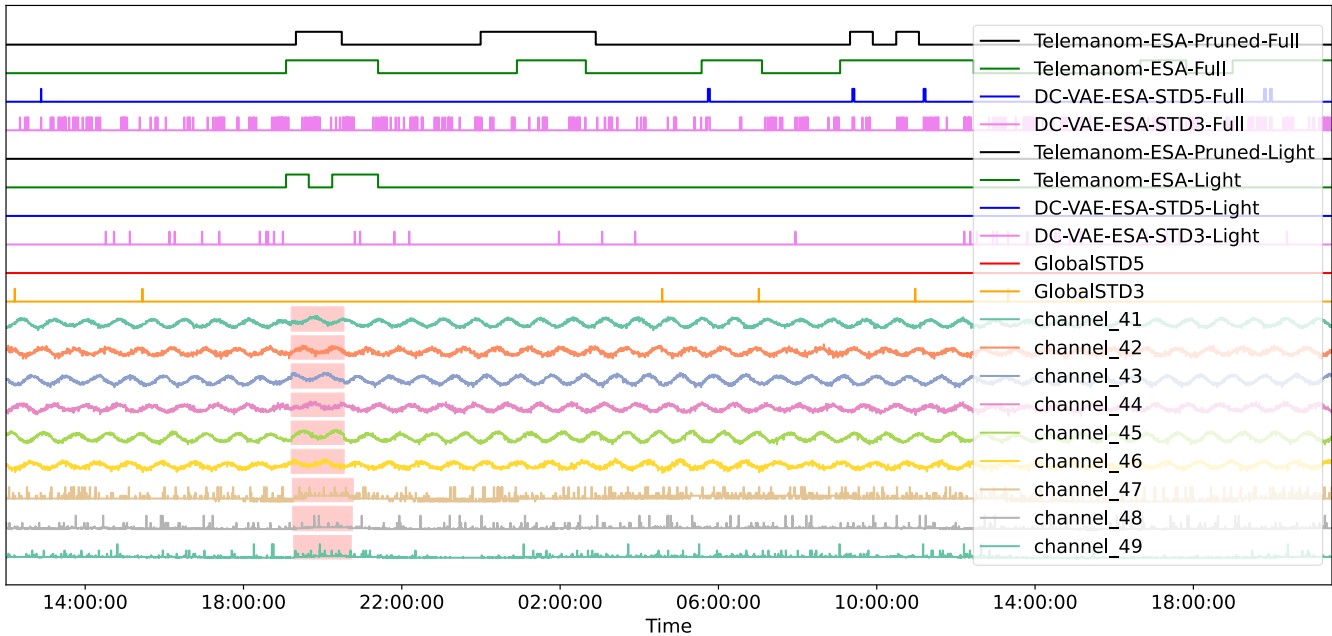

**Fig. 18.** Detections of rare nominal event id_51 (marked in red) for Mission1. It is reasonably detected only by Telemanom-ESA. Surprisingly, Telemanom-ESA trained on the lightweight subset was also able to detect this.

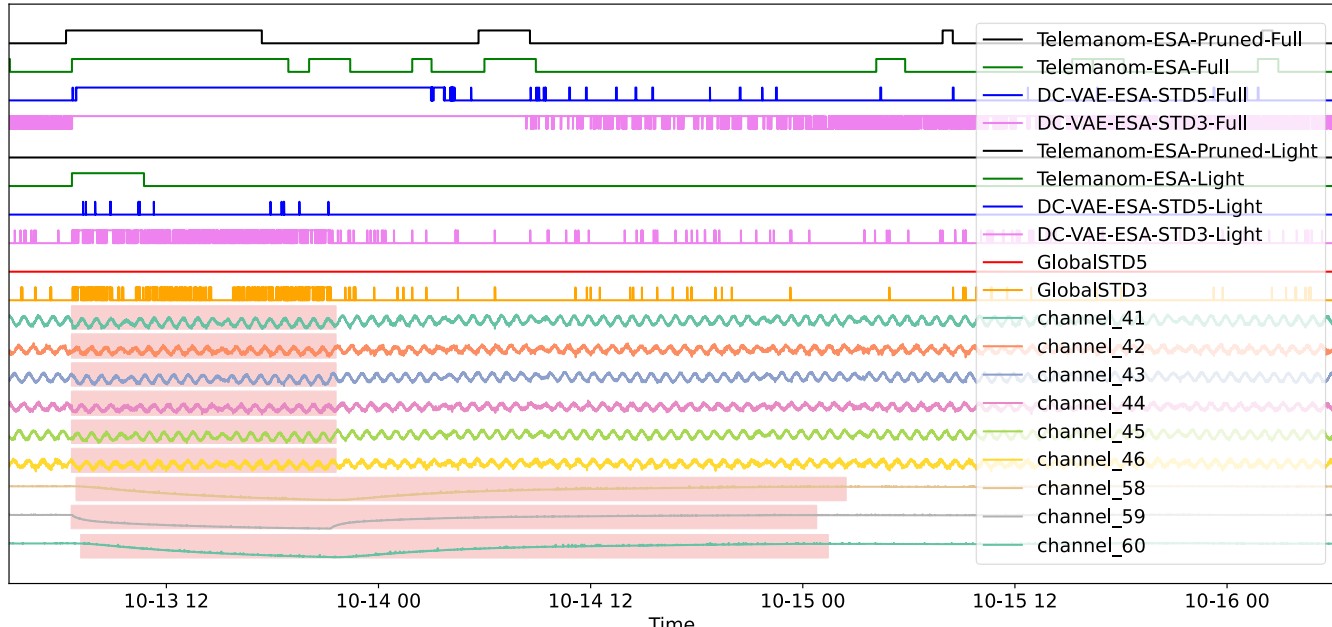

**Fig. 19**. Detections of anomaly id_138 (marked in red) for Mission1. It is clearly visible in channels 58-60, so it is detected well by models trained on full sets of channels. However, it is not so easy using only the lightweight subset, i.e., Telemanom-ESA-Pruned-Light shows no response.

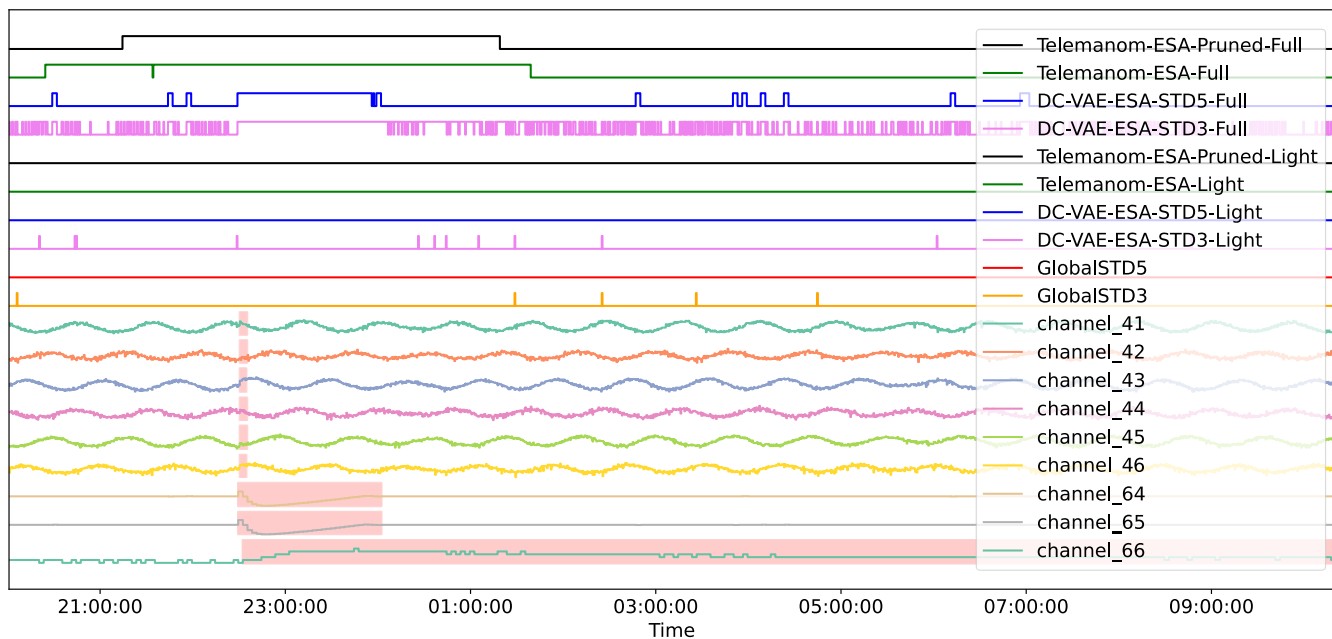

**Fig. 20.** Detections of anomaly id_153 (marked in red) for Mission1. It is not detected when using only the lightweight subset of channels 41-46. For the full set, it is detected by all algorithms. Telemanom-ESA-Full detects it too early.

European Space Agency Dataset and Benchmark for Anomaly Detection in Real-World Time Series

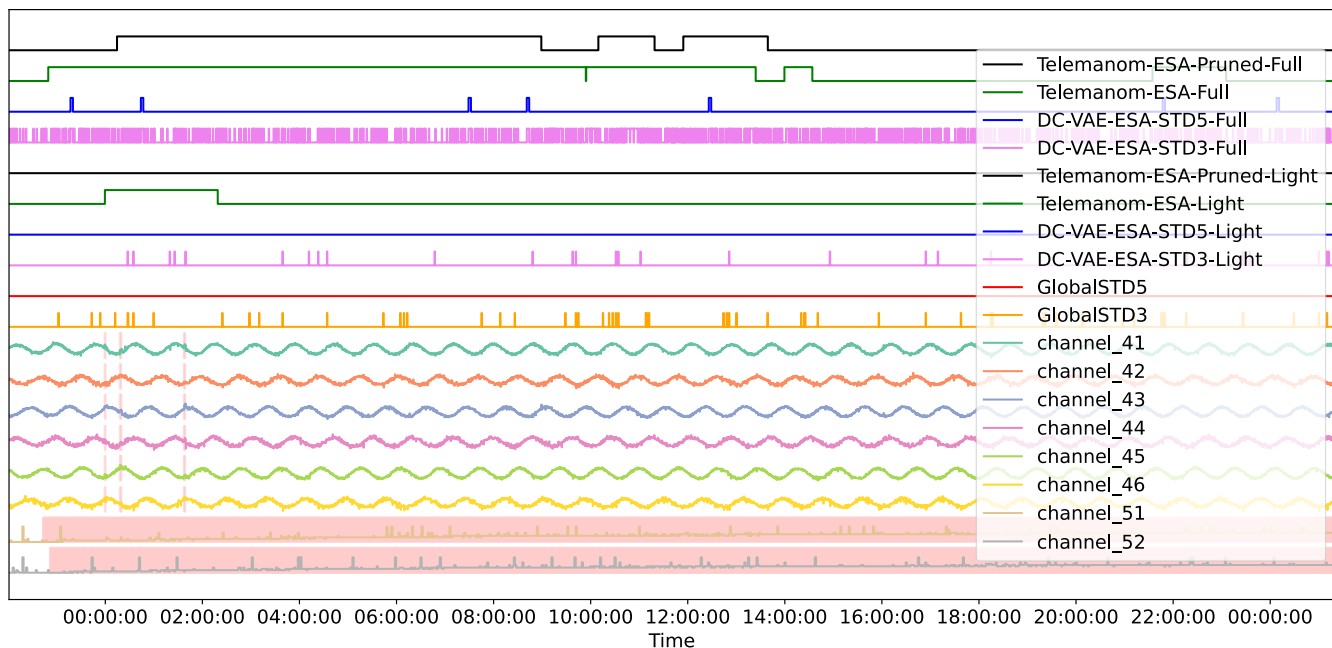

**Fig. 21.** Detections of rare nominal event id_155 (marked in red) for Mission1. Only Telemanom-ESA was able to correctly detect this event in both lightweight and full sets of channels.

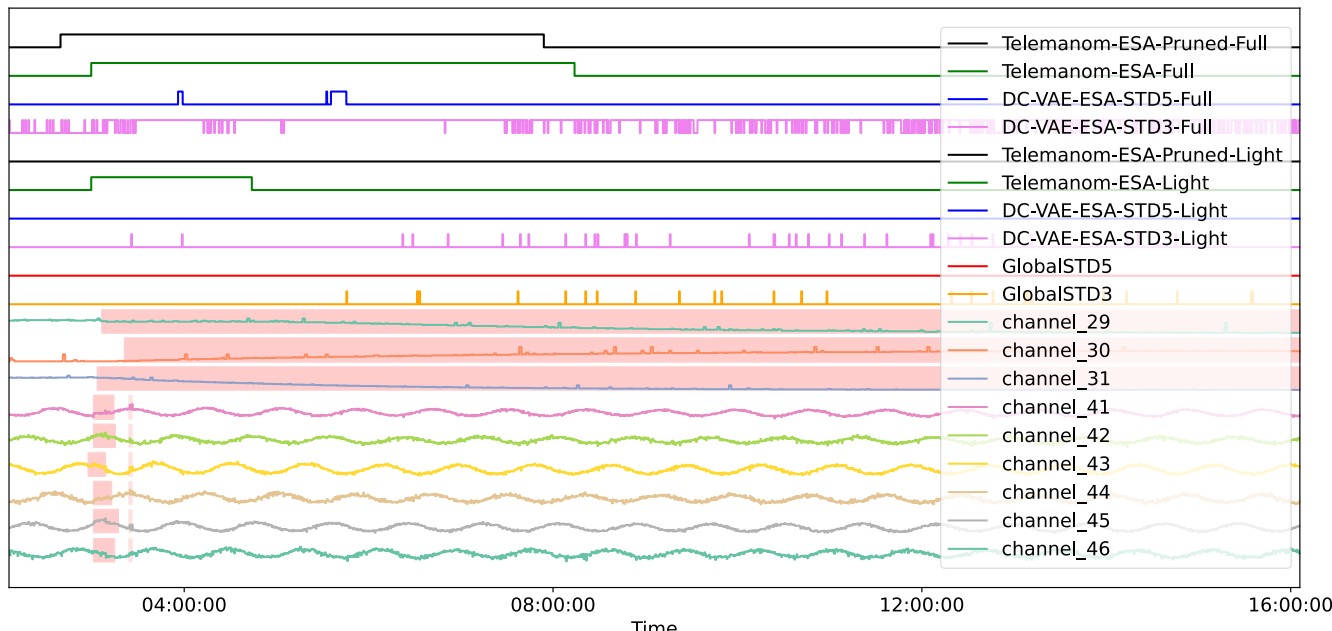

**Fig. 22.** Detection of rare nominal event id_159 (marked in red) for Mission1. Only Telemanom-ESA was able to correctly detect this event in both lightweight and full sets of channels, with a good timing. DC-VAE-ESA-STD3-Full also seems to detected it relatively well.

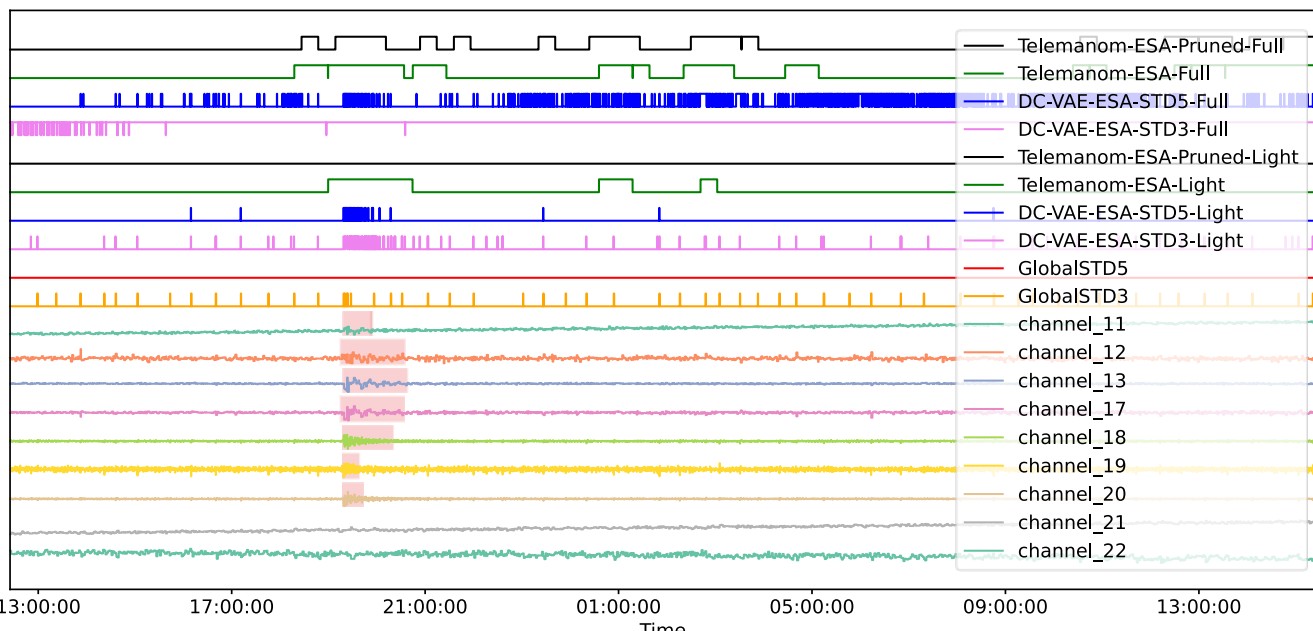

**Fig. 23.** Detection of anomaly id_631 (marked in red) for Mission2. This anomaly is not so easy to spot manually but was detected by most algorithms, surprisingly, not by Telemanom-ESA-Pruned-Light.

### 4.2. Results for anomalies only

The analysis of the results for anomalies alone (excluding rare nominal events and communication gaps) in Table 17 is important for understanding the performance of the algorithms in detecting the actual anomalies desired by SOEs. In this analysis, any true positives, false positives, or false negatives related to events different than anomalies are ignored (see implementation details in Appendix Section 3.2.2). For Mission2, there are only 9 anomalies in the full test set and only 4 anomalies in the lightweight test set (see Table 21), so the results should be interpreted with caution. A more reliable analysis can be conducted for Mission1 with 55 and 29 anomalies, respectively (see Table 20).

TABLE 17

BENCHMARKING RESULTS FOR DETECTION OF ANOMALIES ONLY IN LIGHTWEIGHT SUBSETS OF CHANNELS AND ALL CHANNELS FOR MISSION1 IN ESA-ADB. BOLDFACED RESULTS INDICATE THE BEST VALUES AMONG ALL ALGORITHMS (EXCLUDING AFTER RATIO OF ADTQC WHICH IS JUST A HELPER VALUE).

**Mission1 – trained and tested on the lightweight subset of channels 41-46 – only anomalies**

| Metric | | PCC | HBOS | iForest | Window iForest | KNN | Global STD3 | Global STD5 | DC-VAE-ESA STD3 | DC-VAE-ESA STD5 | Teleman-ESA | Teleman-ESA-Pruned |
|---|---|---|---|---|---|---|---|---|---|---|---|---|
| Event-wise | Precision | < 0.001 | < 0.001 | < 0.001 | < 0.001 | < 0.001 | < 0.001 | 0.205 | 0.001 | 0.021 | 0.074 | **0.999** |
| | Recall | 0.310 | 0.379 | 0.414 | 0.552 | 0.448 | 0.310 | 0.241 | 0.310 | 0.241 | **0.931** | 0.862 |
| | F0.5 | < 0.001 | < 0.001 | < 0.001 | < 0.001 | < 0.001 | < 0.001 | 0.211 | 0.001 | 0.026 | 0.090 | **0.968** |
| Channel-aware | Precision | Not available for unsupervised algorithms | | | | | 0.310 | 0.241 | 0.302 | 0.241 | **0.931** | 0.529 |
| | Recall | | | | | | 0.282 | 0.241 | 0.285 | 0.241 | **0.882** | 0.862 |
| | F0.5 | | | | | | 0.293 | 0.241 | 0.297 | 0.241 | **0.914** | 0.722 |
| Alarming precision | | 0.102 | 0.054 | 0.444 | 0.889 | 0.120 | 0.034 | 0.024 | 0.048 | 0.010 | 0.818 | **0.862** |
| ADTQC | After ratio | 0.889 | 0.636 | 0.500 | 0.063 | 0.385 | 0.889 | 1.000 | 1.000 | 1.000 | 0.037 | 0.040 |
| | Score | 0.826 | 0.676 | 0.730 | 0.308 | 0.670 | 0.826 | 0.919 | 0.911 | **0.921** | 0.220 | 0.159 |
| Affiliation-based | Precision | 0.536 | 0.543 | 0.532 | 0.562 | 0.521 | 0.561 | 0.919 | 0.559 | 0.906 | 0.774 | **0.927** |
| | Recall | 0.276 | 0.352 | 0.294 | 0.366 | 0.271 | 0.335 | 0.854 | 0.279 | 0.850 | 0.673 | **0.859** |
| | F0.5 | 0.451 | 0.490 | 0.458 | 0.508 | 0.440 | 0.494 | 0.906 | 0.466 | 0.894 | 0.752 | **0.912** |

**Mission1 – trained and tested on the full set of channels – only anomalies**

| Metric | | PCC | HBOS | iForest | Window iForest | KNN | Global STD3 | Global STD5 | DC-VAE-ESA STD3 | DC-VAE-ESA STD5 | Teleman-ESA | Teleman-ESA-Pruned |
|---|---|---|---|---|---|---|---|---|---|---|---|---|
| Event-wise | Precision | < 0.001 | < 0.001 | < 0.001 | Out-of-memory | Out-of-memory | < 0.001 | 0.001 | < 0.001 | 0.003 | 0.004 | **0.032** |
| | Recall | 0.891 | 0.964 | 0.945 | | | 0.873 | 0.818 | 0.891 | 0.818 | **0.945** | 0.909 |
| | F0.5 | < 0.001 | < 0.001 | < 0.001 | | | < 0.001 | 0.002 | < 0.001 | 0.004 | 0.005 | **0.039** |
| Subsystem-aware | Precision | Not available for unsupervised algorithms | | | | | 0.491 | **0.782** | 0.424 | 0.648 | 0.712 | 0.355 |
| | Recall | | | | | | 0.721 | 0.676 | 0.739 | 0.721 | 0.855 | **0.909** |
| | F0.5 | | | | | | 0.507 | **0.748** | 0.448 | 0.644 | 0.717 | 0.397 |
| Channel-aware | Precision | Not available for unsupervised algorithms | | | | | 0.327 | 0.355 | 0.272 | 0.311 | **0.497** | 0.195 |
| | Recall | | | | | | 0.332 | 0.298 | 0.398 | 0.324 | **0.561** | 0.705 |
| | F0.5 | | | | | | 0.309 | 0.315 | 0.272 | 0.291 | **0.472** | 0.217 |
| Alarming precision | | 0.005 | 0.008 | 0.005 | | | 0.009 | 0.088 | 0.003 | **0.020** | 0.132 | 0.278 |
| ADTQC | After ratio | 0.633 | 0.415 | 0.423 | | | 0.708 | 0.756 | 0.673 | 0.733 | 0.327 | 0.380 |
| | Score | **0.611** | 0.553 | 0.633 | | | 0.728 | 0.654 | 0.730 | 0.654 | 0.561 | 0.536 |
| Affiliation-based | Precision | 0.527 | 0.512 | 0.501 | | | 0.521 | 0.531 | 0.512 | **0.531** | 0.512 | 0.611 |
| | Recall | 0.486 | 0.563 | 0.445 | | | 0.462 | 0.434 | 0.452 | 0.473 | **0.344** | 0.436 |
| | F0.5 | 0.519 | 0.521 | 0.489 | | | 0.508 | 0.508 | 0.499 | 0.518 | **0.467** | 0.566 |

TABLE 18

BENCHMARKING RESULTS FOR DETECTION OF ANOMALIES ONLY IN LIGHTWEIGHT SUBSETS OF CHANNELS AND ALL CHANNELS FOR MISSION2 IN ESA-ADB. BOLDFACED RESULTS INDICATE THE BEST VALUES AMONG ALL ALGORITHMS (EXCLUDING AFTER RATIO OF ADTQC WHICH IS JUST A HELPER VALUE).

| Mission2 – trained and tested on the lightweight subset of channels 18-28 – only anomalies | | | | | | | | | | | |
|---|---|---|---|---|---|---|---|---|---|---|---|
| Metric | | PCC | HBOS | iForest | Window iForest | KNN | Global STD3 | Global STD5 | DC-VAE-ESA STD3 | DC-VAE-ESA STD5 | Teleman-ESA | Teleman-ESA-Pruned |
| Event-wise | Precision | < 0.001 | 0.000 | **0.004** | 0.000 | < 0.001 | < 0.001 | < 0.001 | < 0.001 | < 0.001 | 0.001 | 0.000 |
| | Recall | **1.000** | 0.000 | **1.000** | 0.000 | **1.000** | **1.000** | **1.000** | **1.000** | **1.000** | **1.000** | 0.000 |
| | F0.5 | < 0.001 | 0.000 | **0.005** | 0.000 | < 0.001 | < 0.001 | < 0.001 | < 0.001 | < 0.001 | 0.001 | 0.000 |
| Channel-aware | Precision | Not available for unsupervised algorithms | | | | | **1.000** | **1.000** | **1.000** | **1.000** | 0.600 | 0.000 |
| | Recall | | | | | | 0.667 | 0.667 | **1.000** | 0.667 | **1.000** | 0.000 |
| | F0.5 | | | | | | 0.909 | 0.909 | **1.000** | 0.909 | 0.652 | 0.000 |
| Alarming precision | | 0.032 | 0.000 | 0.143 | 0.000 | 0.029 | 0.026 | 0.036 | 0.027 | 0.037 | **1.000** | 0.000 |
| ADTQC | After ratio | 1.000 | - | 1.000 | - | 1.000 | 1.000 | 1.000 | 1.000 | 1.000 | 0.000 | - |
| | Score | **1.000** | - | **1.000** | - | **1.000** | **1.000** | **1.000** | **1.000** | **1.000** | 0.358 | - |
| Affiliation-based | Precision | 0.845 | 0.500 | **1.000** | 0.500 | 0.705 | 0.826 | 0.894 | 0.816 | 0.950 | 0.781 | 0.500 |
| | Recall | 0.925 | 0.000 | 0.971 | 0.000 | 0.517 | 0.862 | 0.994 | 0.888 | 0.981 | **1.000** | 0.000 |
| | F0.5 | 0.860 | 0.000 | **0.994** | 0.000 | 0.657 | 0.833 | 0.912 | 0.830 | 0.956 | 0.817 | 0.000 |
| **Mission2 – trained and tested on the full set of channels – only anomalies** | | | | | | | | | | | |
| Metric | | PCC | HBOS | iForest | Window iForest | KNN | Global STD3 | Global STD5 | DC-VAE-ESA STD3 | DC-VAE-ESA STD5 | Teleman-ESA | Teleman-ESA-Pruned |
| Event-wise | Precision | **0.001** | < 0.001 | < 0.001 | < 0.001 | Out-of-memory | < 0.001 | **0.001** | < 0.001 | < 0.001 | **0.001** | **0.001** |
| | Recall | 0.667 | 0.667 | 0.667 | 0.500 | | 0.667 | 0.167 | 0.833 | 0.667 | **1.000** | **1.000** |
| | F0.5 | **0.001** | < 0.001 | < 0.001 | < 0.001 | | < 0.001 | **0.001** | < 0.001 | < 0.001 | **0.001** | **0.001** |
| Subsystem-aware | Precision | Not available for unsupervised algorithms | | | | | 0.167 | 0.000 | 0.333 | 0.333 | **0.417** | 0.278 |
| | Recall | | | | | | 0.167 | 0.000 | 0.500 | 0.333 | 0.833 | **1.000** |
| | F0.5 | | | | | | 0.167 | 0.000 | 0.352 | 0.333 | **0.452** | 0.324 |
| Channel-aware | Precision | Not available for unsupervised algorithms | | | | | 0.083 | 0.000 | 0.095 | 0.111 | **0.296** | 0.082 |
| | Recall | | | | | | 0.021 | 0.000 | 0.229 | 0.042 | 0.573 | **0.833** |
| | F0.5 | | | | | | 0.052 | 0.000 | 0.098 | 0.083 | **0.325** | 0.096 |
| Alarming precision | | 0.364 | 0.308 | 0.143 | 0.158 | | 0.031 | **1.000** | 0.026 | 0.040 | 0.375 | 0.462 |
| ADTQC | After ratio | 0.750 | 0.500 | 0.500 | 0.333 | | **1.000** | **1.000** | 0.600 | 0.750 | 0.500 | 0.500 |
| | Score | 0.542 | 0.489 | 0.493 | 0.437 | | **0.992** | 0.612 | 0.698 | 0.709 | 0.660 | 0.766 |
| Affiliation-based | Precision | 0.660 | 0.608 | 0.618 | 0.616 | | 0.523 | 0.500 | 0.539 | 0.418 | **0.671** | 0.620 |
| | Recall | 0.380 | 0.333 | 0.358 | 0.355 | | 0.318 | 0.000 | 0.522 | 0.398 | **0.709** | 0.604 |
| | F0.5 | 0.575 | 0.522 | 0.539 | 0.537 | | 0.466 | 0.000 | 0.536 | 0.414 | **0.678** | 0.617 |

European Space Agency Dataset and Benchmark for Anomaly Detection in Real-World Time Series

### 4.3. Results for lightweight test sets using algorithms trained on full sets

For algorithms that provide separate anomaly scores for each channel, it is possible to limit the analysis of the global scores to an arbitrary subset of the channels used in training. It is especially useful to directly compare the results between models trained on full sets of channels and models trained only on lightweight subsets. Such a comparison is presented in Table 19 for the DC-VAE-ESA and Telemanom-ESA algorithms. GlobalSTD is omitted because its results do not depend on the number of training channels.

TABLE 19

BENCHMARKING RESULTS FOR DETECTION OF ALL EVENTS IN LIGHTWEIGHT TEST SETS IN ESA-ADB BY ALGORITHMS TRAINED ON LIGHTWEIGHT AND FULL SETS OF CHANNELS. BOLDFACED RESULTS INDICATE THE BETTER VALUE FOR EACH PAIR OF TRAINING SETS FOR EACH ALGORITHM (EXCLUDING AFTER RATIO OF ADTQC WHICH IS JUST A HELPER VALUE)

| Mission1 – tested on the lightweight test set | | | | | | | | |
|---|---|---|---|---|---|---|---|---|
| Algorithm → | | DC-VAE-ESA STD3 | | DC-VAE-ESA STD5 | | Teleman-ESA | | Teleman-ESA-Pruned | |
| Trained on → | | Light | Full | Light | Full | Light | Full | Light | Full |
| Event-wise | Precision | 0.001 | **0.008** | 0.014 | **0.216** | **0.148** | 0.027 | **0.999** | 0.043 |
| | Recall | **0.576** | 0.167 | **0.318** | 0.076 | **0.894** | 0.439 | 0.424 | 0.848 |
| | F0.5 | 0.001 | **0.009** | 0.017 | **0.158** | **0.178** | 0.033 | **0.786** | 0.054 |
| Channel-aware | Precision | **0.568** | 0.167 | **0.318** | 0.076 | **0.894** | 0.439 | 0.424 | **0.833** |
| | Recall | **0.442** | 0.101 | **0.207** | 0.066 | **0.738** | 0.328 | 0.275 | **0.848** |
| | F0.5 | **0.506** | 0.134 | **0.262** | 0.071 | **0.837** | 0.377 | 0.362 | **0.834** |
| Alarming precision | | 0.052 | **0.072** | 0.034 | **0.119** | **0.868** | 0.659 | **0.875** | 0.505 |
| ADTQC | After ratio | 0.921 | 0.909 | **0.952** | 0.800 | 0.136 | 0.586 | 0.143 | 0.286 |
| | Score | **0.805** | 0.607 | **0.799** | 0.728 | 0.428 | **0.625** | 0.197 | **0.431** |
| Affiliation-based | Precision | **0.577** | 0.562 | **0.741** | 0.524 | **0.727** | 0.616 | **0.711** | 0.621 |
| | Recall | **0.373** | 0.238 | **0.555** | 0.071 | **0.662** | 0.400 | 0.423 | **0.512** |
| | F0.5 | **0.520** | 0.441 | **0.694** | 0.231 | **0.713** | 0.556 | **0.626** | 0.596 |
| Mission2 – tested on the lightweight test set | | | | | | | | |
| Algorithm → | | DC-VAE-ESA STD3 | | DC-VAE-ESA STD5 | | Teleman-ESA | | Teleman-ESA-Pruned | |
| Trained on → | | Light | Full | Light | Full | Light | Full | Light | Full |
| Event-wise | Precision | **0.003** | **0.003** | **0.064** | 0.017 | **0.188** | 0.152 | **0.978** | 0.268 |
| | Recall | **1.000** | **1.000** | **1.000** | **1.000** | 0.986 | **0.989** | 0.540 | **0.911** |
| | F0.5 | 0.003 | **0.004** | **0.079** | 0.021 | **0.224** | 0.183 | **0.842** | 0.312 |
| Channel-aware | Precision | 0.904 | **0.912** | **0.995** | 0.985 | 0.831 | **0.875** | 0.465 | **0.690** |
| | Recall | **0.554** | 0.543 | **0.451** | 0.445 | **0.870** | 0.739 | 0.384 | **0.848** |
| | F0.5 | 0.787 | **0.788** | **0.783** | 0.772 | 0.822 | **0.823** | 0.442 | **0.708** |
| Alarming precision | | 0.052 | **0.034** | **0.068** | 0.046 | **0.912** | 0.907 | **0.862** | 0.861 |
| ADTQC | After ratio | 0.908 | 0.848 | 0.991 | 0.966 | 0.087 | 0.105 | 0.351 | 0.350 |
| | Score | **0.996** | 0.934 | **0.997** | 0.989 | 0.507 | **0.508** | **0.757** | 0.676 |
| Affiliation-based | Precision | **0.680** | 0.675 | **0.939** | 0.914 | **0.688** | 0.681 | **0.759** | 0.738 |
| | Recall | 0.293 | **0.345** | **0.788** | 0.782 | **0.544** | 0.503 | 0.530 | **0.623** |
| | F0.5 | 0.538 | **0.566** | **0.904** | 0.884 | **0.654** | 0.636 | 0.699 | **0.712** |

*4.4. Results for different mission phases*

It is a common practice to periodically retrain or adapt algorithms when new telemetry becomes available from satellites, especially in the presence of significant changes in operational conditions. The experiments in this section simulate such an approach in ESA-ADB to assess the robustness of algorithms to changing conditions and to identify the earliest mission phase in which reliable detectors can be trained. These aspects are crucial for the selection of algorithms in different mission phases. Some classic algorithms may perform much better than others in early mission phases when very limited data is available, but they may be overcome by deep learning techniques in late mission phases. The goal of this section is to provide a basic analysis of these aspects in ESA-ADB. For this purpose, the effect of training set size (representing different mission phases) on the corrected event-wise F0.5-score for the test set is analyzed for the lightweight subsets of each. The analysis for full sets is not conducted as the scores are very low even for the longest training set. There are 5 training set lengths (phases) proposed for Mission1 and 4 for Mission2 following the idea presented in Fig. 24. Starting from just a few percent of the mission timeline (initial phases) to 50% of the mission (the default setting in ESA-ADB). The statistics of the phases are listed in Table 20 (Mission1) and Table 21 (Mission2).

As visualized in Table 22, there is a clear correlation between the training set length and the event-wise F0.5 scores for test sets for both missions. Especially significant improvements are visible between phases 2 and 3 for Mission1 and phases 1 and 2 for Mission 2. A clear example is Windowed iForest for which the event-wise F0.5-score goes from 0.020 to 0.901 for Mission2 in the phase 2. Based on this observation, the minimal reasonable training length can be estimated to be 21 months for Mission1 and 5 months for Mission2. Surprisingly, the longest training sets do not always ensure the best results. There are some exceptions for which training on the longest training set does not give optimal results, i.e., PCC, DC-VAE-ESA, and Telemanom-ESA. We can only hypothesize what is the reason behind that, but it may be related to the concept drift present in the data.

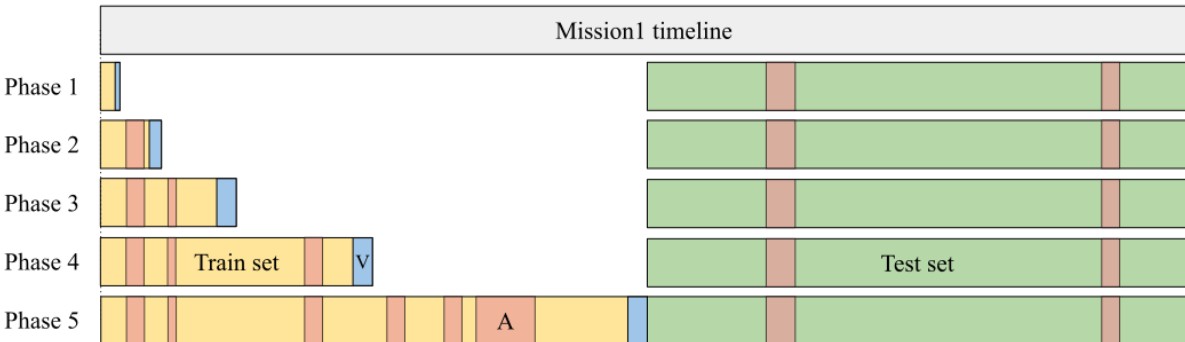

Fig. 24. Illustration of the idea of mission phases for Mission1. "A" marks light red anomalous fragments and "V" marks blue validation fragments.

TABLE 20

STATISTICS OF TRAINING, VALIDATION, AND TEST SETS FOR DIFFERENT PHASES OF MISSION1 CONSIDERING THE FULL SET (TOP PANEL) AND THE LIGHTWEIGHT SUBSET OF CHANNELS (BOTTOM PANEL)

| Mission1 – the lightweight subset | Phase 1 | | Phase 2 | | Phase 3 | | Phase 4 | | Phase 5 | | Test |
|---|---|---|---|---|---|---|---|---|---|---|---|
| | Train | Val | Train | Val | Train | Val | Train | Val | Train | Val | |
| **Data points** | **1,125,600** | **314,399** | **3,900,977** | **997,530** | **8,900,105** | **1,479,360** | **19,171,279** | **1,463,274** | **39,774,080** | **1,479,370** | **40,925,288** |
| Telecommands' executions | 7,769 | 15,918 | 94,426 | 45,194 | 271,882 | 13,295 | 414,927 | 9,001 | 764,648 | 60,157 | 769,917 |
| Duration (anonymised) | 9 weeks | 3 weeks | 8 months | 2 months | 18 months | 3 months | 39 months | 3 months | 81 months | 3 months | 84 months |
| Annotated points [%] | 1.41 | 17.29 | 2.76 | 1.49 | 3.24 | 0.02 | 1.84 | 0.11 | 1.74 | 1.23 | 1.81 |
| **Annotated events** | **1** | **1** | **6** | **3** | **17** | **2** | **28** | **1** | **52** | **3** | **65** |
| Anomalies | 0 | 1 | 3 | 1 | 5 | 0 | 10 | 0 | 22 | 2 | 29 |
| Rare nominal events | 1 | 0 | 3 | 2 | 9 | 2 | 14 | 1 | 26 | 1 | 36 |
| Communication gaps | 0 | 0 | 0 | 0 | 3 | 0 | 4 | 0 | 4 | 0 | 0 |
| Univariate / Multivariate | 0 / 1 | 0 / 1 | 0 / 6 | 0 / 3 | 0 / 14 | 0 / 2 | 0 / 24 | 0 / 1 | 0 / 48 | 0 / 3 | 1 / 64 |
| Global / Local | 1 / 0 | 1 / 0 | 4 / 2 | 2 / 1 | 11 / 3 | 1 / 1 | 18 / 6 | 1 / 0 | 39 / 9 | 3 / 0 | 40 / 25 |
| Point / Subsequence | 0 / 1 | 0 / 1 | 0 / 6 | 0 / 3 | 0 / 14 | 0 / 2 | 0 / 24 | 0 / 1 | 1 / 47 | 2 / 1 | 9 / 56 |
| **Distinct event classes** | **1** | **1** | **5** | **2** | **10** | **2** | **15** | **1** | **17** | **2** | **13** |

| Mission1 - the full set | Phase 1 | | Phase 2 | | Phase 3 | | Phase 4 | | Phase 5 | | Test |
|---|---|---|---|---|---|---|---|---|---|---|---|
| | Train | Val | Train | Val | Train | Val | Train | Val | Train | Val | |
| **Data points** | **8,954,221** | **2,176,171** | **29,416,435** | **7,890,008** | **68,888,013** | **10,761,293** | **144,775,815** | **10,273,971** | **305,515,601** | **10,741,556** | **428,599,738** |
| Annotated points [%] | 2.11 | 10.62 | 1.87 | 1.52 | 1.96 | 0.93 | 1.32 | 0.03 | 1.33 | 1.62 | 2.25 |
| **Annotated events** | **5** | **4** | **20** | **8** | **54** | **4** | **73** | **1** | **104** | **5** | **91** |
| Anomalies | 4 | 1 | 13 | 2 | 27 | 2 | 40 | 0 | 59 | 4 | 55 |
| Rare nominal events | 1 | 3 | 7 | 6 | 24 | 2 | 29 | 1 | 41 | 1 | 36 |
| Communication gaps | 0 | 0 | 0 | 0 | 3 | 0 | 4 | 0 | 4 | 0 | 0 |
| Univariate / Multivariate | 3 / 2 | 3 / 1 | 11 / 9 | 5 / 3 | 31 / 20 | 0 / 4 | 31 / 38 | 0 / 1 | 31 / 69 | 0 / 5 | 1 / 90 |
| Global / Local | 3 / 2 | 4 / 0 | 11 / 9 | 5 / 3 | 36 / 15 | 1 / 3 | 44 / 25 | 1 / 0 | 67 / 33 | 3 / 2 | 43 / 48 |
| Point / Subsequence | 0 / 5 | 0 / 4 | 0 / 20 | 0 / 8 | 0 / 51 | 0 / 4 | 1 / 68 | 0 / 1 | 2 / 98 | 2 / 3 | 9 / 82 |
| **Distinct event classes** | **3** | **2** | **6** | **3** | **11** | **3** | **16** | **1** | **18** | **3** | **17** |

European Space Agency Dataset and Benchmark for Anomaly Detection in Real-World Time Series

TABLE 21

STATISTICS OF TRAINING, VALIDATION, AND TEST SETS FOR DIFFERENT PHASES OF MISSION2 CONSIDERING THE FULL SET (TOP PANEL) AND THE LIGHTWEIGHT SUBSET OF CHANNELS (BOTTOM PANEL). THERE ARE NO COMMUNICATION GAPS AND ALL EVENTS ARE OF SUBSEQUENCE TYPE, SO THESE STATISTICS ARE OMITTED.

| Mission2 – the lightweight subset | Phase 1 | | Phase 2 | | Phase 3 | | Phase 4 | | Test |
|---|---|---|---|---|---|---|---|---|---|
| | Train | Val | Train | Val | Train | Val | Train | Val | |
| **Data points** | **1,457,269** | **506,869** | **7,741,250** | **2,032,657** | **15,714,523** | **3,867,017** | **34,998,975** | **5,830,297** | **46,153,954** |
| Telecommands' executions | 34,185 | 11,694 | 179,930 | 48,313 | 372,643 | 93,496 | 815,370 | 130,968 | 1,077,677 |
| Duration (anonymised) | 3 weeks | 1 week | 4 months | 1 month | 8 months | 2 months | 18 months | 3 months | 21 months |
| Annotated points [%] | 0.83 | 0.49 | 2.62 | 2.02 | 1.94 | 3.10 | 3.74 | 1.02 | 2.02 |
| **Annotated events** | **14** | **4** | **83** | **19** | **140** | **27** | **246** | **27** | **349** |
| Anomalies | 0 | 0 | 2 | 2 | 11 | 2 | 18 | 0 | 4 |
| Rare nominal events | 14 | 4 | 81 | 17 | 129 | 25 | 228 | 27 | 345 |
| Univariate / Multivariate | 0 / 14 | 0 / 4 | 0 / 83 | 0 / 19 | 0 / 140 | 0 / 27 | 1 / 245 | 0 / 27 | 1 / 348 |
| Global / Local | 12 / 2 | 3 / 1 | 67 / 16 | 17 / 2 | 119 / 21 | 24 / 3 | 214 / 32 | 26 / 1 | 333 / 16 |
| **Distinct event classes** | **3** | **3** | **12** | **6** | **15** | **9** | **21** | **5** | **22** |
| **Mission2 – the full set** | Phase 1 | | Phase 2 | | Phase 3 | | Phase 4 | | Test |
| | Train | Val | Train | Val | Train | Val | Train | Val | |
| **Data points** | **13,914,918** | **4,841,396** | **74,356,579** | **19,067,743** | **151,093,710** | **37,624,768** | **338,658,318** | **56,746,734** | **444,603,954** |
| Annotated points [%] | 0.20 | 0.12 | 0.64 | 0.51 | 0.50 | 0.59 | 0.66 | 0.21 | 0.54 |
| **Annotated events** | **14** | **4** | **85** | **22** | **146** | **28** | **256** | **27** | **361** |
| Anomalies | 0 | 0 | 4 | 5 | 16 | 3 | 25 | 0 | 9 |
| Rare nominal events | 14 | 4 | 81 | 17 | 130 | 25 | 231 | 27 | 352 |
| Univariate / Multivariate | 0 / 14 | 0 / 4 | 1 / 84 | 2 / 20 | 3 / 143 | 1 / 27 | 5 / 251 | 0 / 27 | 4 / 357 |
| Global / Local | 12 / 2 | 3 / 1 | 67 / 18 | 17 / 5 | 120 / 26 | 24 / 4 | 217 / 39 | 26 / 1 | 340 / 21 |
| **Distinct event classes** | **3** | **3** | **14** | **8** | **18** | **10** | **24** | **5** | **26** |

European Space Agency Dataset and Benchmark for Anomaly Detection in Real-World Time Series

TABLE 22

THE EFFECT OF MISSION PHASE ON THE CORRECTED EVENT-WISE $F0.5$-SCORE FOR SELECTED ALGORITHMS TRAINED AND TESTED ON THE LIGHTWEIGHT SUBSETS OF CHANNELS FROM MISSIONS IN ESA-ADB. BOLDFACED RESULTS INDICATE THE BEST VALUES AMONG ALL PHASES.

**Mission1 – trained and tested on lightweight subset of channels 41-46**

| Phase | PCC | HBOS | iForest | Window iForest | KNN | Global STD3 | Global STD5 | DC-VAE-ESA STD3 | DC-VAE-ESA STD5 | Teleman-ESA | Teleman-ESA-Pruned |
|---|---|---|---|---|---|---|---|---|---|---|---|
| 1 | | | **< 0.001** | | | < 0.001 | 0.041 | < 0.001 | 0.007 | 0.059 | 0.227 |
| 2 | | | | | | | 0.037 | | 0.012 | 0.058 | 0.311 |
| 3 | | | | | | | 0.104 | 0.007 | **0.085** | 0.122 | 0.776 |
| 4 | | | | | | | 0.217 | **0.009** | 0.030 | **0.309** | 0.776 |
| 5 | | | | | | **0.001** | **0.253** | 0.003 | 0.075 | 0.178 | **0.786** |

**Mission2 – trained and tested on lightweight subset of channels 18-28**

| Phase | PCC | HBOS | iForest | Window iForest | KNN | Global STD3 | Global STD5 | DC-VAE-ESA STD3 | DC-VAE-ESA STD5 | Teleman-ESA | Teleman-ESA-Pruned |
|---|---|---|---|---|---|---|---|---|---|---|---|
| 1 | < 0.001 | < 0.001 | 0.006 | 0.020 | < 0.001 | < 0.001 | < 0.001 | < 0.001 | < 0.001 | 0.234 | 0.622 |
| 2 | **0.062** | 0.007 | 0.456 | 0.901 | | 0.001 | 0.011 | | 0.001 | **0.259** | 0.757 |
| 3 | 0.013 | 0.040 | 0.585 | 0.947 | | 0.006 | 0.014 | 0.001 | 0.009 | 0.253 | 0.731 |
| 4 | 0.036 | **0.068** | **0.609** | **0.949** | **0.001** | **0.007** | **0.075** | **0.003** | **0.079** | 0.224 | **0.842** |

European Space Agency Dataset and Benchmark for Anomaly Detection in Real-World Time Series

*4.5. Computational resources and limitations*

Experiments were run on three different machines:
1. Nvidia Tesla T4 GPU (16 GB VRAM), Intel Xeon Gold 5222 CPU 3.80 GHz, and 64 GB RAM, (for CPU-intensive and memory-intensive algorithms)
2. Nvidia 3060 RTX GPU (6 GB VRAM), Intel i7-10870H CPU 2.20 GHz, and 32 GB RAM
3. Nvidia 3090 RTX GPU (24 GB VRAM), Intel i7-8700H CPU 3.20 GHz, and 32 GB RAM (for GPU-intensive algorithms)

Given the limited resources, there are limits to the amount of time and memory that each algorithm can run. The algorithm is rejected with an *out-of-memory error* if Machine 1 goes out of RAM. Algorithms are rejected with an *out-of-time error* if it takes more than 5 days to train or test a CPU-intensive algorithm on Machine 1, or a GPU-intensive algorithm on Machine 3.

*4.6. Processing times*

Algorithms for satellite telemetry monitoring must not only be accurate but also fast enough to run in real-time on computational resources available to mission control and, in the extreme case, on board satellites. We measured the times of training (Table 23) and execution (Table 24) of algorithms on our hardware resources (listed in Appendix Section 4.5). These numbers are not directly comparable because the algorithms were run in parallel processes on different machines. They give a rough approximation of the computational burden of each algorithm based on a single run in ESA-ADB. The training and execution times do not include resampling which was done once as an intermediate step before all experiments. The resampling of the test sets took about 1.5 hours for Mission1 and around 1 hour for Mission2, both on Machine 2.

The deep learning-based Telemanom has the longest training and execution times (excluding the execution time of KNN for channels 18-28 of Mission2), but it is still fast enough to provide real-time anomaly detection in both missions using the proposed resampling (0.033 Hz for Mission1, 0.056 Hz for Mission2). The total execution time (including resampling) for the full Mission1 test set is 3.5h which is just 0.02% of the test set duration, for Mission2 it is 4.5h and 0.08%, respectively. Thus, real-time execution should be possible even for sampling rates higher than 30 Hz. Moreover, in our previous works, we have shown that Telemanom can be run in real-time on-board the OPS-SAT satellite with a limited number of channels [55]. The important advantage of simple algorithms is that they are very fast and their training and execution times do not grow significantly with the number of channels, so it may be feasible to retrain them frequently during a mission.

TABLE 23
TRAINING TIMES (IN SECONDS) OF ALGORITHMS USED IN ESA-ADB

| Algorithm | Mission1 train set | | Mission2 train set | |
|---|---|---|---|---|
| | channels 41-46 | Full | channels 18-28 | Full |
| PCC | 90 | 143 | 63 | 75 |
| HBOS | 110 | 111 | 66 | 68 |
| iForest | 655 | 714 | 345 | 308 |
| Windowed iForest | 2833 (0.8h) | Out-of-memory | 1998 (0.6h) | 14585 (4h) |
| KNN | 3844 (1h) | Out-of-memory | 4754 (1.5h) | Out-of-memory |
| GlobalSTD | 101 | 108 | 60 | 90 |
| DC-VAE-ESA | 13466 (3.7h) | 18210 (5h) | 12440 (3.5h) | 4679 (1.3h) |
| Telemanom-ESA | 13115 (3.6h) | 30451 (8.5h) | 19725 (5.5h) | 12328 (3.5h) |

TABLE 24
EXECUTION TIMES (IN SECONDS) OF ALGORITHMS USED IN ESA-ADB

| Algorithm | Mission1 test set | | Mission2 test set | |
|---|---|---|---|---|
| | channels 41-46 | Full | channels 18-28 | Full |
| PCC | 124 | 141 | 73 | 76 |
| HBOS | 135 | 137 | 74 | 76 |
| iForest | 393 | 369 | 199 | 174 |
| Windowed iForest | 586 | Out-of-memory | 381 | 939 |
| KNN | 1233 | Out-of-memory | 21673 (6h) | Out-of-memory |
| GlobalSTD | 178 | 182 | 95 | 289 |
| DC-VAE-ESA | 5251 (1.5h) | 6010 (1.7h) | 3068 (0.9h) | 7900 (2.2h) |
| Telemanom-ESA | 6931 (1.9h) | 7271 (2h) | 4666 (1.3h) | 11078 (3.1h) |