# OpenReview forum: "European Space Agency Dataset and Benchmark for Anomaly Detection in Real-World Time Series"
_NeurIPS.cc/2025/Datasets_and_Benchmarks_Track — Submitted to NeurIPS 2025 Datasets and Benchmarks Track_

### Official Review · Reviewer_h7mA · 2025-06-29

**Rating:** 4
**Confidence:** 4

**Summary:**

The paper presents ESA-ADB, a new benchmark developed by the European Space Agency for anomaly detection in spacecraft telemetry time series. It features an extensive dataset comprising raw telemetry data from three ESA missions, including 224 channels, 821 telecommands, and 1,430 expert-annotated events. The benchmark introduces a rigorous evaluation framework with nine practical requirements and five specialized metrics to assess algorithm performance in realistic operational settings. Initial results show that commonly used anomaly detection methods fall short, emphasizing the dataset’s value as a challenging and realistic testbed for future research.

**Dataset Code Accessibility:**

Yes

**Ethical Considerations:**

No, there are no or only very minor ethics concerns

**Final Justification:**

The authors have addressed part of my concerns, while there is still concern regarding limited baselines. Thereby I keep my score as borderline accept.

**Limitations Weaknesses:**

1. Although the motivation and dataset is promising, there is still one concern regarding the generalizability of the TSAD approach trained from the data provided by one spacecraft to the others. As we know that the technology develops quikly, the spacecraft in the future may not use the same type of the sensors and structures, thereby the generalizability of the TSAD approach to other spacecraft is very important. The authors are encouraged to provide more discussion regarding this concern.


2. It would be also interesting to benchmark several recent works for time series anomaly detection, e..g, [1] [2] [3].

[1] Zhou, Qihang, et al. "Label-free multivariate time series anomaly detection." IEEE Transactions on Knowledge and Data Engineering 36.7 (2024): 3166-3179.

[2] Boniol, Paul, et al. "Adecimo: Model selection for time series anomaly detection." 2024 IEEE 40th International Conference on Data Engineering (ICDE). IEEE, 2024.

[3] Chen, Feiyi, et al. "Lara: A light and anti-overfitting retraining approach for unsupervised time series anomaly detection." Proceedings of the ACM Web Conference 2024. 2024.

[4] Xu, Hongzuo, et al. "Calibrated one-class classification for unsupervised time series anomaly detection." IEEE Transactions on Knowledge and Data Engineering (2024).

3. As reported by [5], there are different ways to conduct F1 evaluation, e.g., F1P A: F1 score with point-adjust; F1: the standard point-wise F1 score; F1T : time-series range-wise F1 score. The authors are also encouraged to deliver more discussion regarding the TSAD evaluation in these various manners when F1 is selected as evaluation metric.

[5] Sarfraz, M. Saquib, et al. "Position: quo vadis, unsupervised time series anomaly detection?." arXiv preprint arXiv:2405.02678 (2024).


4. The authors are encouraged to add one more table to make comparison with other TSAD dataset regarding some statistics.

5. Lack of the description of the quality control of the annotation, the authors are encouraged to deliver more discussion on this concern.


6. The authors are encouraged to provide more qualitative results of the evaluated baselines for TSAD in the main paper.

**Strengths Contributions:**

1. The paper is driven by a clear and impactful motivation, automating anomaly detection in spacecraft telemetry to reduce human error and operational costs. This addresses a pressing need in space mission control and gives the work strong practical significance. I believe this dataset will contribute to the community.

2. The authors provide an unparalleled dataset containing raw telemetry from three ESA spacecraft, annotated with expert input. Its size, diversity, and fidelity to real-world conditions make it a valuable resource for developing and testing anomaly detection methods.

3. The benchmark benefits from close collaboration between machine learning researchers and spacecraft operations engineers, ensuring that the data selection, annotations, and evaluation framework are aligned with the realities of operational environments.

---

> ### Author Rebuttal · Authors · 2025-07-30
>
> Thank you for your thoughtful review! We appreciate your time and valuable feedback, and have done our best to address your "Limitations Weaknesses" comments below. We hope that our responses resolve your concerns.
>
> 1. As mentioned in Section 2.1 “Dataset collection and curation” of the manuscript, we selected 3 missions of different types, purposes, and orbits to cover a diverse spectrum of sensors and use cases. Channels in the dataset cover common subsystems of all modern satellites (see annotation under Table 1.) that are unlikely to drastically change in next years. Moreover, we argue that the most important generalizability aspect of our work is on the application of algorithms in real-world scenarios with hundreds of channels (growing with each new generation of satellites), spatiotemporal dependencies between channels (common to all spacecrafts), diverse data types, telecommands, and data gaps (inevitable for data collected on the ground).
> We agree that this discussion would be worth mentioning in the camera-ready version of the paper.
>
> 2. As mentioned the Section 3.1 “Algorithms selection” of the manuscript, more advanced algorithms usually need a significant adaptation to meet the real-world requirements mentioned in Table 2. It was already time consuming to adapt the TimeEval framework and selected algorithms (i.e., Telemanom and DC-VAE) to real-world settings, so we decided to stop further exploration (especially after unsuccessful experiments with transformers described in the Appendix Section 3.4.4. “Experiments with transformers”), and engage community into this process by organizing the Kaggle competition with continuous evaluation. We deemed our current selection of algorithms enough to conclude that there is a significant gap between good performance of algorithms in current benchmarks and their usefulness in real-world scenarios.
>
> 3. We are aware of the mentioned paper by Sarfraz et al. (2024). However, the corrected event-wise F-score evaluation adopted in our benchmark is based on the paper by Sehili and Zhang (2024) which discusses disadvantages of point-adjust and point-wise scores in a similar way as Sarfraz et al. (2024). We provide a paragraph on this topic in lines 239-246 of the manuscript. In terms of the range-wise F-score assessment, we use the affiliation-based F-score by Huet et al. (2024) as one of the metrics. It is discussed in detail in the Appendix Section 3.2.3. "ESA-ADB metrics definitions" and it also gives arguments equivalent to Sarfraz et al. (2024). We will be happy to refer to Sarfraz et al. (2024) in the camera-ready version of the manuscript to make the discussion complete.
>
> M. El Amine Sehili and Z. Zhang, “Multivariate Time Series Anomaly Detection: Fancy Algorithms and Flawed Evaluation Methodology,” in Performance Evaluation and Benchmarking, R. Nambiar and M. Poess, Eds., Cham: Springer Nature Switzerland, 2024, pp. 1–17. doi: 10.1007/978-3-031-68031-1_1.
>
> A. Huet, J. M. Navarro, and D. Rossi, “Local Evaluation of Time Series Anomaly Detection Algorithms,” in Proceedings of the 28th ACM SIGKDD Conference on Knowledge Discovery and Data Mining, Washington DC USA: ACM, Aug. 2022, pp. 635–645. doi: 10.1145/3534678.3539339.
>
> 4. We provide a detailed comparison of statistics with other related TSAD datasets in Appendix Table 9. We guess that the comment asks to provide additional table for some other datasets, e.g., outside of the space telemetry domain, like SWaT, WADI, SMD, or Exathlon. We will be happy to do so in the camera-ready version of the manuscript.
>
> 5. The annotation process and quality control of the dataset are described in Section 2.1 “Dataset collection and curation” and Appendix 2.3 “Annotation details”. More details are given in our previous publication referenced in the bibliography [25] “Annotating Large Satellite Telemetry Dataset For ESA International AI Anomaly Detection Benchmark”. We established a consensus between spacecraft operations engineers and ML experts on how to annotate each specific type of anomaly. This consensus was later confronted with the results from algorithms in the iterative refinement process described in [25].
>
> 6. We are limited by the maximum of 9 content pages enforced by the NeurIPS policies. Thus, we decided to include the qualitative results in the Appendix and focus more on the discussion of the results in the main text.

---

> > ### Comment · Reviewer_h7mA · 2025-08-02
> > **Further Questions**
> >
> > Thank you for your response.
> >
> > Regarding your answer **We provide a detailed comparison of statistics with other related TSAD datasets in Appendix Table 9. We guess that the comment asks to provide additional table for some other datasets, e.g., outside of the space telemetry domain, like SWaT, WADI, SMD, or Exathlon. We will be happy to do so in the camera-ready version of the manuscript.**. Could you please provide this table during the discussion period together with analysis? In that case reviewers could check the content.
> >
> > Regarding your answer **As mentioned the Section 3.1 “Algorithms selection” of the manuscript, more advanced algorithms usually need a significant adaptation to meet the real-world requirements mentioned in Table 2. It was already time consuming to adapt the TimeEval framework and selected algorithms (i.e., Telemanom and DC-VAE) to real-world settings, so we decided to stop further exploration (especially after unsuccessful experiments with transformers described in the Appendix Section 3.4.4. “Experiments with transformers”), and engage community into this process by organizing the Kaggle competition with continuous evaluation. We deemed our current selection of algorithms enough to conclude that there is a significant gap between good performance of algorithms in current benchmarks and their usefulness in real-world scenarios.**, I still have the concern that baselines are not enough and not very recent. As the first benchmark built based on this new dataset, adapting existing works which are related to your task for experiments is quite common and should be presented to improve the significance of discussion.
> >
> > Thanks.

---

> > > ### Author Response · Authors · 2025-08-09
> > >
> > > As requested, below is the table with comparison of ESA-ADB with datasets from outside of the spacecraft telemetry domain
> > >
> > > | Dataset name | Number of channels              | Total volume          | Number of annotated events |
> > > |--------------|---------------------------------|-----------------------|----------------------------|
> > > | ESA-ADB      | 3 fragments with 1045 channels  | 2,296,122,157 samples | 1,430 (1.14% of samples)   |
> > > | SWaT         | 36 fragments with 51 channels   | 944,911 samples       | 36 (12.14% of samples)     |
> > > | WADI         | 15 fragments with 127 channels  | 962,172 samples       | 15 (5.77% of samples)      |
> > > | SMD          | 28 fragments with 38 channels   | 708,420 samples       | 67 (1.02% of samples)      |
> > > | TELCO        | 1 fragment with 12 channels     | 732,607 samples       | 36 (1.25% of samples)      |
> > > | Exathlon     | 93 fragments with 2283 channels | 5,332,588,023 samples | 97 (37.97% of samples)     |
> > >
> > > As already mentioned in the appendix, the Exathlon benchmark is the only related public dataset with volume comparable to ESA-ADB, but it has been criticized due to unrealistic anomaly density. Other datasets can be considered toy examples in comparison to ESA-ADB.
> > >
> > > We understand the concern about baselines. Unfortunately, there is nothing we can do about it at this stage, because adapting and training a new baseline for real-world data in ESA-ADB usually require substantial amount of time. However, during ECML PKDD conference, we will summarize the Kaggle challenge which will extend the list of new baselines.
> > >
> > > Regards

---

> > > > ### Comment · Reviewer_h7mA · 2025-08-09
> > > > **Response to the authors**
> > > >
> > > > Thank you for your detailed response, I will keep my score.

---

### Official Review · Reviewer_9xqc · 2025-06-29

**Rating:** 3
**Confidence:** 4

**Summary:**

This paper introduces ESA-ADB, a large-scale benchmark dataset for anomaly detection in spacecraft sensor time series, developed collaboratively by the European Space Agency and ML researchers. It contains several years of real telemetry data from 3 spacecraft, comprising 224 sensor channels, 821 control signals, and 1,430 annotated events — making it the most extensive dataset of its kind. The benchmark defines 9 domain-specific requirements and 5 evaluation metrics tailored to operational anomaly detection. Initial results show that current anomaly detection algorithms, even with adaptations, struggle with the dataset’s complexity. ESA-ADB presents a significant open challenge and is the focus of an ongoing Kaggle competition.

**Dataset Code Accessibility:**

Yes

**Dataset Code Comments:**

There is a link of the dataset, and a github link for the code.

**Ethical Considerations:**

No, there are no or only very minor ethics concerns

**Limitations Weaknesses:**

1. While the dataset is large, the paper may lack sufficient interpretability tools, baselines, or visualizations to help new researchers understand the data and its challenges.

2. The work focuses mainly on data release and benchmarking. There’s little to no innovation on modeling or detection algorithms beyond minor adaptations.

3. Although 5 metrics are provided, the paper could benefit from incorporating additional aspects such as uncertainty quantification or computational efficiency—important in real-time systems.

4. While it’s rich for space telemetry, it’s unclear whether the benchmark generalizes well to other high-dimensional time series domains like medical or industrial systems.

5. The process and consistency of the 1,430 anomaly annotations are not deeply discussed, raising concerns about subjectivity or label noise.

**Strengths Contributions:**

1. The dataset is derived from actual spacecraft telemetry, making it highly representative of real operational challenges in anomaly detection.

2. With data from 3 large spacecraft, 224 channels, 821 control signals, and 1,430 labeled anomalies, ESA-ADB is one of the largest and most detailed datasets in this domain.

3. The benchmark specifies 9 concrete requirements and 5 evaluation metrics, which provides a structured framework for model assessment beyond generic metrics like accuracy or AUC.

---

> ### Author Rebuttal · Authors · 2025-07-30
>
> Thank you for your thoughtful review! We appreciate your time and valuable feedback, and have done our best to address your "Limitations Weaknesses" comments  below. We hope that our responses resolve your concerns.
>
> 1. We are not sure how to understand the “interpretability tools and baselines” in this context, but for the visualization purposes, we recommend our free online OXI tool mentioned in the main text and provide a related helper script extract_fragments_for_OXI_annotator.py in our GitHub repository. Visualization notebooks are also provided in our GitHub repository and as a part of our Kaggle challenge (we cannot provide links due to NeurIPS policies of the rebuttal). In the appendix of the paper, we tried to include representative figures to help new researchers understand the data and its challenges.
>
> 2. Yes, the paper focuses on the dataset and benchmarking. That is why it was submitted specifically to the Datasets and Benchmarks Track which does not require novelty in the algorithmic part. Nevertheless, the proposed adaptations are minor in terms of algorithmic part, but very important in terms of practical applications in real-world scenarios.
>
> 3. The aspect of computational efficiency is discussed in the last sections of the Appendix “4.5. Computational resources and limitations” and “4.6. Processing times”. The inference time of all proposed algorithms allows for “real-time” execution with inference times shorter than the typical times between samples (~30s for Mission1 and ~18s for Mission2). This is why we did not direct more attention to this aspect. Additionally, the benchmark is primarily designed for the applications in ground operations (access to powerful GPUs and infrequently downlinked data packages from spacecrafts), not for the on-board applications in which the computational efficiency and uncertainty quantification would play critical roles. The aspect of uncertainty quantification was discussed during our project, but domain experts decided that it is a minor problem in ground operations - all alarms have to be anyway assessed in terms of certainty by human operators.
>
> 4. This aspect is not straightforward to assess and we do not claim in the manuscript that the benchmark generalizes to other domains (i.e., that algorithms with good results on our benchmark are likely to perform well also in medical or industrial scenarios). However, the benchmark does cover many common issues of real-world scenarios (high dimensionality, diverse data types, correlated channels, environmental variables, data gaps) that need to be resolved in practical applications. Thus, we claim that the benchmark generalizes well at least in terms of practical utility of algorithms.
>
> 5. The annotation process and quality control of the dataset are described in Section 2.1 “Dataset collection and curation” and Appendix 2.3 “Annotation details”. More details are given in our previous publication referenced in the bibliography [25] “Annotating Large Satellite Telemetry Dataset For ESA International AI Anomaly Detection Benchmark”.
> The dataset comes from real missions, so some level of subjectivity is inevitable, but we established a consensus between all spacecraft operations engineers on how to annotate each specific type of anomaly. This consensus was later confronted with the results from algorithms in the iterative refinement process described in [25].

---

> > ### Comment · Reviewer_9xqc · 2025-08-05
> >
> > My concerns have not been adequately addressed, so I will maintain my original score.

---

> > ### Author Response · Authors · 2025-08-06
> >
> > Would you be so kind to elaborate on how we could improve our work to address the comments, so we can improve the manuscript in the future? We are not sure what are your expectation, especially about comments 1, 4, and 5. Thanks.

---

> > > ### Comment · Reviewer_9xqc · 2025-08-06
> > >
> > > The benchmark shows compelling results for space telemetry data, but its applicability to other high-dimensional time series domains (e.g., medical or industrial systems) remains unclear. To strengthen the work’s broader relevance, consider briefly discussing how the chosen features or anomaly characteristics might translate—or pose challenges—to these domains. For instance, are there assumptions unique to spacecraft telemetry (e.g., sensor noise profiles, sampling rates) that could limit generalization? A short comparison to public benchmarks from other domains could help contextualize this.

---

> > > > ### Author Response · Authors · 2025-08-06
> > > >
> > > > Thanks. We would like to clarify that our benchmark does not aim to cover other domains (e.g., medical or industrial systems) and we do not make any such claims in the manuscript. We've just realized that the current title of the manuscript might inadvertently suggest broader applicability, so we propose changing "Time Series" to "Spacecraft Telemetry" in the title, resulting in: "European Space Agency Dataset and Benchmark for Anomaly Detection in Real-World Spacecraft Telemetry" - if this would address your doubts.

---

### Official Review · Reviewer_42eX · 2025-07-02

**Rating:** 4
**Confidence:** 3

**Summary:**

This paper introduces ESA-ADB, a new benchmark for time-series anomaly detection in spacecraft telemetry, developed in collaboration with the European Space Agency. It includes a large-scale real-world dataset (ESA-AD) from three missions, an evaluation framework with operationally relevant metrics and requirements, and baseline results from eight adapted algorithms. The benchmark addresses key challenges such as high dimensionality, nonstationarity, and irregular sampling, and is fully open-sourced with accompanying code and a Kaggle competition to encourage further research.

**Dataset Code Accessibility:**

Yes

**Ethical Considerations:**

No, there are no or only very minor ethics concerns

**Final Justification:**

I thank the authors for the detailed response and clarification provided. After reviewing the explanation and conducting my own assessment, I have decided to maintain my current rating.

**Limitations Weaknesses:**

1. Some algorithms (like KNN) fail to scale on the full-channel dataset due to memory or runtime issues, highlighting a need for more scalable TSAD methods.
2. In some missions, the event distribution is imbalanced, which may skew the evaluation. The anomaly-only metrics could be emphasized more in the main paper.
3. The selection of lightweight channel subsets seems subjective — adding quantitative justification or sensitivity analysis would strengthen the design.

**Strengths Contributions:**

1. This is the largest real-world spacecraft telemetry anomaly detection dataset to date, covering multiple missions, channels, and annotated events. The data is complex and closely reflects real-world conditions.
2. The evaluation framework is well thought out, with clearly defined operational requirements and targeted metrics.
3. Both the data and code are fully open-source, and the associated Kaggle competition encourages reproducibility and community engagement.
4. The benchmark was developed in collaboration with ESA engineers, ensuring practical relevance and alignment with real operational needs.
5. Baseline algorithms were adapted for real-world constraints, revealing current limitations in handling high-dimensional telemetry.

---

> ### Author Rebuttal · Authors · 2025-07-30
>
> Thank you for your thoughtful review! We appreciate your time and valuable feedback, and have done our best to address your "Limitations Weaknesses" comments below. We hope that our responses resolve your concerns.
>
> 1. We fully agree with this point. Our modification of Telemanom algorithm was mainly focused on this aspect – to make Telemanom more scalable.
>
> 2. If we understand correctly, the comment mentions the imbalance between the number of rare nominal events and anomalies. In the current ESA-ADB setting, we focus mainly on detecting both rare nominal events and anomalies (all outliers), because majority of popular TSAD methods cannot distinguish between these two categories effectively (as explained in the Appendix 3.2.5. “Approach for rare nominal events”). However, we included the anomaly-only metrics to highlight the ultimate practical goal and baseline for future works. We agree that this aspect would be worth emphasizing clearly in the camera-ready version of the paper.
>
> 3. We admit that the selection of lightweight subsets may seem subjective, because it was primarily directed by the suggestions from domain experts (e.g., these channels are commonly monitored by operators, they contain challenging anomalies to detect, and they can be analyzed in isolation from other channels to some extent). However, it was later quantitatively justified by ML experts based on the distribution of annotated events across channels, specifically, channels in the lightweight subsets are affected for a majority of annotated events – 118 out of 200 events (59% cases) in Mission1 and 622 out of 644 events (97% cases) in Mission2. Moreover, ML experts confirmed that a vast majority of challenging events (i.e., not detected by simple algorithms) are indeed included in the lightweight subsets. We agree that this justification will be worth mentioning in the camera-ready version of the paper.

---

> > ### Comment · Reviewer_42eX · 2025-08-06
> >
> > Thank you for answering my questions. Including some of these clarifications in the paper would be useful.

---

### Official Review · Reviewer_HVrP · 2025-07-02

**Rating:** 2
**Confidence:** 4

**Summary:**

In this paper, the authors introduce ESA-ADB, a comprehensive benchmark for anomaly detection in spacecraft sensor data, developed through collaboration between the European Space Operations Center and experts from academia and industry. Spacecraft time series are highly complex—characterized by high dimensionality, nonstationarity, nonlinearity, irregular sampling, and intricate spatial-temporal dependencies—making anomaly detection a critical yet challenging task for both ground and in-orbit operations. ESA-ADB addresses the lack of realistic benchmarks by providing a large-scale dataset containing several years of raw telemetry from three major spacecraft. This includes 224 sensor channels, 821 control signals, and 1,430 manually annotated events, making it the largest publicly available dataset of its kind. The benchmark also defines nine operational requirements and five evaluation metrics to guide and assess anomaly detection performance. Experimental results reveal that many commonly used algorithms, even when adapted, still fall short of the reliability needed for deployment in real-world space missions.

**Dataset Code Accessibility:**

Yes

**Dataset Code Comments:**

Code and dataset and algorithms are provided and seem easy to use

**Ethical Considerations:**

No, there are no or only very minor ethics concerns

**Final Justification:**

I appreciate the feedback from authors. Unfortunately, the authors are not willing to integrate recent benchmark practices, more baselines, etc., as requested by multiple reviewers. Therefore, I will maintain my score. The responses also indicate strong opinionated authors who, unfortunately, do not back up their claims with real facts. For example, the authors do not report results with established measures in the community because they claim their benchmark needs a binary decision for the anomalies. However, this is a design choice that can change. Ultimately, every benchmark wants such a binary decision to identify anomalies but that's impossible to happen in practice due to the continuous time-series domain (and not discrete like other domains). Hence, insisting on such design choices, make this benchmark extremely difficult to integrate with existing benchmarks and does not permit the use of existing methods, which is unreasonable.

**Limitations Weaknesses:**

W1. The paper would benefit from having more plots showing examples of the data and their anomalies. Current examples are limited and, honestly, demonstrate similar flaws to what has been discussed in the community for some years. It's unclear why some anomalies are marked and other, with similar patterns, are not.

W2. The paper contains several statements about flawed datasets and issues. The language needs to be toned down, especially considering that the provided dataset is not really solving any of those issues; on the contrary, from the plots, likely it contains similar mistakes. Your focus is on a specialized dataset, not a general-purpose one, so such strong language seems inappropriate for your goal

W3. There has been substantial progress in the community, despite criticisms. For example, wTSB-AD solves several such issues by hand-curating/selecting heterogenous datasets. Therefore, prior mentions on flaws etc. may be obsolete? or reduced at this point..

"The elephant in the room: Towards a reliable time-series anomaly detection benchmark." Advances in Neural Information Processing Systems 37 (2024): 108231-108261.

W4. Along with datasets, there is also substantial progress on how methods have to be evaluated. The authors follow criticism in datasets, but ignore criticism in evaluation measures, making their work incomplete.

"VUS: effective and efficient accuracy measures for time-series anomaly detection." The VLDB Journal 34.3 (2025): 32.

W5. Results are difficult to trust. We have F score close to 0 in many cases. This may be due to some difficulty but also likely demonstrates randomness/flaws (W1). What is the explanation for such low scores?

**Strengths Contributions:**

S1. New specialized benchmarks are necessary to advance time-series anomaly detection
S2. The dataset is unique, opening new challenges for the community
S3. A number of methods were adopted but their performance is limited, demonstrating the need for more attention

---

> ### Author Rebuttal · Authors · 2025-07-30
>
> We have done our best to address your concerns below. We hope that our responses will resolve them.
>
> W1:
> We provide 15 plots showing representative examples of the data and their anomalies (2 in the main text – Fig. 1-2, and 13 in the Appendix – Fig. 5-10 and Fig. 17-23). We believe these plots sufficiently convey the essential characteristics of the dataset. The comment does not give any specific examples of flaws in our dataset, so it is difficult to address it. However, a detailed discussion on how we address the main flaws from the literature is given in Table 10 of the Appendix.
>
> W2:
> We argue that our dataset addresses the aforementioned flaws and issues (as outlined in Table 10 of the Appendix), at least to the extent feasible in real-world datasets. Perhaps more importantly, it is the first dataset that enables a realistic assessment of anomaly detection, incorporating the full complexity of real-world data. We admit that our benchmark is specialized for spacecraft operations and our statements are based on our experience in this specific domain - where anomaly detection is one of the critical tasks. In the manuscript, we wanted to emphasize that popular datasets of spacecraft telemetry (and data of similar characteristics as listed in Appendix Section 2.6. “Comparison to related public datasets”), e.g., NASA SMAP and MSL, cannot be used for realistic assessment of algorithms and community should stop rely on them when designing algorithms for real missions.
> We would appreciate specific examples of the inappropriate strong language or statements that we should tone down. Also, the comment does not give any specific examples of mistakes in our dataset, so it is difficult to address this aspect.
>
> W3:
> Thank you for bringing the TSB-AD paper to our attention. We regret that we had no chance to acquaint ourselves with it before compiling our dataset in early 2024. It is good to see this progress in the domain. We will be happy to refer to TSB-AD in the camera-ready version of the manuscript, mentioning it as an optimistic step towards better evaluation.
> However, the main novelty of ESA-ADB over TSB-AD is still in its “real-world” setting as underlined by the title of our submission. TSB-AD, being a collection of simplified datasets (global annotations, relatively short fragments, no context drifts, uniform sampling, no missing data) assessed with point-wise and range-wise measures, is good as a generic benchmark for algorithms. While ESA-ADB focuses on practical aspects of anomaly detection systems. The “elephant in the room” revealed by ESA-ADB is that algorithms which perform well on benchmarks like TSB-AD often fail in operational settings, e.g., they care more about range-wise coverage than event-wise consistency (lack of redundant short alarms), they do not offer proper thresholding schemes, they cannot identify affected channels, they do not learn from historical anomalies, they struggle with larger numbers of channels, and they require significant modifications to handle variety of real-world problems.
>
> W4:
> We do not ignore criticism of evaluation measures. We mention it directly in the manuscript:
> - line 43: “many publicly available datasets, benchmarks, and metrics for TSAD are flawed and cannot be used for an unbiased evaluation of emerging machine learning (ML) techniques, especially in complex real-world settings [18–21].”
> - line 217: "Many recent advances criticize popular sample-wise and point-adjust protocols for being overoptimistic, and propose better alternatives [18, 19, 21, 30, 31, 40–47]".
> Moreover, a detailed analysis of all recent metrics is available in Section 3.2.1. of the Appendix "Analysis of metrics from the literature". It gives main reasons on why we selected or rejected specific metrics. This analysis includes the VUS (Volume Under the Surface) metric suggested by the reviewer.
>
> W5:
> The main explanation of event-wise F-scores close to 0 is given in lines 272-274: “Unsupervised algorithms perform very poorly for Mission1 in terms of event-wise scores. […] The main problem of these algorithms is a massive number of false detections caused by the noise and varying sampling rates in the data, as visible in the examples in Appendix 4.1”. The event-wise assessment (especially when using F0.5 score) strongly penalizes all point-wise false positives (much stronger than any range-based metrics). This is in accordance with the requirements posed by domain experts, because a large number of point-wise false positives is currently one of the main blockers for wider adoption of algorithms in practice.

---

> > ### Author Response · Authors · 2025-08-08
> >
> > Thanks for clarifications, we provide answers to specific questions and doubts below. Our main conclusion for now is that the main reasons to reject the paper are that we do not follow the setup proposed in the TSB-AD and do not use VUS as one of the metrics. However, while the goal of TSB-AD and VUS is to provide a generic benchmark for TSAD algorithms, ESA-ADB focuses on a specific real-world scenario (i.e., TSAD in space operations) that represents real needs and challenges when implementing TSAD algorithms in practice. TSB-AD and ESA-ADB are complementary works and have different goals. That is why we still claim that VUS in not well-suited for our task and SOTAs may differ between these benchmarks.
> >
> > **“Reviewers requested improvements in visualizations/examples”** – reviewers did not specify what is wrong with the current visualizations. Should we include visualizations of more events (which ones and how many?) or all 1430 events in the dataset?
> >
> > **“We saw the visualizations, and they are not good. For example, in Figure 10 in the appendix, what should an algorithm predict?”** – Figure 10 is actually meant to convey the important points that Reviewer asks about. The blue area is a rare nominal event directly related to telecommands available in the dataset and is expected by operators. The blue areas are not annotated similarly across channels, because channels are simply not affected in the same way by this event – it is one of the key features of our dataset. We know exactly where this event starts, but as mentioned in our article [25] *“Endpoints of anomalies are usually less important for SOEs and also harder to accurately and objectively define. [...] To provide consistent annotations, the endpoint of an anomaly was defined as the point at which the channel goes back to the previous nominal state or establishes a new one.”*. That’s also why you observe the difference in endpoints for the red areas, because channel_20 goes back to its “nominal state” earlier. The red area is a genuine anomaly which is completely not related to the following rare nominal event – it is caused by an unknown external force (e.g., micrometeorite impact). Majority of popular TSAD algorithms cannot distinguish these two categories, so we decided that in the current default setting of ESA-ADB should be to detect both. However, we provide a separate analysis for genuine anomalies in Appendix 4.2. “Results for anomalies only” and a proposed approach to evaluate rare nominal events once suitable approaches are developed.
> >
> > **“If I want to design a new method for this dataset, I have no idea where to start and what to achieve”** – This was exactly our first thought when we were initially asked by ESOC engineers to design an anomaly detection algorithm for the real-world space operations. Current TSAD datasets and benchmarks do not reflect complexities of real-world problems like this, so we had to create a new benchmark. Our work is the first comprehensive take on how to properly evaluate TSAD in real operations according to domain experts. The benefit for the community is clear: good results on ESA-ADB translate directly to usefulness of the algorithm in real missions.
> >
> > **"Following on that, reviewers requested more baselines."** – We understand the need of more baselines, but running a new algorithm on a large real-world dataset is not as straightforward as for simplified datasets. The primary example is our described modification of Telemanom - one of the most commonly used TSAD algorithms in spacecraft telemetry. It took us a lot of work to make it fulfil basic requirements of the real-world scenario. The final version described in the manuscript achieved reasonable results on the lightweight subsets of channels, confirming that the task is solvable, but for the full sets, the complexity of the task dramatically increases and shows an open problem for the future. This is why we launched the Kaggle competition to engage community into the process and check what is possible.
> >
> > **"But is it so unreasonable to report results on some other evaluation measures, which have been widely used in the community?"** – It is unreasonable to calculate metrics if they are not well-suited for the problem. Our task requires binary outputs from algorithms and does not prioritize precise anomaly interval identification, so VUS is not well-suited.
> >
> > **"Are you following similar problematic practices for your models/results?"** – The “problematic practices” mentioned by Reviewer are related to how TimeEval is used, not to how it works in general. Therefore, the suggestion that we follow the same practices is unfounded (however, we think that penalizing algorithms for going out-of-time or out-of-memory is justifiable in practical scenarios like ours). In fact, we’ve implemented some modifications to TimeEval to make it better suited for real-world scenarios (e.g., section 3.2 “Real-world evaluation of unsupervised algorithm”).

---

### Note · Authors · 2025-08-15

Concluding the main rebuttal points, we feel that our work is being penalized for addressing real-world challenges in time series anomaly detection (TSAD) instead of proposing a generic benchmark.

While reviewers expressed concerns about the universality of our benchmark, we would like to emphasize that we never claim universality in the manuscript. Our position is that existing TSAD benchmarks often prioritize simplicity and universality at the cost of realism. In contrast, our benchmark deliberately incorporates the full spectrum of real-world challenges, using spacecraft telemetry as the primary case study. This necessarily comes at the expense of universality, but it represents a crucial step toward enabling practical deployment of TSAD algorithms in real operational settings.

Reviewers requested the inclusion of additional algorithms, but a key point of our work is precisely that the majority of available implementations are not designed to meet the real-world requirements given in the paper. This is why we were only able to present results for 8 out of the 71 algorithms in the TimeEval framework. Each additional method would require a dedicated study to properly adapt it to the demands of our benchmark, as we demonstrated with Telemanom and DC-VAE. This challenge underscores the very motivation for our benchmark: to drive progress in adapting and developing methods suitable for deployment in realistic scenarios.

Finally, we acknowledge the reviewers’ concerns regarding the labeling process. Large-scale real-world datasets cannot be annotated with perfect objectivity, and some degree of subjectivity is inevitable. For this reason, we dedicated substantial effort to ensure consistent labeling, and published a separate paper [25] about annotation methodology, quality control measures, and design decisions. Moreover, the annotations were reviewed and approved by domain experts at ESA who work with this data on a daily basis.

---

### Decision · Program_Chairs · 2025-09-18

**Decision:**

Reject

**Comment:**

**Summary (1–2 lines):** Large, real-world spacecraft-telemetry TSAD benchmark (3 missions; 224 channels; 821 telecommands; 1,430 events) with 9 operational requirements, 5 metrics, code, and a companion Kaggle challenge.

**Strengths**

* Valuable, hard-to-obtain real telemetry with expert annotations; strong collaboration with ESA.
* Clear operational framing (event-wise alarms, channel attribution, latency constraints).
* Open data/code; baseline adaptations expose gaps between lab benchmarks and ops.

**Weaknesses (central to decision)**

* Baseline coverage is narrow and not up-to-date; limited alignment with widely used TSAD evaluation practices (e.g., range/point variants, VUS) and few raw cross-metric numbers.
* Figures/qualitative examples insufficient to unambiguously convey what constitutes an anomaly vs rare nominal events across channels; labeling/QA process not quantitatively evidenced (e.g., IAA).
* Some design choices (binary outputs, TimeEval adaptations, lightweight subsets) are insufficiently justified with sensitivity analyses; cross-mission/generalization evidence is limited.
* Croissant/metadata gaps and many inaccessible resources in the report reduce polish.

**Rebuttal & discussion:** Authors clarified scope (spacecraft-specific), proposed title narrowing, defended metric choices, added a cross-domain dataset table, and explained visualization/endpoint decisions. However, two reviewers’ core concerns (metrics alignment, stronger baselines, clearer visuals/labeling evidence) remain largely unresolved; scores stabilized at **2, 3, 4, 4**.

**Decision & rationale:** **Reject.** The dataset is promising and relevant, but current evidence, evaluation breadth, and presentation fall short of the bar for D\&B acceptance this year, especially given split reviews and unresolved methodological concerns.

**Notes to authors (actionable):**

* Report results under additional community metrics (range-F1 variants, point-wise, VUS), even if secondary; include raw numbers and rank correlations with your event-wise metric.
* Expand baselines with recent TSAD methods; document any adaptations, time/memory limits, and fairness controls.
* Strengthen qualitative guidance: per-event, multi-channel plots with clear legends; add a concise “what is the anomaly here?” caption; release labeling guidelines and IAA statistics.
* Provide cross-mission transfer and subset-selection sensitivity; discuss runtime/scale trade-offs.
* Fix Croissant/resource accessibility and populate RAI fields.
* If the scope is spacecraft-only, adopt the narrower title and state non-claims of generality up front.